# A PATH-NORM TOOLKIT FOR MODERN NETWORKS: CONSEQUENCES, PROMISES AND CHALLENGES

**Antoine Gonon, Nicolas Brisebarre, Elisa Riccietti & Rémi Gribonval**
Univ Lyon, EnsL, UCBL, CNRS, Inria, LIP, F-69342, LYON Cedex 07, France

## ABSTRACT

This work introduces the first toolkit around path-norms that fully encompasses general DAG ReLU networks with biases, skip connections and any operation based on the extraction of order statistics: max pooling, GroupSort etc. This toolkit notably allows us to establish generalization bounds for modern neural networks that are not only the most widely applicable path-norm based ones, but also recover or beat the sharpest known bounds of this type. These extended path-norms further enjoy the usual benefits of path-norms: ease of computation, invariance under the symmetries of the network, and improved sharpness on layered fully-connected networks compared to the product of operator norms, another complexity measure most commonly used.

The versatility of the toolkit and its ease of implementation allow us to challenge the concrete promises of path-norm-based generalization bounds, by numerically evaluating the sharpest known bounds for ResNets on ImageNet.

## 1 INTRODUCTION

Developing a thorough understanding of theoretical properties of neural networks is key to achieve central objectives such as efficient and trustworthy training, robustness to adversarial attacks (e.g. via Lipschitz bounds), or statistical soundness guarantees (via so-called generalization bounds).

The so-called path-norms and path-lifting are promising concepts to theoretically analyze neural networks: the $L^1$ path-norm has been used to derive generalization guarantees (Neyshabur et al., 2015; Barron & Klusowski, 2019), and the path-lifting has led for example to identifiability guarantees (Bona-Pellissier et al., 2022; Stock & Gribonval, 2023) and characterizations of properties of the dynamics of training algorithms (Marcotte et al., 2023).

Yet, current definitions of path-norms and of path-lifting are severely limited: they only cover simple models unable to combine in a single framework pooling layers, skip connections, biases, or even multi-dimensional output (Neyshabur et al., 2015; Kawaguchi et al., 2017; Bona-Pellissier et al., 2022; Stock & Gribonval, 2023). Thus, the promises of existing theoretical guarantees based on these tools are currently out of reach: they cannot even be tested on standard modern networks.

Due to the lack of versatility of these tools, known results have only been tested on toy examples. This prevents us from both understanding the reach of these tools and from diagnosing their strengths and weaknesses, which is necessary to either improve them in order to make them actually operational, if possible, or to identify without concession the gap between theory and practice, in particular for generalization bounds.

*This work adresses the challenge of making these tools fully compatible with modern networks, and to concretely assess them on standard real-world examples.* First, **it formalizes a definition of path-lifting (and path-norms) adapted to very generic ReLU networks**, covering any DAG architecture (in particular with skip connections), including in the presence of max/average-pooling (and even more generally $k$-max-pooling, which extracts the $k$-th largest coordinate, recovering max-pooling for $k = 1$) and/or biases. This covers a wide variety of modern networks (notably ResNets, VGGs, U-nets, ReLU MobileNets, Inception nets, Alexnet)[1], and recovers previously known definitions of these tools in simpler settings such as layered fully-connected networks.

---

[1]The conclusion discusses networks not covered by the framework and adaptations needed to cover them.

The immediate interests of these tools are: 1) *path-norms are easy to compute*[2] on modern networks via a single forward-pass; 2) *path-norms are invariant under neuron permutations and parameter rescalings that leave the network invariant*; and 3) *the path-norms yield Lipschitz bounds*. These properties were known (but scattered in the literature) in the restricted case of layered fully-connected ReLU networks primarily without biases (and without average/$k$-max-pooling nor skip connections) (Neyshabur et al., 2015; Neyshabur, 2017; Furusho, 2020; Jiang et al., 2020; Dziugaite et al., 2020; Bona-Pellissier et al., 2022; Stock & Gribonval, 2023). They are generalized here for generic DAG ReLU networks with most of the standard ingredients of modern networks (pooling, skip connections...), with the notable exception of the attention mechanism.

Moreover, *path-norms tightly lower bound products of operator norms*, another complexity measure that does not enjoy the same invariances as path-norms, despite being widely used for Lipschitz bounds (*e.g.*, to control adversarial robustness) (Neyshabur et al., 2018; Gonon et al., 2023) or generalization bounds (Neyshabur et al., 2015; Bartlett et al., 2017; Golowich et al., 2018). This bound, which was only known for *scalar-valued* layered fully-connected ReLU networks without biases (Neyshabur et al., 2015), is again generalized here to generic *vector-valued* DAG ReLU networks. This requires introducing so-called *mixed* path-norms and extending products of operator norms.

Second, **this work also establishes a new generalization bound for modern ReLU networks based on their corresponding $L^1$ path-norm**. This bound covers arbitrary output dimension (while previous work focused on scalar dimension, see Table 1), generic DAG ReLU network architectures with average/$k$-max-pooling, skip connections and biases. The achieved generalization bound recovers or beats the sharpest known ones of this type, that were so far only available in simpler restricted settings, see Table 1 for an overview. Among the technical ingredients used in the proof of this generalization bound, *the new contraction lemmas and the new peeling argument are among the main theoretical contributions of this work. The first new contraction lemma extends the classical ones with scalar $t_i \in \mathbb{R}$, and contractions $f_i$ of the form $\mathbb{E}_{\boldsymbol{\varepsilon}} g \left( \sup_{t \in T} \sum_{i \in I} \varepsilon_i f_i(t_i) \right) \leqslant \mathbb{E}_{\boldsymbol{\varepsilon}} g \left( \sup_{t \in T} \sum_{i \in I} \varepsilon_i t_i \right)$ (Ledoux & Talagrand, 1991, Theorem 4.12), with convex non-decreasing $g$, to situations where there are multiples independent copies indexed by $z \in Z$ of the latter: $\mathbb{E}_{\boldsymbol{\varepsilon}} \max_{z \in Z} g \left( \sup_{t \in T^z} \sum_{i \in I} \varepsilon_{i,z} f_{i,z}(t_i) \right) \leqslant \mathbb{E}_{\boldsymbol{\varepsilon}} \max_{z \in Z} g \left( \sup_{t \in T^z} \sum_{i \in I} \varepsilon_{i,z} t_i \right)$. The second new contraction lemma deals with vector-valued $t_i \in \mathbb{R}^W$, and functions $f_i$ that compute the $k$-th largest input's coordinate, to cope with $k$-max-pooling neurons, and it also handles multiple independent copies indexed by $z \in Z$.* The most closely related lemma we could find is the vector-valued one in Maurer (2016) established with a different technique, and that holds only for $g = \mathrm{id}$ with a single copy ($|Z| = 1$). *The peeling argument* reduces the Rademacher complexity of the whole model to that of the inputs by peeling off the neurons of the model one by one. This is inspired by the peeling argument of Golowich et al. (2018), which is however specific to layered fully-connected ReLU networks with layer-wise constraints on the weights. Substantial additional ingredients are developed to handle *arbitrary DAG ReLU networks* (as there is no longer such a thing as a *layer* to peel), *with not only ReLU but also $k$-max-pooling and identity neurons* (where Golowich et al. (2018) has only ReLU neurons), and leveraging *only a global constraint through the path-norm* (instead of layerwise constraints on operator norms). The analysis notably makes use of the rescaling invariance of the proposed generalized path-lifting.

The versatility of the proposed tools **enables us to compute for the first time generalization bounds based on the $L^1$ path-norm on networks really used in practice**. This is the opportunity to assess the current state of the gap between theory and practice, and to diagnose possible room for improvements. As a concrete example, we demonstrate that on ResNet18 trained on ImageNet: 1) the proposed generalization bound can be numerically computed; 2) for a (dense) ResNet18 trained with standard tools, roughly *30 orders of magnitude* would need to be gained for this path-norm based bound to match practically observed generalization error; 3) the same bound evaluated on a *sparse* ResNet18 (trained with standard sparsification techniques) is decreased by up to 13 orders of magnitude. We conclude the paper by discussing promising leads to reduce this gap.

**Paper structure.** Section 2 introduces the ReLU networks being considered, and generalizes to this model the central definitions and results related to the path-lifting, the path-activations and path-norms. Section 3 state a versatile generalization bound for such networks based on path-norm, and sketches its proof. Section 4 reports numerical experiments on ImageNet and ResNets. Related works are discussed along the way, and we refer to Appendix K for more details.

---

[2] Code for reproducibility is in Gonon et al. (2024).

Table 1: Generalization bounds (up to universal multiplicative constants) for a ReLU network estimator with parameters restricted to belong to some subset $\Theta \subseteq \mathbb{R}^G$ (Definition 2.2), learned from $n$ iid training points when 1) the loss $\hat{y} \in (\mathbb{R}^{d_{\text{out}}}, \|\cdot\|_2) \mapsto \ell(\hat{y}, y) \in \mathbb{R}$ is $L$-Lipschitz for every $y$, and 2) inputs are bounded in $L^\infty$-norm by $B \geqslant 1$. Here, $d_{\text{in}}/d_{\text{out}}$ are the input/output dimensions, $K = \max_{v \in N_{*\text{-pool}}} |\operatorname{ant}(v)|$ is the maximum kernel size (Definition 2.2) of the $*$-max-pooling neurons, $M_d$ is the matrix of layer $d$ for a *layered fully-connected* network (LFCN) without bias $R_{\boldsymbol{\theta}}(x) = M_D \operatorname{ReLU}(M_{D-1} \ldots \operatorname{ReLU}(M_1 x))$, $D$ is the depth. Note that having $r$ in the bound is more desirable than having $R$ since $r \leqslant R$ and $R$ can be arbitrarily large even when $r = 0$ (Theorem C.1 and Figure 2 in appendix). This is because $R$ decouples the layers without taking into account rescaling invariances.

| | Architecture | Parameter set $\Theta$ | Generalization bound |
|---|---|---|---|
| (Kakade et al., 2008, Eq. (5)) (Bach, 2024, Sec. 4.5.3) | LFCN with depth $D = 1$, no bias, $d_{\text{out}} = 1$ (linear regression) | $\|\boldsymbol{\theta}\|_1 = \|\Phi(\boldsymbol{\theta})\|_1 \leqslant r$ | $\frac{LB}{\sqrt{n}} \times r \sqrt{\ln(d_{\text{in}})}$ |
| (E et al., 2022, Thm. 6) (Bach, 2017, Proposition 7) | LFCN with $D = 2$, no bias, $d_{\text{out}} = 1$ (two-layer network) | $\|\Phi(\boldsymbol{\theta})\|_1 \leqslant r$ | $\frac{LB}{\sqrt{n}} \times r \sqrt{\ln(d_{\text{in}})}$ |
| (Neyshabur et al., 2015, Corollary 7) | DAG, no bias, $d_{\text{out}} = 1$ | $\|\Phi(\boldsymbol{\theta})\|_1 \leqslant r$ | $\frac{LB}{\sqrt{n}} \times 2^D r \sqrt{\ln(d_{\text{in}})}$ |
| (Golowich et al., 2018, Theorem 3.2) | LFCN with arbitrary $D$, no bias, $d_{\text{out}} = 1$ | $\prod_{d=1}^{D} \|M_d\|_{1,\infty} \leqslant R$ | $\frac{LB}{\sqrt{n}} \times R \sqrt{D + \ln(d_{\text{in}})}$ |
| (Barron & Klusowski, 2019, Corollary 2) | LFCN with arbitrary $D$, no bias, $d_{\text{out}} = 1$ | $\|\Phi(\boldsymbol{\theta})\|_1 \leqslant r$ | $\frac{LB}{\sqrt{n}} \times r \sqrt{D + \ln(d_{\text{in}})}$ |
| Here, Theorem 3.1 | DAG, *with biases*, arbitrary $d_{\text{out}}$, with ReLU, identity and $k$-max-pooling neurons for $k \in \{k_1, \ldots, k_P\} \subset \{1, \ldots, K\}$ | $\|\Phi(\boldsymbol{\theta})\|_1 \leqslant r$ | $\frac{LB}{\sqrt{n}} \times r \sqrt{D \ln(PK) + \ln(d_{\text{in}} d_{\text{out}})}$ |

## 2 ReLU MODEL AND PATH-LIFTING

Section 2.1 defines a general DAG ReLU model that covers modern architectures. Section 2.2 then introduces the so-called path-norms and extends related known results to this general model.

### 2.1 ReLU MODEL THAT COVERS MODERN NETWORKS

Definition 2.2 introduces the model considered here (extending, *e.g.*, DeVore et al. (2021)).

**Definition 2.1.** *The ReLU function is defined as* $\operatorname{ReLU}(x) := x \mathbb{1}_{x \geqslant 0}$ *for* $x \in \mathbb{R}$. *The $k$-max-pooling function* $k\text{-}\mathtt{pool}(x) := x_{(k)}$ *returns the $k$-th largest coordinate of* $x \in \mathbb{R}^d$.

**Definition 2.2.** *Consider a Directed Acyclic Graph (DAG)* $G = (N, E)$ *with edges* $E$, *and vertices* $N$ *called neurons. For a neuron* $v$, *the sets* $\operatorname{ant}(v), \operatorname{suc}(v)$ *of antecedents and successors of* $v$ *are* $\operatorname{ant}(v) := \{u \in N, u \to v \in E\}, \operatorname{suc}(v) := \{u \in N, v \to u \in E\}$. *Neurons with no antecedents (resp. no successors) are called input (resp. output) neurons, and their set is denoted* $N_{in}$ *(resp. $N_{out}$). Input and output dimensions are respectively* $d_{in} := |N_{in}|$ *and* $d_{out} := |N_{out}|$.

- *A **ReLU neural network architecture** is a tuple* $(G, (\rho_v)_{v \in N \setminus N_{in}})$ *composed of a DAG* $G = (N, E)$ *with attributes* $\rho_v \in \{\operatorname{id}, \operatorname{ReLU}\} \cup \{k\text{-}\mathtt{pool}, k \in \mathbb{N}_{>0}\}$ *for* $v \in N \setminus (N_{out} \cup N_{in})$ *and* $\rho_v = \operatorname{id}$ *for* $v \in N_{out}$. *We will again denote the tuple* $(G, (\rho_v)_{v \in N \setminus N_{in}})$ *by* $G$, *and it will be clear from context whether the results depend only on* $G = (N, E)$ *or also on its attributes. Define* $N_\rho := \{v \in N, \rho_v = \rho\}$ *for an activation* $\rho$, *and* $N_{*\text{-}\mathtt{pool}} := \cup_{k \in \mathbb{N}_{>0}} N_{k\text{-}\mathtt{pool}}$. *A neuron in* $N_{*\text{-}\mathtt{pool}}$ *is called a $*$-max-pooling neuron. For* $v \in N_{*\text{-}\mathtt{pool}}$, *its kernel size is defined as being* $|\operatorname{ant}(v)|$.

- ***Parameters** associated with this architecture are vectors[3]* $\boldsymbol{\theta} \in \mathbb{R}^G := \mathbb{R}^{E \cup N \setminus N_{in}}$. *We call bias* $b_v := \boldsymbol{\theta}_v$ *the coordinate associated with a neuron* $v$ *(input neurons have no bias), and denote* $\boldsymbol{\theta}^{u \to v}$ *the weight associated with an edge* $u \to v \in E$. *We will often denote* $\boldsymbol{\theta}^{\to v} := (\boldsymbol{\theta}^{u \to v})_{u \in \operatorname{ant}(v)}$ *and* $\boldsymbol{\theta}^{v \to} := (\boldsymbol{\theta}^{u \to v})_{u \in \operatorname{suc}(v)}$.

- *The **realization** of a neural network with parameters* $\boldsymbol{\theta} \in \mathbb{R}^G$ *is the function* $R_{\boldsymbol{\theta}}^G : \mathbb{R}^{N_{in}} \to \mathbb{R}^{N_{out}}$ *(simply denoted* $R_{\boldsymbol{\theta}}$ *when* $G$ *is clear from the context) defined for every input* $x \in \mathbb{R}^{N_{in}}$ *as*

$$R_{\boldsymbol{\theta}}(x) := (v(\boldsymbol{\theta}, x))_{v \in N_{out}},$$

---

[3]For an index set $I$, denote $\mathbb{R}^I = \{(\boldsymbol{\theta}_i)_{i \in I}, \boldsymbol{\theta}_i \in \mathbb{R}\}$.

*where we use the same symbol $v$ to denote a neuron $v \in N$ and the associated function $v(\boldsymbol{\theta}, x)$, defined as $v(\boldsymbol{\theta}, x) := x_v$ for an input neuron $v$, and defined by induction otherwise*

$$v(\boldsymbol{\theta}, x) := \begin{cases} \rho_v(b_v + \sum_{u \in \text{ant}(v)} u(\boldsymbol{\theta}, x)\boldsymbol{\theta}^{u \to v}) & \text{if } \rho_v = \text{ReLU} \text{ or } \rho_v = \text{id}, \\ k\text{-pool}\left((b_v + u(\boldsymbol{\theta}, x)\boldsymbol{\theta}^{u \to v})_{u \in \text{ant}(v)}\right) & \text{if } \rho_v = k\text{-pool}. \end{cases} \quad (1)$$

Such a model indeed encompasses modern networks via the following implementations:

• *Max-pooling:* set $\rho_v = k\text{-pool}$ for $k = 1$, $b_v = 0$ and $\boldsymbol{\theta}^{u \to v} = 1$ for every $u \in \text{ant}(v)$.

• *Average-pooling:* set $\rho_v = \text{id}$, $b_v = 0$ and $\boldsymbol{\theta}^{u \to v} = 1/|\text{ant}(v)|$ for every $u \in \text{ant}(v)$.

• *GroupSort/Top-k operator:* use the DAG structure and $*$-max-pooling neurons (Anil et al., 2019; Sander et al., 2023).

• *Batch normalization:* set $\rho_v = \text{id}$ and weights accordingly. Batch normalization layers only differ from standard affine layers by the way their parameters are updated during training.

• *Skip connections:* via the DAG structure, the outputs of any past layers can be added to the pre-activation of any neuron by adding connections from these layers to the given neuron.

• *Convolutional layers:* consider them as (doubly) circulant/Toeplitz fully connected layers.

## 2.2 PATH-LIFTING, PATH-ACTIVATIONS AND PATH-NORMS

Given a general DAG ReLU network $G$ as in Definition 2.2, it is possible to define a set of paths $\mathcal{P}^G$, a path-lifting $\Phi^G$ and path-activations $\boldsymbol{A}^G$, see Definition A.3 in the supplementary. The $L^q$ path-norm is simply $\|\Phi^G(\boldsymbol{\theta})\|_q$. Some bounds we later establish also exploit so-called mixed path-norms $\|\Phi^G(\boldsymbol{\theta})\|_{q,r}$ ($= \|\Phi^G(\boldsymbol{\theta})\|_q$ when $q = r$) introduced in Definition A.4. Superscript $G$ is omitted when obvious from the context. The interest is that (mixed) path-norms, which are easy to compute, can be interpreted as Lipschitz bounds of the network, smaller than another Lipschitz bound based on products of operator norms. We give here a high-level overview of the definitions and properties, and refer to Appendix A for formal definitions, proofs and technical details. We highlight that the definitions and properties coincide with previously known ones in classical simpler settings.

**Path-lifting and path-activations: fundamental properties.** The path-lifting $\Phi$ and the path-activations $\boldsymbol{A}$ are defined to ensure the next fundamental properties: 1) for each parameter $\boldsymbol{\theta}$, the path-lifting $\Phi(\boldsymbol{\theta}) \in \mathbb{R}^{\mathcal{P}}$ is independent of the inputs $x$, and polynomial in the parameters $\boldsymbol{\theta}$ in a way that it is invariant under all neuron-wise rescaling symmetries[4] networks.; 2) $\boldsymbol{A}(\boldsymbol{\theta}, x) \in \mathbb{R}^{\mathcal{P} \times (d_{\text{in}}+1)}$ takes a finite number of values and is piece-wise constant as a function of $(\boldsymbol{\theta}, x)$; and 3) denoting $\Phi^{\to v}$ and $\boldsymbol{A}^{\to v}$ to be the same objects but associated with the graph deduced from $G$ by keeping only the largest subgraph of $G$ with the same inputs as $G$ and with single output $v$, the output of every neuron $v$ can be written as

$$v(\boldsymbol{\theta}, x) = \left\langle \Phi^{\to v}(\boldsymbol{\theta}), \boldsymbol{A}^{\to v}(\boldsymbol{\theta}, x) \begin{pmatrix} x \\ 1 \end{pmatrix} \right\rangle. \quad (2)$$

Compared to previous definitions given in simpler models (no $k$-max-pooling even for a single given $k$, no skip connections, no biases, one-dimensional output and/or layered network) (Kawaguchi et al., 2017; Bona-Pellissier et al., 2022; Stock & Gribonval, 2023), the main novelty is essentially to properly define the path-activations $A(\boldsymbol{\theta}, x)$ in the presence of $*$-max-pooling neurons: when going through a $k$-max-pooling neuron, a path stays active only if the previous neuron of the path is the first in lexicographic order to be the $k$-th largest input of this pooling neuron.

**Path-norms are easy to compute.** It is mentioned in Dziugaite et al. (2020, Appendix C.6.5) and Jiang et al. (2020, Equation (44)) (without proof) that for layered fully-connected ReLU networks without biases, the $L^2$ path-norm can be computed in a single forward pass with the formula: $\|\Phi(\boldsymbol{\theta})\|_2^2 = \|R_{|\boldsymbol{\theta}|^2}(\mathbf{1})\|_1$, where $|\alpha|^q$ is the vector $\alpha$ with $x \mapsto |x|^q$ applied coordinate-wise and where $\mathbf{1}$ is the constant input equal to one. This can be proved in a straightforward way, using Equation (2), and extended to mixed path-norm: $\|\Phi(\boldsymbol{\theta})\|_{q,r} = \| |R_{|\boldsymbol{\theta}|^q}(\mathbf{1})|^{1/q} \|_r$. However, Appendix A

---

[4]Because of positive-homogeneity of the considered activations functions, the realized function is preserved (Stock & Gribonval, 2023) when the incoming weights and the bias of a neuron are multiplied by $\lambda > 0$, while its outgoing weights are divided by $\lambda$. Path-norms inherit such symmetries and are further invariant to certain neuron permutations, typically within each layer in the case of layered fully-connected

shows that this formula is *false* as soon as there is at least one $*$-max-pooling neuron, and that easy computation remains possible by first replacing the activation function of $*$-max-pooling neurons with the identity before doing the forward pass. Average-pooling neurons also need to be explicitly modeled as described after Definition 2.2, to apply $x \mapsto |x|^q$ to their weights.

**The mixed $L^{1,r}$ path-norm is a Lipschitz bound.** Equation (2) is fundamental to understand the role of path-norms. It shows that $\Phi$ contains information about the slopes of the function realized by the network on each region where $A$ is constant. A formal result that goes in this direction is the Lipschitz bound $\|R_{\boldsymbol{\theta}}(x) - R_{\boldsymbol{\theta}}(x')\|_r \leqslant \|\Phi(\boldsymbol{\theta})\|_{1,r}\|x - x'\|_\infty$, previously only known in the specific case $r = 1$, for *scalar-valued* layered fully-connected ReLU networks (Neyshabur, 2017, before Section 3.4) (Furusho, 2020, Theorem 5), which does not require mixed path-norms since $\|\Phi(\boldsymbol{\theta})\|_{q,r} = \|\Phi(\boldsymbol{\theta})\|_q$ for *scalar-valued* networks. This is generalized to the more general case of Definition 2.2 in Lemma A.2. This allows for leveraging generic generalization bounds that apply to the set of all $L$-Lipschitz functions $f : [0,1]^{d_{in}} \to [0,1]$, however these bounds suffer from the curse of dimensionality (von Luxburg & Bousquet, 2004), unlike the bounds established in Section 3.

**Path-norms tightly lower bound products of operator norms.** For *layered fully-connected* ReLU networks (LFCN) with no bias, *i.e.*, $R_{\boldsymbol{\theta}}(x) = M_D \operatorname{ReLU}(M_{D-1} \dots \operatorname{ReLU}(M_1 x))$, with matrices $M_1, \dots, M_D$, another known Lipschitz bound is (Neyshabur et al., 2018; Gonon et al., 2023) $\prod_{d=1}^{D} \|M_d\|_{q,\infty}$ with $q = 1$, where $\|M\|_{q,\infty}$ is the maximum $L^q$ norm of a row of matrix $M$. This product is also used to derive generalization guarantees (Neyshabur et al., 2015; Bartlett et al., 2017; Golowich et al., 2018). So which one of path-norm and products of operator norms should be used? There are at least three reasons to consider the path-norm. **First**, it holds $\|\Phi(\boldsymbol{\theta})\|_{q,\infty} \leqslant \prod_{d=1}^{D} \|M_d\|_{q,\infty}$ (Theorem C.1), with equality if the parameters are properly rescaled. This is known for *scalar-valued* LFCNs without biases (Neyshabur et al., 2015, Theorem 5). Appendix A generalizes it to DAGs as in Definition 2.2. The difficulty is to define the equivalent of the product of operator norms with an arbitrary DAG and in the presence of biases. Apart from that, the proof is essentially the same as in Neyshabur et al. (2015), with a similar rescaling that leads to equality of both measures, see Algorithm 1. **Second**, there are cases where the product of operator norms is arbitrarily large while the path-norm is zero (see Figure 2 in Appendix C). Thus, it is *not desirable* to have a generalization bound that depends on this product of operator norms since, compared to the path-norm, it fails to capture the complexity of the network end-to-end by *decoupling the layers of neurons* one from each other. **Third**, it has been empirically observed that products of operator norms *negatively* correlate with the empirical generalization error while the path-norm *positively* correlates (Jiang et al., 2020, Table 2)(Dziugaite et al., 2020, Figure 1).

## 3 GENERALIZATION BOUND

The generalization bound of this section is based on path-norm for general DAG ReLU network. It encompasses modern networks, recovers or beats the sharpest known bounds of this type, and applies to the cross-entropy loss. The top-one accuracy loss is not directly covered, but can be controlled via a bound on the margin-loss, as detailed at the end of this section.

### 3.1 MAIN RESULT

To state the main result let us recall the definition of the generalization error.

**Definition 3.1.** *(Generalization error) Consider an architecture $G$ (Definition 2.2) with input and output dimensions $d_{in}$ and $d_{out}$, and a so-called loss function $\ell : \mathbb{R}^{d_{out}} \times \mathbb{R}^{d_{out}} \to \mathbb{R}$. The $\ell$-generalization error of parameters $\boldsymbol{\theta}$ on a collection $Z$ of $n \in \mathbb{N}_{>0}$ pairs of input/output $z_i = (x_i, y_i) \in \mathbb{R}^{d_{in}} \times \mathbb{R}^{d_{out}}$ and with respect to a probability measure $\mu$ on $\mathbb{R}^{d_{in}} \times \mathbb{R}^{d_{out}}$ is:*

$$\ell\text{-generalization error}(\boldsymbol{\theta}, Z, \mu) := \underbrace{\mathbb{E}_{(\mathbf{X}_0, \mathbf{Y}_0) \sim \mu} \left( \ell \left( R_{\boldsymbol{\theta}}(\mathbf{X}_0), \mathbf{Y}_0 \right) \right)}_{\text{test error}} - \underbrace{\frac{1}{n} \sum_{i=1}^{n} \ell \left( R_{\boldsymbol{\theta}}(x_i), y_i \right)}_{\text{training error when trained on } Z}.$$

While all the other results hold for arbitrary biases, the next theorem holds **only if the $*$-max-pooling neurons have null biases**. For simplicity, we state the result when *all* the biases are null. We then explain how to extend the result when neurons $u \notin N_{*\text{-pool}}$ may have $b_u \neq 0$.

**Theorem 3.1.** *Consider a ReLU neural network architecture $G$ (Definition 2.2) with null biases, with input/output dimensions $d_{in}/d_{out}$. Denote $D$ its depth (the maximal length of a path from an*

input to an output), $P := |\{k \in \mathbb{N}_{>0}, \exists u \in N_{k\text{-pool}}\}|$ the number of distinct types of $*$-max-pooling neurons in $G$, $K := \max_{u \in N_{*\text{-pool}}} |\operatorname{ant}(u)|$ its maximal kernel size ($K := 1$ if $P = 0$). Assume $N_{in} \cap N_{out} = \emptyset$. Consider a so-called loss function $\ell : \mathbb{R}^{d_{out}} \times \mathbb{R}^{d_{out}} \to \mathbb{R}$ and $L > 0$ such that

$$\ell(\hat{y}_1, y) - \ell(\hat{y}_2, y) \leqslant L\|\hat{y}_1 - \hat{y}_2\|_2, \quad \forall y, \hat{y}_1, \hat{y}_2 \in \operatorname{support}(\mathbf{Y}_1). \tag{3}$$

Consider $n + 1$ iid random variables $\mathbf{Z}_i = (\mathbf{X}_i, \mathbf{Y}_i) \sim \mu$, $0 \leqslant i \leqslant n$, with $\mu$ a probability measure on input/output pairs in $\mathbb{R}^{d_{in}} \times \mathbb{R}^{d_{out}}$, and denote $\mathbf{Z} = (\mathbf{Z}_i)_{i=1,\ldots,n}$. Define $\sigma := \left(\mathbb{E}_{\mathbf{X}} \max(n, \sum_{i=1}^n \|\mathbf{X}_i\|_\infty^2)\right)^{1/2}$.

For any set of parameters $\mathbf{\Theta}$ and any estimator $\hat{\boldsymbol{\theta}} : \mathbf{Z} \mapsto \hat{\boldsymbol{\theta}}(\mathbf{Z}) \in \mathbf{\Theta}$ it holds[5][6]:

$$\mathbb{E}_{\mathbf{Z}}\ell\text{-generalization error}(\hat{\boldsymbol{\theta}}(\mathbf{Z}), \mathbf{Z}, \mu) \leqslant \frac{4\sigma}{n} LC \sup_{\boldsymbol{\theta} \in \mathbf{\Theta}} \|\Phi(\boldsymbol{\theta})\|_1$$

with (log being the natural logarithm)

$$C := \left( D \log((3 + 2P)K) + \log\left( \frac{3 + 2P}{1 + P} d_{in} d_{out} \right) \right)^{1/2}.$$

Any neural network with nonzero biases can be transformed into an equivalent network (with same path-norms and same $R_{\boldsymbol{\theta}}$) with null biases for every $v \notin N_{*\text{-pool}}$: add an input neuron $v_{\text{bias}}$ with constant input equal to one, add edges between this input neuron and every neuron $v \notin N_{*\text{-pool}}$ with parameter $\boldsymbol{\theta}^{v_{\text{bias}} \to v} := b_v$, and set $b_v = 0$. Thus the same result holds with $b_v \neq 0$ for $v \notin N_{*\text{-pool}}$, with $d_{in}$ replaced by $d_{in} + 1$ in the definition of $C$, and with an additional constant input coordinate equal to one in the definition of $\sigma$ so that $\sigma = \left(\mathbb{E}_{\mathbf{X}} \max\left(n, \max_{u=1,\ldots,d_{in}} \sum_{i=1}^n (\mathbf{X}_i)_u^2\right)\right)^{1/2} \geqslant \sqrt{n}$. The proof in Appendix F is directly given for networks with nonzero biases (except $*$-max-pooling neurons), using this construction.

Theorem 3.1 applies to the cross-entropy loss with $L = \sqrt{2}$ (see Appendix G) if the labels $y$ are one-hot encodings[7]. A final softmax layer can be incorporated for free to the model by putting it in the loss. This does not change the bound since it is 1-Lipschitz with respect to the $L^2$-norm (this is a simple consequence of the computations made in Appendix G).

On ImageNet, it holds $1/\sqrt{n} \leqslant \sigma/n \leqslant 2.6/\sqrt{n}$ (Section 4). This yields a bounds that decays in $\mathcal{O}(n^{-1/2})$ which is better than the generic $\mathcal{O}(n^{-1/d_{in}})$ generalization bound for Lipschitz functions (von Luxburg & Bousquet, 2004, Thm. 18) that suffer from the curse of dimensionality. Besides its wider range of applicability, this bounds also recovers or beats the sharpest known ones based on path-norm, see Table 1.

Finally, note that Theorem 3.1 can be tightened: the same bound holds without counting the identity neurons when computing $D$. Indeed, for any neural network and parameters $\boldsymbol{\theta}$, it is possible to remove all the neurons $v \in N_{\text{id}}$ by adding a new edge $u \to w$ for any $u \in \operatorname{ant}(v), w \in \operatorname{suc}(v)$ with new parameter $\boldsymbol{\theta}^{u \to v} \boldsymbol{\theta}^{v \to w}$ (if this edge already exists, just add the latter to its already existing parameter). This still realizes the same function, with the same path-norm, but with less neurons, and thus with $D$ possibly decreased. The proof technique would also yield a tighter bound but not by much: the occurences of 3 in $C$ would be replaced by 2.

*Sketch of proof for Theorem 3.1.* The proof idea is explained below. Details are in Appendix F.

**Already known ingredients.** Classical arguments (Shalev-Shwartz & Ben-David, 2014, Theorem 26.3)(Maurer, 2016), that are valid for any model, bound the expected generalization error by the Rademacher complexity of the model. It remains to bound the latter, and this gets specific to neural

---

[5]The definition of the generalization error (Definition 3.1) has been given for deterministic $\boldsymbol{\theta}$ to keep things simple. The careful reader will have noted that the term corresponding to the test error in Definition 3.1 has to be modified when $\boldsymbol{\theta}$ is a function of $\mathbf{Z}$. Indeed, the expectation has to be taken on an iid copy $\mathbf{Z}_0$ conditionally on each $\mathbf{Z}_i$, $i = 1, \ldots, n$: the test error should be defined as $\mathbb{E}_{\mathbf{Z}_0 \sim \mu}\left(\ell(R_{\hat{\boldsymbol{\theta}}(\mathbf{Z})}(\mathbf{X}_0), \mathbf{Y}_0)|\mathbf{Z}\right)$. This correct definition will be used in the proof, but it has no importance here to understand the statement of the theorem.

[6]Classical concentration results (Boucheron et al., 2013) can be used to deduce a bound that holds with high probability under additional mild assumptions on the loss.

[7]A vector $y$ is a one-hot encoding of a class $c$ if $y = (\mathbb{1}_{c'=c})_{c' \in \{1,\ldots,d_{out}\}}$.

networks. In the case of a layered fully-connected ReLU neural network with no biases and scalar output (and no skip connections nor $k$-max-pooling even for a single given $k$), Golowich et al. (2018) proved that it is possible to bound this Rademacher complexity with no exponential factor in the depth, by peeling, one by one, each layer off the Rademacher complexity. To get more specific, for a class of functions $\boldsymbol{F}$ and a function $\Psi : \mathbb{R} \to \mathbb{R}$, denote $\mathrm{Rad} \circ \Psi(\boldsymbol{F}) = \mathbb{E}_{\boldsymbol{\varepsilon}} \Psi(\sup_{f \in \boldsymbol{F}} \sum_{i=1}^{n} \varepsilon_i f(x_i))$ the Rademacher complexity of $\boldsymbol{F}$ associated with $n$ inputs $x_i$ and $\Psi$, where the $\varepsilon_i$ are iid Rademacher variables ($\varepsilon_i = 1$ or $-1$ with equal probability). The goal for a generalization bound is to bound this in the case $\Psi(x) = \mathrm{id}(x) = x$. In the specific case where $\boldsymbol{F}_D$ is the class of functions that correspond to layered fully-connected ReLU networks with depth $D$, assuming that some operator norm of each layer $d$ is bounded by $r_d$, Golowich et al. (2018) basically guarantees $\mathrm{Rad} \circ \Psi_{\lambda}(\boldsymbol{F}_D) \leqslant 2 \, \mathrm{Rad} \circ \Psi_{\lambda r_D}(\boldsymbol{F}_{D-1})$ for every $\lambda > 0$, where $\Psi_{\lambda}(x) = \exp(\lambda x)$. Compared to previous works of Golowich et al. (2018) that were directly working with $\Psi = \mathrm{id}$ instead of $\Psi_{\lambda}$, the important point is that working with $\Psi_{\lambda}$ gets the 2 outside of the exponential. Iterating over the depth $D$, optimizing over $\lambda$, and taking a logarithm at the end yields (by Jensen's inequality) a bound on $\mathrm{Rad} \circ \mathrm{id}(\boldsymbol{F}_D)$ with a dependence on $D$ that grows as $\sqrt{D \log(2)}$ instead of $2^D$ for previous approaches.

**Novelties for general DAG ReLU networks.** Compared to the setup of Golowich et al. (2018), there are at least three difficulties to do something similar here. First, *the neurons are not organized in layers* as the model can be an arbitrary DAG. So what should be peeled off one by one? Second, *the neurons are not necessarily ReLU neurons* as their activation function might be the identity (average-pooling) or $*$-max-pooling. Finally, Golowich et al. (2018) has a constraint on the weights of each layer, which makes it possible to pop out the constant $r_d$ when layer $d$ is peeled off. *Here, the only constraint is global*, since it constrains the paths of the network through $\|\Phi(\boldsymbol{\theta})\|_1 \leqslant r$. In particular, due to rescalings, the weights of a given neuron could be arbitrarily large or small under this constraint.

The first difficulty is primarily addressed using a new peeling lemma (Appendix E) that exploits a new contraction lemma (Appendix D). The second difficulty is resolved by splitting the ReLU, $k$-max-pooling and identity neurons in different groups before each peeling step. This makes the $\log(2)$ term in Golowich et al. (2018) a $\log(3 + 2P)$ term here ($P$ being the number of different $k$'s for which $k$-max-pooling neurons are considered). Finally, the third obstacle is overcome by *rescaling the parameters* to normalize the vector of incoming weights of each neuron. This type of rescaling has also been used in Neyshabur et al. (2015); Barron & Klusowski (2019).  □

**Remark 3.1** (Improved bound with assumptions on $*$-max-pooling neurons). *In the specific case where there is a single type of $k$-max-pooling neurons ($P = 1$), assuming that these $k$-max-pooling neurons are grouped in layers, and that there are no skip connections going over these $k$-max-pooling layers (satisfied by ResNets, not satisfied by U-nets), a sharpened peeling argument can yield the same bound but with $C$ replaced by $C_{sharpened} = (D \log(3) + M \log(K) + \log((d_{in} + 1)d_{out}))^{1/2}$ with $M$ being the number of $k$-max-pooling layers (cf. Appendix E). The details are tedious so we only mention this result without proof. This basically improves $\sqrt{D \log(5K)}$ into $\sqrt{D \log(3) + M \log(K)}$. For Resnet152, $K = 9$, $D = 152$ and $M = 1$, $\sqrt{D \log(5K)} \simeq 24$ while $\sqrt{D \log(3) + M \log(K)} \simeq 13$.*

### 3.2 How to deal with the top-1 accuracy loss?

Theorem 3.1 *does not apply to the top-1 accuracy loss* as Equation (3) cannot be satisfied for any finite $L > 0$ in general (see Appendix H). It is still possible to bound the expected (test) top-1 accuracy by the so-called *margin loss* achieved at training (Bartlett et al., 2017, Lemma A.4). The margin-loss is a relaxed definition of the top-1 accuracy loss. A corollary of Theorem 3.1 is the next result proved in Appendix I.

**Theorem 3.2** (Bound on the probability of misclassification). *Consider the setting of Theorem 3.1. Assume that the labels are indices $y \in \{1, \ldots, d_{out}\}$. For any $\gamma > 0$, it holds*

$$
\mathbb{P}\left( \arg\max_c R_{\boldsymbol{\theta}}(\mathbf{X}_1)_c \neq \mathbf{Y}_1 \right) \leqslant \frac{1}{n} \sum_{i=1}^{n} \mathbb{1}_{(R_{\hat{\boldsymbol{\theta}}(\mathbf{Z})}(\mathbf{X}_i))_{\mathbf{Y}_i} \leqslant \gamma + \max_{c \neq \mathbf{Y}_i} (R_{\hat{\boldsymbol{\theta}}(\mathbf{Z})}(\mathbf{X}_i))_c}
$$
$$
+ \frac{8\sigma}{n} C \frac{\sup_{\boldsymbol{\theta}} \|\Phi(\boldsymbol{\theta})\|_1}{\gamma}. \tag{4}
$$

Table 2: Numerical evaluations on ResNets and ImageNet1k with 2 significant digits. Multiplying by the Lipschitz constant $L$ of the loss and the path-norm gives the bound in Theorem 3.1. The second line reports the values when the analysis is sharpened for max-pooling neurons, see Remark 3.1.

| ResNet | 18 | 34 | 50 | 101 | 152 |
|---|---|---|---|---|---|
| $\frac{4}{\sqrt{n}}CB =$ | 0.088 | 0.11 | 0.14 | 0.19 | 0.23 |
| $\frac{4}{\sqrt{n}}C_{\text{sharpened}}B =$ | 0.060 | 0.072 | 0.082 | 0.11 | 0.13 |

Table 3: Path-norms of pretrained ResNets available on PyTorch, computed in float32.

| ResNet | 18 | 34 | 50 | 101 | 152 |
|---|---|---|---|---|---|
| $\|\Phi(\boldsymbol{\theta})\|_1$ | $1.3 \times 10^{30}$ | overflow | overflow | overflow | overflow |
| $\|\Phi(\boldsymbol{\theta})\|_2$ | $2.5 \times 10^2$ | $1.1 \times 10^2$ | $2.0 \times 10^8$ | $2.9 \times 10^9$ | $8.9 \times 10^{10}$ |
| $\|\Phi(\boldsymbol{\theta})\|_4$ | $7.2 \times 10^{-6}$ | $4.9 \times 10^{-6}$ | $6.7 \times 10^{-4}$ | $3.0 \times 10^{-4}$ | $1.5 \times 10^{-4}$ |

Note that the result is homogeneous: scaling both the outputs of the model and $\gamma$ by the same scalar leaves the classifier *and the corresponding bound* unchanged.

## 4 EXPERIMENTS

Theorem 3.1 gives the first path-norm generalization bound that can be applied to modern networks (with average/∗-max-pooling, skip connections etc.). This bound is also the sharpest known bound of this type (Table 1). Since this bound is also easy to compute, the goal of this section is to numerically challenge for the first time the sharpest generalization bounds based on path-norm on modern networks. Note also that path-norms tightly lower bound products of operator norms (Appendix C) so that this also challenges the latter.

**When would the bound be informative?** For ResNets trained on ImageNet, the training error associated with cross-entropy is typically between 1 and 2, and the top-1 training error is typically less than 0.30. The same orders of magnitude apply to the empirical generalization error. To ensure that the test error (either for cross-entropy or top-1 accuracy) is of the same order as the training error, *the bound should basically be of order* 1.

For parameters $\boldsymbol{\theta}$ learned from training data, Theorem 3.1 and Theorem 3.2 allow to bound the expected loss in terms of a performance measure (that depends on a free choice of $\gamma > 0$ for the top-1 accuracy) on training data plus a term bounded by $\frac{4\sigma}{n}C \times L \times \|\Phi(\boldsymbol{\theta})\|_1$. The Lipschitz constant $L$ is $\sqrt{2}$ for cross-entropy, and $2/\gamma$ for the top-1 accuracy.

**Evaluation of $\frac{4\sigma}{n}C$ for ResNets on ImageNet.** We further bound $\sigma/n$ by $B/\sqrt{n}$, where $B \simeq 2.6$ is the maximum $L^\infty$-norm of the images of ImageNet normalized for inference. We at most lose a factor $B$ compared to the bound directly involving $\sigma$ since it also holds $\sigma/n \geqslant 1/\sqrt{n}$ by definition of $\sigma$. We train on 99% of ImageNet so that $n = 1268355$. Moreover, recall that $C = (D \log((3 + 2P)K) + \log(\frac{3+2P}{1+P}(d_{\text{in}} + 1)d_{\text{out}}))^{1/2}$. For ResNets, $P = 1$ (as there are only classical max-pooling neurons, corresponding to $k$-max-pooling with $k = 1$), the kernel size is $K = 9$, $d_{\text{in}} = 224 \times 224 \times 3$, $d_{\text{out}} = 1000$, and the depth is $D = 2 + \#$ basic blocks $\times \#$ conv per basic block, with the different values available in Appendix J. The values for $4BC/\sqrt{n}$ are reported in Table 2. Given these results and the values of the Lipschitz constant $L$, on ResNet18, *the bound would be informative only when* $\|\Phi(\boldsymbol{\theta})\|_1 \lesssim 10$ *or* $\|\Phi(\boldsymbol{\theta})\|_1/\gamma \lesssim 10$ respectively for the cross-entropy and the top-1 accuracy.

We now compute the path-norms of trained ResNets, both dense and sparse, using the simple formula proved in Theorem A.1 in appendix.

$L^1$ **path-norm of pretrained ResNets are** 30 **orders of magnitude too large.** Table 3 shows that the $L^1$ path-norm is 30 orders of magnitude too large to make the bound informative for the cross-entropy loss. The choice of $\gamma$ is discussed in Appendix J, where we observe that there is no possible choice that leads to an informative bound for top-1 accuracy in this situation.

**Sparse ResNets can decrease the bounds by 13 orders of magnitude.** We just saw that pretrained ResNets have very large $L^1$ path-norm. Does every network with a good test top-1 accuracy have

such a large $L^1$ path-norm? Since any zero in the parameters $\boldsymbol{\theta}$ leads to many zero coordinates in $\Phi(\boldsymbol{\theta})$, we now investigate whether sparse versions of ResNet18 trained on ImageNet have a smaller path-norm. Sparse networks are obtained with iterative magnitude pruning plus rewinding, with hyperparameters similar to the one in Frankle et al. (2021, Appendix A.3). Results show that the $L^1$ path-norm decreases from $\simeq 10^{30}$ for the dense network to $\simeq 10^{17}$ after 19 pruning iterations, basically losing between a half and one order of magnitude per pruning iteration. Moreover, the test top-1 accuracy is better than with the dense network for the first 11 pruning iterations, and after 19 iterations, the test top-1 accuracy is still way better than what would be obtained by guessing at random, so this is still a non-trivial matter to bound the generalization error for the last iteration. Details are in Appendix J. This shows that there are indeed practically trainable networks with much smaller $L^1$ path-norm that perform well. It remains open whether alternative training techniques, possibly with path-norm regularization, could lead to networks combining good performance and informative generalization bounds.

**Additional observations: increased depth and train size.** In practice, increasing the size of the network (*i.e.* the number of parameters) or the number of training samples can improve generalization. We can, again, assess for the first time whether the bounds based on path-norms follows the same trend for standard modern networks. Table 3 shows that path-norms of pretrained ResNets available on PyTorch roughly increase with depth. This is complementary to Dziugaite et al. (2020, Figure 1) where it is empirically observed on *simple layered fully-connected* models that path-norm has difficulty to correlate positively with the generalization error when the depth grows. For increasing training sizes, we did not observe a clear trend for the $L^1$ path-norm, which seems to mildly evolve with the number of epochs rather than with the train size, see Appendix J for details.

## 5 CONCLUSION

**Contribution.** To the best of our knowledge, this work is the first to introduce path-norm related tools for general DAG ReLU networks (with average/$*$-max-pooling, skip connections), and Theorem 3.1 is the first generalization bound valid for such networks based on path-norm. This bound recovers or beats the sharpest known ones of the same type. Its ease of computation leads to the first experiments on modern networks that assess the promises of such approaches. A gap between theory and practice is observed for a dense version of ResNet18 trained with standard tools: the bound is 30 orders of magnitude too large on ImageNet.

**Possible leads to close the gap between theory and practice.** 1) Without changing the bound of Theorem 3.1, sparsity seems promising to reduce the path-norm by several orders of magnitude without changing the performance of the network. 2) Theorem 3.1 results from the worst situation (that can be met) where all the inputs activate all the paths of the network simultaneously. Bounds involving the expected path-activations could be tighter. The coordinates of $\Phi(\boldsymbol{\theta})$ are elementary bricks that can be summed to get the slopes of $R_{\boldsymbol{\theta}}$ on the different region where $R_{\boldsymbol{\theta}}$ is affine (Arora et al., 2017), $\|\Phi(\boldsymbol{\theta})\|_1$ is the sum of all the bricks in absolue value, resulting in a worst-case uniform bound for all the slopes. Ideally, the bound should rather depend on the expected slopes over the different regions, weighted by the probability of falling into these regions. 3) Weight sharing may leave room for sharpened analysis (Pitas et al., 2019; Galanti et al., 2023). 4) A $k$-max-pooling neuron with kernel size $K$ only activates $1/K$ of the paths, but the bound sums the coordinates of $\Phi$ related to these $K$ paths. This may lead to a bound $K$ times too large in general (or even more in the presence of multiple maxpooling layers). 5) Possible bounds involving the $L^q$ path-norm for $q > 1$ deserve a particular attention, since numerical evaluations show that they are several orders of magnitude below the $L^1$ norm.

**Extensions to other architectures.** Despite its applicability to a wide range of standard modern networks, the generalization bound in Theorem 3.1 does not cover networks with other activations than ReLU, identity, and $*$-max-pooling. The same proof technique could be extended to new activations that: 1) are positively homogeneous, so that the weights can be rescaled without changing the associated function; and 2) satisfy a contraction lemma similar to the one established here for ReLU and max neurons (typically requiring the activation to be Lipschitz). A plausible candidate is Leaky ReLU. For smooth approximations of the ReLU, such as the SiLU (for Efficient Nets) and the Hardswish (for MobileNet-V3), parts of the technical lemmas related to contraction may extend since they are Lipschitz, but these activations are not positively homogeneous.

ACKNOWLEDGMENTS

This work was supported in part by the AllegroAssai ANR-19-CHIA-0009 and NuSCAP ANR-20-CE48-0014 projects of the French Agence Nationale de la Recherche.

The authors thank the Blaise Pascal Center for the computational means. It uses the SIDUS (Quemener & Corvellec, 2013) solution developed by Emmanuel Quemener.

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

## Supplementary material

## A  MODEL'S BASICS

Definition A.3 introduces the path-lifting and the path-activations associated with the general model described in Definition 2.2. We choose to call it path-lifting as in Bona-Pellissier et al. (2022) (and not an embedding as initially named in Stock & Gribonval (2023), since it is not injective). Formally, a lifting in the sense of category theory would factorize the mapping $\boldsymbol{\theta} \mapsto R_{\boldsymbol{\theta}}$, which is not the case of $\boldsymbol{\theta} \mapsto \Phi(\boldsymbol{\theta})$, but is the case with the extended mapping $\boldsymbol{\theta} \to (\mathrm{sgn}(\boldsymbol{\theta}), \Phi(\boldsymbol{\theta}))$. We chose the name lifting for $\Phi(\boldsymbol{\theta})$ for the sake of brevity.

**Definition A.1** (Paths and depth in a DAG). *Consider a DAG $G = (N, E)$ as in Definition 2.2. A path of $G$ is any sequence of neurons $v_0, \dots, v_d$ such that each $v_i \to v_{i+1}$ is an edge in $G$. Such a path is denoted $p = v_0 \to \dots \to v_d$. This includes paths reduced to a single $v \in N$, denoted $p = v$. The length of a path is $\mathtt{length}(p) = d$ (the number of edges). We will denote $p_\ell := v_\ell$ the $\ell$-th neuron for a general $\ell \in \{0, \dots, \mathtt{length}(p)\}$ and use the shorthand $p_{\mathrm{end}} = v_{\mathtt{length}(p)}$ for the last neuron. The depth of the graph $G$ is the maximum length over all of its paths. If $v_{d+1} \in \mathrm{suc}(p_{\mathrm{end}})$ then $p \to v_{d+1}$ denotes the path $v_0 \to \dots \to v_d \to v_{d+1}$. We denote by $\mathcal{P}^G$ (or simply $\mathcal{P}$) the set of paths ending at an output neuron of $G$.*

**Definition A.2** (Sub-graph ending at a given neuron). *Given a neuron $v$ of a DAG $G$, we denote $G^{\to v}$ the graph deduced from $G$ by keeping only the largest subgraph with the same inputs as $G$ and with $v$ as a single output: every neuron $u$ with no path to reach $v$ through the edges of $G$ is removed, as well as all its incoming and outcoming edges. We will use the shorthand $\mathcal{P}^{\to v} := \mathcal{P}^{G^{\to v}}$ to denote the set of paths in $G$ ending at $v$.*

**Definition A.3** (Path-lifting and path-activations). *Consider a ReLU neural network architecture $G$ as in Definition 2.2 and parameters $\boldsymbol{\theta} \in \mathbb{R}^G$ associated with $G$. For $p \in \mathcal{P}$, define*

$$\Phi_p(\boldsymbol{\theta}) := \begin{cases} \displaystyle\prod_{\ell=1}^{\mathtt{length}(p)} \boldsymbol{\theta}^{v_{\ell-1} \to v_\ell} & \text{if } p_0 \in N_{in}, \\ b_{p_0} \displaystyle\prod_{\ell=1}^{\mathtt{length}(p)} \boldsymbol{\theta}^{v_{\ell-1} \to v_\ell} & \text{otherwise}, \end{cases}$$

*where an empty product is equal to 1 by convention. The path-lifting $\Phi^G(\boldsymbol{\theta})$ of $\boldsymbol{\theta}$ is*

$$\Phi^G(\boldsymbol{\theta}) := (\Phi_p(\boldsymbol{\theta}))_{p \in \mathcal{P}^G}.$$

*This is often denoted $\Phi$ when the graph $G$ is clear from the context. We will use the shorthand $\Phi^{\to v} := \Phi^{G^{\to v}}$ to denote the path-lifting associated with $G^{\to v}$ (Definition A.2).*

*Consider an input $x$ of $G$. The activation of an edge $u \to v$ on $(\boldsymbol{\theta}, x)$ is defined to be $a_{u \to v}(\boldsymbol{\theta}, x) := 1$ when $v$ is an identity neuron; $a_{u \to v}(\boldsymbol{\theta}, x) := \mathbb{1}_{v(\boldsymbol{\theta}, x) > 0}$ when $v$ is a ReLU neuron; and when $v$ is a $k$-max-pooling neuron, define $a_{u \to v}(\boldsymbol{\theta}, x) := 1$ if the neuron $u$ is the first in $\mathrm{ant}(v)$ in lexicographic order to satisfy $u(\boldsymbol{\theta}, x) := k\text{-pool}\left((w(\boldsymbol{\theta}, x))_{w \in \mathrm{ant}(v)}\right)$ and $a_{u \to v}(\boldsymbol{\theta}, x) := 0$ otherwise. The activation of a neuron $v$ on $(\boldsymbol{\theta}, x)$ is defined to be $a_v(\boldsymbol{\theta}, x) := 1$ if $v$ is an input neuron, an identity neuron, or a $k$-max-pooling neuron, and $a_v(\boldsymbol{\theta}, x) := \mathbb{1}_{v(\boldsymbol{\theta}, x) > 0}$ if $v$ is a ReLU neuron. We then define the activation of a path $p \in \mathcal{P}$ with respect to input $x$ and parameters $\boldsymbol{\theta}$ as: $a_p(\boldsymbol{\theta}, x) := a_{p_0}(\boldsymbol{\theta}, x) \prod_{\ell=1}^{\mathtt{length}(p)} a_{v_{\ell-1} \to v_\ell}(\boldsymbol{\theta}, x)$ (with an empty product set to one by convention). Consider a new symbol $v_{bias}$ that is not used for denoting neurons. The path-activations matrix $\boldsymbol{A}(\boldsymbol{\theta}, x)$ is defined as the matrix in $\mathbb{R}^{\mathcal{P} \times (N_{in} \cup \{v_{bias}\})}$ such that for any path $p \in \mathcal{P}$ and neuron $u \in N_{in} \cup \{v_{bias}\}$*

$$(\boldsymbol{A}(\boldsymbol{\theta}, x))_{p,u} := \begin{cases} a_p(\boldsymbol{\theta}, x) \mathbb{1}_{p_0 = u} & \text{if } u \in N_{in}, \\ a_p(\boldsymbol{\theta}, x) & \text{otherwise when } u = v_{bias}. \end{cases}$$

The next lemma shows how the path-lifting and the path-activations offer an equivalent way to define the model of Definition 2.2.

**Lemma A.1.** *Consider a ReLU network as in Definition 2.2. For every neuron $v$, every input $x$ and every parameters $\boldsymbol{\theta}$:*

$$v(\boldsymbol{\theta}, x) = \left\langle \Phi^{\to v}(\boldsymbol{\theta}), \boldsymbol{A}^{\to v}(\boldsymbol{\theta}, x) \begin{pmatrix} x \\ 1 \end{pmatrix} \right\rangle. \tag{5}$$

*Proof of Lemma A.1.* For any neuron $v$, recall that $\mathcal{P}^{\to v}$ is the set of paths ending at neuron $v$ (Definition A.2). We want to prove

$$v(\boldsymbol{\theta}, x) = \left\langle \Phi^{\to v}(\boldsymbol{\theta}), \boldsymbol{A}^{\to v}(\boldsymbol{\theta}, x) \begin{pmatrix} x \\ 1 \end{pmatrix} \right\rangle$$

$$= \sum_{p \in \mathcal{P}^{\to v}} \Phi_p(\boldsymbol{\theta}) a_p(\boldsymbol{\theta}, x) x_{p_0}. \tag{6}$$

where we recall that $p_0$ denotes the first neuron of a path $p$ (Definition A.1). For paths starting at an input neuron, $x_{p_0}$ is simply the corresponding coordinate of $x$. For other paths, we use the convention $x_u := 1$ for any neuron $u \in N \setminus N_{\text{in}}$.

The proof of Equation (6) goes by induction on a topological sorting Cormen et al. (2009) of the neurons. We start with input neurons since by Definition 2.2, these are the ones without antecedents so they are the first to appear in a topological sorting. Consider an input neuron $v$. The only path in $\mathcal{P}^{\to v}$ is $p = v$. By Definition A.3, it holds $\Phi_p(\boldsymbol{\theta}) = 1$ (empty product) and $a_p(\boldsymbol{\theta}, x) = a_v(\boldsymbol{\theta}, x) = 1$. Moreover, we have $v(\boldsymbol{\theta}, x) = x_v$ (Definition 2.2). This proves Equation (6) for input neurons.

Now, consider $v \notin N_{\text{in}}$ and assume that Equation (6) holds true for every neuron $u \in \text{ant}(v)$. We first prove by cases that

$$v(\boldsymbol{\theta}, x) = a_v(\boldsymbol{\theta}, x) b_v + \sum_{u \in \text{ant}(v)} u(\boldsymbol{\theta}, x) a_{u \to v}(\boldsymbol{\theta}, x) \boldsymbol{\theta}^{u \to v}. \tag{7}$$

**Case of an identity neuron.** If $v$ is an identity neuron, then by Definition 2.2

$$v(\boldsymbol{\theta}, x) = b_v + \sum_{u \in \text{ant}(v)} u(\boldsymbol{\theta}, x) \boldsymbol{\theta}^{u \to v}.$$

Moreover, by Definition A.3 we have $a_v(\boldsymbol{\theta}, x) = 1$ and $a_{u \to v}(\boldsymbol{\theta}, x) = 1$, since $v$ is an identity neuron, so that the previous equality can also be written as

$$v(\boldsymbol{\theta}, x) = a_v(\boldsymbol{\theta}, x) b_v + \sum_{u \in \text{ant}(v)} u(\boldsymbol{\theta}, x) a_{u \to v}(\boldsymbol{\theta}, x) \boldsymbol{\theta}^{u \to v}.$$

This shows Equation (7) in this case.

**Case of a ReLU neuron.** If $v$ is a ReLU neuron then similarly

$$v(\boldsymbol{\theta}, x) = \text{ReLU}\left( b_v + \sum_{u \in \text{ant}(v)} u(\boldsymbol{\theta}, x) \boldsymbol{\theta}^{u \to v} \right) = \mathbb{1}_{v(\boldsymbol{\theta}, x) > 0} \left( b_v + \sum_{u \in \text{ant}(v)} u(\boldsymbol{\theta}, x) \boldsymbol{\theta}^{u \to v} \right)$$

$$= \underbrace{\mathbb{1}_{v(\boldsymbol{\theta}, x) > 0}}_{=a_v(\boldsymbol{\theta}, x)} b_v + \sum_{u \in \text{ant}(v)} u(\boldsymbol{\theta}, x) \underbrace{\mathbb{1}_{v(\boldsymbol{\theta}, x) > 0}}_{=a_{u \to v}(\boldsymbol{\theta}, x)} \boldsymbol{\theta}^{u \to v}$$

This shows again Equation (7).

**Case of a $k$-max-pooling neuron.** When $v$ is a $k$-max-pooling neuron, it holds:

$$v(\boldsymbol{\theta}, x) = k\text{-pool}\left( (b_v + u(\boldsymbol{\theta}, x) \boldsymbol{\theta}^{u \to v})_{u \in \text{ant}(v)} \right)$$

$$= b_v + \sum_{u \in \text{ant}(v)} \underbrace{\mathbb{1}_{u \text{ is the first to realize this } k\text{-pool}}}_{=a_{u \to v}(\boldsymbol{\theta}, x) \text{ (Definition A.3)}} u(\boldsymbol{\theta}, x) \boldsymbol{\theta}^{u \to v} \qquad \text{(Definition 2.1)}$$

$$= \underbrace{a_v(\boldsymbol{\theta}, x)}_{=1 \text{ (Definition A.3)}} b_v + \sum_{u \in \text{ant}(v)} u(\boldsymbol{\theta}, x) a_{u \to v}(\boldsymbol{\theta}, x) \boldsymbol{\theta}^{u \to v}.$$

This finishes proving Equation (7) in every case. Using the induction hypothesis on the antecedents of $v$, Equation (7) implies

$$v(\boldsymbol{\theta}, x) = a_v(\boldsymbol{\theta}, x) b_v + \sum_{u \in \text{ant}(v)} \left( \sum_{p \in \mathcal{P}^{\to u}} \Phi_p(\boldsymbol{\theta}) a_p(\boldsymbol{\theta}, x) x_{p_0} \right) a_{u \to v}(\boldsymbol{\theta}, x) \boldsymbol{\theta}^{u \to v}.$$

We want to prove that this is equal to

$$\sum_{p \in \mathcal{P}^{\to v}} \Phi_p(\boldsymbol{\theta}) a_p(\boldsymbol{\theta}, x) x_{p_0}.$$

A path $\tilde{p} \in \mathcal{P}^{\to v}$ is either the path $\tilde{p} = v$ starting and ending at $v$, or it can be written in a unique way as $\tilde{p} = p \to v$ where $p \in \mathcal{P}^{\to u}$ is a path ending at an antecedent $u$ of $v$. For the simple path $\tilde{p} = v$, it holds by definition (Definition A.3): $a_{\tilde{p}}(\boldsymbol{\theta}, x) = a_v(\boldsymbol{\theta}, x)$, $\Phi_{\tilde{p}}(\boldsymbol{\theta}) = b_v$ and by the convention used in this proof we have $x_{\tilde{p}_0} = x_v = 1$ since $v$ is not an input neuron. This shows:

$$a_v(\boldsymbol{\theta}, x) b_v = \Phi_{\tilde{p}}(\boldsymbol{\theta}) a_{\tilde{p}}(\boldsymbol{\theta}, x) x_{\tilde{p}_0}.$$

When $\tilde{p} = p \to v$ as above, it holds by definition: $x_{p_0} = x_{\tilde{p}_0}$, $\Phi_{\tilde{p}}(\boldsymbol{\theta}) = \Phi_p(\boldsymbol{\theta})\boldsymbol{\theta}^{u \to v}$ and $a_{\tilde{p}}(\boldsymbol{\theta}, x) = a_p(\boldsymbol{\theta}, x) a_{u \to v}(\boldsymbol{\theta}, x)$. It then holds:

$$\Phi_p(\boldsymbol{\theta}) a_p(\boldsymbol{\theta}, x) x_{p_0} a_{u \to v}(\boldsymbol{\theta}, x) \boldsymbol{\theta}^{u \to v} = \Phi_{\tilde{p}}(\boldsymbol{\theta}) a_{\tilde{p}}(\boldsymbol{\theta}, x) x_{\tilde{p}_0}.$$

This concludes the induction and proves the result. $\qquad \square$

A straightforward consequence of Lemma A.1 is that path-norms can be used to bound from above the Lipschitz constant of $x \mapsto R_{\boldsymbol{\theta}}(x)$.

**Definition A.4.** *(Mixed path-norm) For $0 < q, r \leqslant \infty$, define:*

$$\|\Phi(\boldsymbol{\theta})\|_{q,r} := \left\| \left( \|\Phi^{\to v}(\boldsymbol{\theta})\|_q \right)_{v \in N_{out}} \right\|_r. \tag{8}$$

**Lemma A.2.** *Consider $0 < r \leqslant \infty$. For every parameters $\boldsymbol{\theta}$ and inputs $x, x'$:*

$$\|R_{\boldsymbol{\theta}}(x) - R_{\boldsymbol{\theta}}(x')\|_r \leqslant \|\Phi(\boldsymbol{\theta})\|_{1,r} \|x - x'\|_\infty.$$

For $r = 1$, we simply have $\|\Phi(\boldsymbol{\theta})\|_{1,r} = \|\Phi(\boldsymbol{\theta})\|_1$. For this *specific value of $r$*, and in the *specific case* of layered fully-connected neural networks without biases, the conclusion of Lemma A.2 is already mentioned in Neyshabur (2017, before Section 3.4), and proven in Furusho (2020, Theorem 5). Lemma A.2 extends it to arbitrary DAG ReLU networks, and to arbitrary $r \in (0, \infty]$. This requires the introduction of *mixed* path-norms, which, to the best of our knowledge, have not been considered before in the literature.

*Proof of Lemma A.2.* Consider parameters $\boldsymbol{\theta}$. Consider inputs $x, x'$ with the same path-activations with respect to $\boldsymbol{\theta}$: $\boldsymbol{A}(\boldsymbol{\theta}, x) = \boldsymbol{A}(\boldsymbol{\theta}, x')$. For $0 < r < \infty$:

$$\|R_{\boldsymbol{\theta}}(x) - R_{\boldsymbol{\theta}}(x')\|_r^r \underset{\text{Lemma A.1}}{=} \sum_{v \in N_{\text{out}}} \left| \left\langle \Phi^{\to v}(\boldsymbol{\theta}), \boldsymbol{A}^{\to v}(\boldsymbol{\theta}, x) \begin{pmatrix} x \\ 1 \end{pmatrix} - \boldsymbol{A}^{\to v}(\boldsymbol{\theta}, x') \begin{pmatrix} x' \\ 1 \end{pmatrix} \right\rangle \right|^r$$

$$\underset{\text{Hölder}}{\leqslant} \sum_{v \in N_{\text{out}}} \|\Phi^{\to v}(\boldsymbol{\theta})\|_1^r \left\| \boldsymbol{A}^{\to v}(\boldsymbol{\theta}, x) \begin{pmatrix} x \\ 1 \end{pmatrix} - \boldsymbol{A}^{\to v}(\boldsymbol{\theta}, x') \begin{pmatrix} x' \\ 1 \end{pmatrix} \right\|_\infty^r$$

$$\underset{\boldsymbol{A}(\boldsymbol{\theta}, x) = \boldsymbol{A}(\boldsymbol{\theta}, x')}{\leqslant} \sum_{v \in N_{\text{out}}} \|\Phi^{\to v}(\boldsymbol{\theta})\|_1^r \left\| \boldsymbol{A}^{\to v}(\boldsymbol{\theta}, x) \left( \begin{pmatrix} x \\ 1 \end{pmatrix} - \begin{pmatrix} x' \\ 1 \end{pmatrix} \right) \right\|_\infty^r$$

$$\underset{\|\boldsymbol{A}^{\to v}(\boldsymbol{\theta}, x) y\|_\infty \leqslant \|y\|_\infty}{\leqslant} \sum_{v \in N_{\text{out}}} \|\Phi^{\to v}(\boldsymbol{\theta})\|_1^r \|x - x'\|_\infty^r$$

$$= \|\Phi(\boldsymbol{\theta})\|_{1,r}^r \|x - x'\|_\infty^r.$$

When $0 < r < \infty$, we just proved the claim locally on each region where the path-activations $\boldsymbol{A}(\boldsymbol{\theta}, \cdot)$ are constant. This stays true on the boundary of these regions by continuity. It then expands to the whole domain by triangular inequality because on any segment joining any pair of inputs, there is a finite number of times when the region changes. The proof can be easily adapted to $r = \infty$. $\square$

Another straightforward but important consequence of Lemma A.1 is that mixed path-norms can be computed in a single forward pass, up to replacing $*$-max-pooling neurons with linear ones.

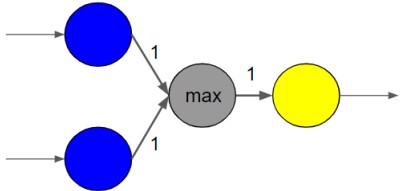

Figure 1: Example of a network where one must replace the max-pooling neuron to compute the path-norm with a single forward pass as in Equation (9).

**Theorem A.1.** *Consider an architecture $G = (N, E, (\rho_v)_{v \in N \setminus N_{in}})$ as in Definition 2.2. Consider the architecture $\tilde{G} := (N, E, (\tilde{\rho}_v)_{v \in N \setminus N_{in}})$ with $\tilde{\rho}_v := \mathrm{id}$ if $v \in N_{\text{*-pool}}$, and $\tilde{\rho}_v := \rho_v$ otherwise. Consider $q \in (0, \infty)$, $r \in (0, \infty]$ and arbitrary parameters $\boldsymbol{\theta} \in \mathbb{R}^G = \mathbb{R}^{\tilde{G}}$. For a vector $\alpha$, denote $|\alpha|^q$ the vector deduced from $\alpha$ by applying $x \mapsto |x|^q$ coordinate-wise. Denote by $\mathbf{1}$ the input full of ones. It holds:*

$$\|\Phi(\boldsymbol{\theta})\|_{q,r} = \| \, |\, \mathrm{R}^{\tilde{G}}_{|\boldsymbol{\theta}|^q}(\mathbf{1})|^{1/q} \, \|_r. \tag{9}$$

*Moreover, the formula is false in general if the $*$-max-pooling neurons have not been replaced with identity ones (i.e. if the forward pass is done on $G$ rather than $\tilde{G}$).*

*Proof of Theorem A.1.* Figure 1 shows that Equation (9) is false if the $*$-max-pooling neurons have not been replaced with identity ones as the forward pass with input $\mathbf{1}$ yields output 1 while the $L^1$ path-norm is 2.

We now establish Equation (9). By continuity in $\boldsymbol{\theta}$ of both sides of Equation (9), it is sufficient to prove it when every coordinate of $\boldsymbol{\theta}$ in $E$ is nonzero. Note that by Definition A.3, the path-lifting $\Phi^G$ and $\Phi^{\tilde{G}}$ associated respectively with $G$ and $\tilde{G}$ are the same since $G$ and $\tilde{G}$ have the same underlying DAG. Denote by $\Phi = \Phi^G = \Phi^{\tilde{G}}$ the common path-lifting, and by $a^{\tilde{G}}$ the path-activations associated with $\tilde{G}$. Denote by $\mathcal{P}^{\to v}$ the set of paths ending at a given neuron $v$ of $\tilde{G}$. According to Lemma A.1, it holds for every output neuron $v$ of $\tilde{G}$ with $x = \mathbf{1}$, using the convention $x_u := 1$ if $u \notin N_{\text{in}}$

$$(R^{\tilde{G}}_{|\boldsymbol{\theta}|^q}(\mathbf{1}))_v = \sum_{p \in \mathcal{P}^{\to v}} \Phi_p(|\boldsymbol{\theta}|^q) a^{\tilde{G}}_p(|\boldsymbol{\theta}|^q, x) x_{p_0} = \sum_{p \in \mathcal{P}^{\to v}} \Phi_p(|\boldsymbol{\theta}|^q) a^{\tilde{G}}_p(|\boldsymbol{\theta}|^q, \mathbf{1}).$$

Since we restricted to $\boldsymbol{\theta}_E$ with nonzero coordinates, $|\boldsymbol{\theta}_E|^q$ has positive coordinates. As $\tilde{G}$ has only identity or ReLU neurons, it follows by a simple induction on the neurons that $u(|\boldsymbol{\theta}|^q, \mathbf{1}) > 0$ for every neuron $u$ of $\tilde{G}$. Thus, every path $p \in \mathcal{P}^{\tilde{G}}$ is active, *i.e.*, $a^{\tilde{G}}_p(|\boldsymbol{\theta}|^q, \mathbf{1}) = 1$. Since by Definition A.3, $\Phi_p(|\boldsymbol{\theta}|^q) = |\Phi_p(\boldsymbol{\theta})|^q$, we obtain:

$$(R^{\tilde{G}}_{|\boldsymbol{\theta}|^q}(\mathbf{1}))_v = \sum_{p \in \mathcal{P}^{\to v}} |\Phi_p(\boldsymbol{\theta})|^q = \|\Phi^{\to v}(\boldsymbol{\theta})\|^q_q.$$

The claim follows by Definition A.4. $\qquad\square$

## B  NORMALIZED PARAMETERS

In the sequel it will be useful to restrict the analysis to *normalized* parameters.

**Definition B.1.** *Consider $0 < q \leqslant \infty$. Parameters $\boldsymbol{\theta}$ are said $q$-normalized if for every $v \in N \setminus (N_{out} \cup N_{in})$:*

$$\|\Phi^{\to v}(\boldsymbol{\theta})\|_q = \| \begin{pmatrix} \boldsymbol{\theta}^{\to v} \\ b_v \end{pmatrix} \|_q \in \{0, 1\},$$

*and if this is 0 then it also holds $\boldsymbol{\theta}^{v\to} = 0$ (**outgoing** edges of $v$).*

Thanks to the rescaling-invariance of ReLU neural network parameterizations, Algorithm 1 allows to rescale *any* parameters $\boldsymbol{\theta}$ into a normalized version $\tilde{\boldsymbol{\theta}}$ such that $R_{\tilde{\boldsymbol{\theta}}} = R_{\boldsymbol{\theta}}$ and $\Phi(\boldsymbol{\theta}) = \Phi(\tilde{\boldsymbol{\theta}})$

(Lemma B.1). The rescaling-invariance are due to the positive-homogeneity of every activation function $\rho \in \{\mathrm{id}, \mathrm{ReLU}, k\text{-}\texttt{pool}\}$: $\rho(x) = \frac{1}{\lambda}\rho(\lambda x)$ for every $\lambda > 0$ and $x \in \mathbb{R}$.

**Algorithm 1** The first line of the algorithm considers a topological sorting of the neurons (Cormen et al., 2009, Section 22.4), *i.e.* an order on the neurons such that if $u \to v$ is an edge then $u$ comes before $v$ in this ordering. Such an order always exists (and it can be computed in linear time). Moreover, note that a classical max-pooling neuron $v$ (corresponding to a $k$-max-pooling neuron with $k = 1$, constant incoming weights all equal to one, and biases set to zero) has not anymore its incoming weights equal to one after normalization, in general. This has no incidence on the validity of the generalization bound on classical max-pooling neurons: rescaling is only used in the proof to reduce to another representation of the parameters that realize the same function and that is more handy to work with.

---

**Algorithm 1** Normalization of parameters for norm $q \in (0, \infty]$

---

1: Consider a topological sorting $v_1, \ldots, v_k$ of the neurons
2: **for** $v = v_1, \ldots, v_k$ **do**
3:     **if** $v \notin N_{\mathrm{in}} \cup N_{\mathrm{out}}$ **then**
4:         $\lambda_v \leftarrow \left\| \begin{pmatrix} \boldsymbol{\theta}^{\to v} \\ b_v \end{pmatrix} \right\|_q$
5:         **if** $\lambda_v = 0$ **then**
6:             $\boldsymbol{\theta}^{v\to} \leftarrow 0$
7:         **else**
8:             $\begin{pmatrix} \boldsymbol{\theta}^{\to v} \\ b_v \end{pmatrix} \leftarrow \frac{1}{\lambda_v} \begin{pmatrix} \boldsymbol{\theta}^{\to v} \\ b_v \end{pmatrix}$           ▷ normalize incoming weights and bias
9:             $\boldsymbol{\theta}^{v\to} \leftarrow \lambda_v \times \boldsymbol{\theta}^{v\to}$           ▷ rescale outgoing weights to preserve the function $R_{\boldsymbol{\theta}}$

---

**Lemma B.1.** *Consider $q \in (0, \infty]$. If $\boldsymbol{\theta}$ is the output of Algorithm 1 for the input $\tilde{\boldsymbol{\theta}}$ then $\boldsymbol{\theta}$ is $q$-normalized (Definition B.1), $R_{\boldsymbol{\theta}} = R_{\tilde{\boldsymbol{\theta}}}$ and $\Phi(\boldsymbol{\theta}) = \Phi(\tilde{\boldsymbol{\theta}})$.*

*Proof.* **Step 1.** We first prove that for every $v \in N \setminus (N_{\mathrm{in}} \cup N_{\mathrm{out}})$:

$$\left\| \begin{pmatrix} \boldsymbol{\theta}^{\to v} \\ b_v \end{pmatrix} \right\|_q \in \{0, 1\}. \tag{10}$$

Consider $v \notin N_{\mathrm{in}} \cup N_{\mathrm{out}}$. Right after $v$ has been processed by the for loop of Algorithm 1 (line 2), it is clear that Equation (10) holds true for $v$. It remains to see that $b_v$ and $\boldsymbol{\theta}^{\to v}$ are not modified until the end of Algorithm 1. A bias can only be modified when this is the turn of the corresponding neuron, so $b_v$ is untouched after the iteration corresponding to $v$. And $\boldsymbol{\theta}^{\to v}$ can only be modified when treating antecedents of $v$, but since antecedents must be before $v$ in any topological sorting, they cannot be processed after $v$. This proves that $\left\| \begin{pmatrix} \boldsymbol{\theta}^{\to v} \\ b_v \end{pmatrix} \right\|_q \in \{0, 1\}$.

**Step 2.** We now prove that if $\left\| \begin{pmatrix} \boldsymbol{\theta}^{\to v} \\ b_v \end{pmatrix} \right\|_q = 0$ then it also holds $\boldsymbol{\theta}^{v\to} = 0$ (*outgoing* edges of $v$).

Indeed, $\left\| \begin{pmatrix} \boldsymbol{\theta}^{\to v} \\ b_v \end{pmatrix} \right\|_q = 0$ implies $\lambda_v = 0$ in Algorithm 1, which implies that $\boldsymbol{\theta}^{v\to} = 0$ right after rescaling (line 6) at the end of the iteration corresponding to $v$. Because the next iterations can only involve multiplication of the coordinates of $\boldsymbol{\theta}^{v\to}$ by scalars, $\boldsymbol{\theta}^{v\to} = 0$ is also satisfied at the end of the algorithm. This proves the claim.

**Step 3.** In order to prove that $\boldsymbol{\theta}$ is $q$-normalized, it only remains to establish that $\|\Phi^{\to v}(\boldsymbol{\theta})\|_q = \left\| \begin{pmatrix} \boldsymbol{\theta}^{\to v} \\ b_v \end{pmatrix} \right\|_q$. We prove this by induction on a topological sorting of the neurons.

A useful fact for the induction is that for every $v \notin N_{\mathrm{in}}$:

$$\Phi^{\to v}(\boldsymbol{\theta}) = \begin{pmatrix} (\Phi^{\to u}(\boldsymbol{\theta})\boldsymbol{\theta}^{u\to v})_{u\in\mathrm{ant}(v)} \\ b_v \end{pmatrix} \tag{11}$$

where we recall that $\Phi^{\to u}(\cdot) = 1$ for input neurons $u$. Equation (11) holds because $\Phi^{\to v}$ is the path-lifting of $G^{\to v}$ (see Definition A.2), and the only paths in $G^{\to v}$ are $p = v$, and the paths going through antecedents of $v$ ($v$ has antecedents since it is not an input neuron).

Let's now start the induction. By definition, the first neuron $v \notin N_{\text{in}}$ in a topological sorting has only input neurons as antecedents. Therefore, $\Phi^{\to u}(\boldsymbol{\theta}) = 1$ for every $u \in \text{ant}(v)$. Using Equation (11), we get

$$\|\Phi^{\to v}(\boldsymbol{\theta})\|_q^q = |b_v|^q + \sum_{u \in \text{ant}(v)} |\boldsymbol{\theta}^{u \to v}|^q = \left\| \begin{pmatrix} \boldsymbol{\theta}^{\to v} \\ b_v \end{pmatrix} \right\|_q^q$$

This shows the claim for $v$.

Assume the result to be true for $v \notin N_{\text{in}}$ and all the neurons before $v$ in the considered topological order (in particular, for every $u \in \text{ant}(u)$). By Equation (11), we have

$$\|\Phi^{\to v}(\boldsymbol{\theta})\|_q^q = |b_v|^q + \sum_{u \in \text{ant}(v)} \|\Phi^{\to u}(\boldsymbol{\theta})\|_q^q |\boldsymbol{\theta}^{u \to v}|^q.$$

The induction hypothesis guarantees that $c_u := \|\Phi^{\to u}(\boldsymbol{\theta})\|_q = \left\| \begin{pmatrix} \boldsymbol{\theta}^{\to u} \\ b_u \end{pmatrix} \right\|_q$ for every $u \in$ ant$(v)$. By Equation (10), we also have $c_u \in \{0, 1\}$ for every $u \in \text{ant}(v)$. When $c_u = 1$, it clearly holds $\|\Phi^{\to u}(\boldsymbol{\theta})\|_q^q |\boldsymbol{\theta}^{u \to v}|^q = |\boldsymbol{\theta}^{u \to v}|^q$. Otherwise, $c_u = 0$, hence $\boldsymbol{\theta}^{u \to} = 0$ as proved above, and we also obtain $\|\Phi^{\to u}(\boldsymbol{\theta})\|_q^q |\boldsymbol{\theta}^{u \to v}|^q = |\boldsymbol{\theta}^{u \to v}|^q$. We deduce that

$$\|\Phi^{\to v}(\boldsymbol{\theta})\|_q^q = |b_v|^q + \sum_{u \in \text{ant}(v)} |\boldsymbol{\theta}^{u \to v}|^q = \left\| \begin{pmatrix} \boldsymbol{\theta}^{\to v} \\ b_v \end{pmatrix} \right\|_q^q.$$

This concludes the induction.

**Step 4.** Each of the activation functions $\rho \in \{\text{id}, \text{ReLU}, k\text{-pool}\}$ is positive homogeneous, so classical arguments for layered fully-connected networsk (see e.g. (Stock & Gribonval, 2023, Lemma 1 and Theorem 1)) are easily adapted to show that $R_{\boldsymbol{\theta}}$ and $\Phi(\boldsymbol{\theta})$ are both unchanged at each iteration of Algorithm 1 where $\lambda_v \neq 0$. Inspecting the iterations involving a neuron such that $\lambda_v = 0$ reveals that: (i) $\Phi_p(\boldsymbol{\theta}) = 0$ is unchanged for all paths $p$ going through this neuron; (ii) $\Phi_p(\boldsymbol{\theta})$ is also unchanged for other paths since they only involve entries of $\boldsymbol{\theta}$ that are kept identical. As a result, $\Phi(\boldsymbol{\theta})$ is also preserved at each such iteration. Finally, for each of the considered activations, $\lambda_v = 0$ implies that $v(\boldsymbol{\theta}, x) = 0$ for any $x$ (Equation (1)), so setting the outgoing weights $\boldsymbol{\theta}^{v \to} = 0$ does not change $R_{\boldsymbol{\theta}}$. This completes the proof. $\qquad\square$

## C  PATH-NORMS AND PRODUCTS OF OPERATOR NORMS

For $0 < q \leqslant \infty$, recall that the mixed (quasi-)norm $\|M\|_{q,\infty}$ of a matrix $M$ is:

$$\|M\|_{q,\infty} = \max_{\text{row of } M} \|\text{row}\|_q.$$

*For $q = 1$, this is the operator norm of $M$ induced by the $L^\infty$-norm on the input and output spaces. As a consequence, this is the Lipschitz constant of $x \mapsto Mx + b$ with respect to these norms, and by composition, it is well-known (see, e.g., Combettes & Pesquet (2020), where tighter bounds are also provided) that it can be used to derive a bound on the Lipschitz constant of layered fully-connected ReLU networks.*

**Lemma C.1** (see, *e.g.*, Combettes & Pesquet (2020)). *For a layered fully-connected ReLU network with biases of the form*

$$R_{\boldsymbol{\theta}}(x) = M_L \text{ReLU}(M_{L-1} \dots \text{ReLU}(M_1 x + b_1) + b_{L-1}) + b_L,$$

*it holds for every $x, x'$ and for $q = 1$:*

$$\|R_{\boldsymbol{\theta}}(x) - R_{\boldsymbol{\theta}}(x')\|_\infty \leqslant \left( \prod_{\ell=1}^L \|M_\ell\|_{q,\infty} \right) \|x - x'\|_\infty.$$

*Proof.* As this is well-known we only reproduce the proof for $L = 2$, and the general case easily follows by an induction: (highlighting in **orange** $x$ and $x'$):

$$
\begin{aligned}
\|R_{\boldsymbol{\theta}}(x) - R_{\boldsymbol{\theta}}(x')\|_\infty &= \|M_2 \operatorname{ReLU}(M_1 \boldsymbol{x} + b_1) + b_2 - M_2 \operatorname{ReLU}(M_1 \boldsymbol{x'} + b_1) + b_2\|_\infty \\
&= \|M_2 \left(\operatorname{ReLU}(M_1 \boldsymbol{x} + b_1) - \operatorname{ReLU}(M_1 \boldsymbol{x'} + b_1)\right)\|_\infty \\
&\leqslant \|M_2\|_{1,\infty} \|\operatorname{ReLU}(M_1 \boldsymbol{x} + b_1) - \operatorname{ReLU}(M_1 \boldsymbol{x'} + b_1)\|_\infty \\
&\leqslant \|M_2\|_{1,\infty} \|M_1 \boldsymbol{x} + b_1 - M_1 \boldsymbol{x'} + b_1\|_\infty \qquad \text{ReLU is 1-Lipschitz} \\
&\leqslant \|M_2\|_{1,\infty} \|M_1 (\boldsymbol{x} - \boldsymbol{x'})\|_\infty \\
&\leqslant \|M_2\|_{1,\infty} \|M_1\|_{1,\infty} \|\boldsymbol{x} - \boldsymbol{x'}\|_\infty .
\end{aligned}
$$

$\square$

Another bound on the Lipchitz constant with respect to the $L^\infty$-norm on both the input and the output is the mixed path-norm $\|\Phi(\boldsymbol{\theta})\|_{1,\infty}$ (Lemma A.2), which is even valid for an arbitrary DAG ReLU network. Is there any relation between the mixed path-norm $\|\Phi(\boldsymbol{\theta})\|_{1,\infty}$ and the product of operator norm $\prod_{\ell=1}^L \|M_\ell\|_{1,\infty}$ for layered fully-connected networks (LFCNs)?

**Comparing the Lipschitz constants for a scalar-valued LFCN.** It is known that for a *scalar-valued* (i.e., with $d_{\text{out}} = 1$) LFCN without biases, corresponding to $R_{\boldsymbol{\theta}}(x) = M_L \operatorname{ReLU}(M_{L-1} \dots \operatorname{ReLU}(M_1 x))$, the bound $\|\Phi(\boldsymbol{\theta})\|_q \leqslant \prod_{\ell=1}^L \|M_\ell\|_{q,\infty}$ holds (Neyshabur et al., 2015, Theorem 5) for every $0 < q \leqslant \infty$, with equality if the parameters have been rescaled properly. In the specific case of a scalar-valued DAG ReLU network, it holds $\|\Phi(\boldsymbol{\theta})\|_q = \|\Phi(\boldsymbol{\theta})\|_{q,\infty}$ (Definition A.4). Thus, the result from Neyshabur et al. (2015) shows that the mixed path-norm $\|\Phi(\boldsymbol{\theta})\|_{1,\infty}$ is a tighter Lipschitz bound than the product of operator norms $\prod_{\ell=1}^L \|M_\ell\|_{1,\infty}$ in the specific case of *scalar-valued* LFCN.

We now extend the comparison made in Neyshabur et al. (2015) from a *scalar-valued LFCN* without biases to an *arbitrary DAG ReLU network*. We will again derive that the mixed path-norm is a lower bound on (an extension of) the product of layers' norm for a general *DAG ReLU network*. Compared to Neyshabur et al. (2015), this requires both our extension of the notion of path-norm to mixed path-norms, and extending the notion of product of layers' norm to deal with a DAG.

*From now on, we always consider $q \in (0, \infty)$ and $r \in (0, \infty]$ if not specified otherwise.*

**From scalar-valued to vector-valued networks: from standard to mixed $L^q$-norms.** Lemma A.2 shows that going from a *scalar-valued* network (LFCN or, more generally a DAG) to a *vector-valued* network requires going from standard $L^q$-norms $\|\Phi(\boldsymbol{\theta})\|_q$, as considered in Neyshabur et al. (2015), to mixed $L^q$-norms $\|\Phi(\boldsymbol{\theta})\|_{q,r}$, which is, to the best of our knowledge, first introduced in Definition A.4.

**From LFCN to DAG: extending the product of layers' norm.** There is no notion a of a "layer" $M_\ell$ in a general DAG, hence the product $\prod_{\ell=1}^L \|M_\ell\|_{q,\infty}$ makes no sense anymore in this general context. We now introduce a quantity that generalizes this product for a general DAG.

**Definition C.1.** *Consider $q \in (0, \infty)$. For practical purposes, we denote $\gamma_v := b_v$ when $v \in N \setminus N_{in}$, and $\gamma_v := 1$ if $v \in N_{in}$. For every path $p \in \mathcal{P}$, consider*

$$
\Pi(\boldsymbol{\theta}, p, q) := \left( \sum_{\ell=0}^{\texttt{length}(p)} |\gamma_{p_\ell}|^q \prod_{k=\ell+1}^{\texttt{length}(p)} \|\boldsymbol{\theta}^{\to p_k}\|_q^q \right)^{1/q}, \tag{12}
$$

*with the convention that an empty product is equal to one. For $q \in (0, \infty)$ and $r \in (0, \infty]$, define:*

$$
\Pi_{q,r}(\boldsymbol{\theta}) := \left\| \left( \max_{p \in \mathcal{P}^{\to v}} \Pi(\boldsymbol{\theta}, p, q) \right)_{v \in N_{out}} \right\|_r. \tag{13}
$$

Let us check that $\Pi_{q,\infty}$ generalizes the product of layers' norm to an arbitrary DAG.

**Lemma C.2.** *Consider $q \in (0, \infty)$. In a LFCN $R_{\boldsymbol{\theta}}(x) = M_L \, \text{ReLU}(M_{L-1} \dots \text{ReLU}(M_1 x))$ with $L$ layers and without biases, it holds:*

$$\Pi_{q,\infty}(\boldsymbol{\theta}) = \prod_{\ell=1}^{L} \|M_\ell\|_{q,\infty}.$$

*Proof.* In a LFCN, all the neurons of two consecutive layers are connected, so $\prod_{\ell=1}^{L} \|M_\ell\|_{q,\infty}$ is also the maximum over all paths starting from an input neuron (and ending at an output neuron, by definition of $\mathcal{P}$) of the product of $L^q$-norms:

$$\prod_{\ell=1}^{L} \|M_\ell\|_{q,\infty} = \max_{p \in \mathcal{P}, p_0 \in N_{\text{in}}} \prod_{\ell=1}^{\text{length}(p)} \|\boldsymbol{\theta}^{\to p_\ell}\|_q.$$

Since there are no biases (corresponding to $\gamma_u = b_u = 0$ for every $u \notin N_{\text{in}}$) we have

$$\Pi(\boldsymbol{\theta}, p, q) = \begin{cases} \prod_{\ell=1}^{\text{length}(p)} \|\boldsymbol{\theta}^{\to p_\ell}\|_q & \text{if } p_0 \in N_{\text{in}}, \\ 0 & \text{otherwise.} \end{cases}$$

This shows the claim:

$$\Pi_{q,\infty}(\boldsymbol{\theta}) = \max_{v \in N_{\text{out}}} \max_{p \in \mathcal{P}^{\to v}} \Pi(\boldsymbol{\theta}, p, q) = \max_{p \in \mathcal{P}} \Pi(\boldsymbol{\theta}, p, q)$$

$$= \max_{p \in \mathcal{P}, p_0 \in N_{\text{in}}} \prod_{\ell=1}^{\text{length}(p)} \|\boldsymbol{\theta}^{\to p_\ell}\|_q = \prod_{\ell=1}^{\text{length}(p)} \|M_\ell\|_{q,\infty}. \qquad \square$$

The next theorem proves that the mixed path-norm is a lower bound on this extended product of layers' norm. In particular, this shows that the Lipschitz bound given by $\|\Phi(\boldsymbol{\theta})\|_{1,\infty}$ (which is computable in a single forward-pass by Theorem A.1) is always at least as good as the one given by $\prod_{\ell=1}^{L} \|M_\ell\|_{1,\infty}$ for a LFCN without biases. This extends as claimed the result from Neyshabur et al. (2015) to the vector-valued case.

**Theorem C.1.** *Consider $q \in (0, \infty)$ and $r \in (0, \infty]$. For every DAG ReLU network (Definition 2.2) and every parameters $\boldsymbol{\theta}$:*

$$\|\Phi(\boldsymbol{\theta})\|_{q,r} \leqslant \Pi_{q,r}(\boldsymbol{\theta}).$$

*If $\boldsymbol{\theta}$ is $q$-normalized (Definition B.1) then:*

$$\|\Phi(\boldsymbol{\theta})\|_{q,r} = \Pi_{q,r}(\boldsymbol{\theta}) = \left\| \left( \left\| \begin{pmatrix} \boldsymbol{\theta}^{\to v} \\ b_v \end{pmatrix} \right\|_q \right)_{v \in N_{out}} \right\|_r.$$

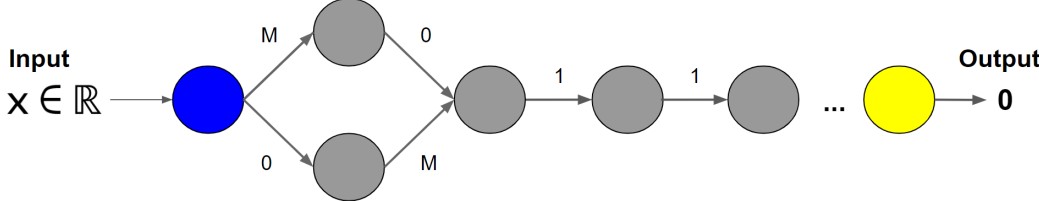

Figure 2: A network which path-norm is zero while the product of operator norms scales as $M^2$.

*Proof of Theorem C.1.* First, when $\boldsymbol{\theta}$ is $q$-normalized, by Definition B.1 and Equation (8):

$$\|\Phi(\boldsymbol{\theta})\|_{q,r} = \left\| \left( \left\| \begin{pmatrix} \boldsymbol{\theta}^{\to v} \\ b_v \end{pmatrix} \right\|_q \right)_{v \in N_{out}} \right\|_r.$$

Let us prove by induction on a topological sorting of the neurons that for every $v \in N$:

$$\|\Phi^{\to v}(\boldsymbol{\theta})\|_q \leqslant \max_{p \in \mathcal{P}^{\to v}} \Pi(\boldsymbol{\theta}, p, q) \tag{14}$$

with equality if $\boldsymbol{\theta}$ is $q$-normalized. As a direct consequence of Equations (8) and (13), this will prove that $\|\Phi(\boldsymbol{\theta})\|_{q,r} \leqslant \Pi_{q,r}(\boldsymbol{\theta})$, with equality when $\boldsymbol{\theta}$ is $q$-normalized, yielding all the claimed results.

We start with $v \in N_{\text{in}}$ since input neurons are the first to appear in a topological sorting. In this case, the only path in $\mathcal{P}^{\to v}$ is $p = v$, for which it holds $\Pi(\boldsymbol{\theta}, p, q) = |\gamma_{p_0}| = |\gamma_v| = 1$ by Definition C.1. Moreover, we also have $\Phi^{\to v}(\boldsymbol{\theta}) = 1$ (Definition A.3). This proves Equation (14) and the case of equality for input neurons even if $\boldsymbol{\theta}$ is not normalized.

Now, consider $v \notin N_{\text{in}}$ and assume Equation (14), and the case of equality for $q$-normalized parameters, to be true for every antecedent of $v$. In this case, we have $\Phi^{\to v}(\boldsymbol{\theta}) = \begin{pmatrix} (\Phi^{\to u}(\boldsymbol{\theta})\boldsymbol{\theta}^{u \to v})_{u \in \text{ant}(v)} \\ b_v \end{pmatrix}$ so it holds

$$\|\Phi^{\to v}(\boldsymbol{\theta})\|_q^q = |b_v|^q + \sum_{u \in \text{ant}(v)} \|\Phi^{\to u}(\boldsymbol{\theta})\|_q^q |\boldsymbol{\theta}^{u \to v}|^q \leqslant |b_v|^q + \|\boldsymbol{\theta}^{\to v}\|_q^q \max_{u \in \text{ant}(v)} \|\Phi^{\to u}(\boldsymbol{\theta})\|_q^q.$$

Using the induction hypothesis on every $u \in \text{ant}(v)$ yields:

$$\|\Phi^{\to v}(\boldsymbol{\theta})\|_q^q \leqslant |b_v|^q + \|\boldsymbol{\theta}^{\to v}\|_q^q \max_{u \in \text{ant}(v)} \max_{p \in \mathcal{P}^{\to u}} (\Pi(\boldsymbol{\theta}, p, q))^q$$

Consider $u \in \text{ant}(v)$ and $p \in \mathcal{P}^{\to u}$, and denote by $\tilde{p} = p \to v$ the path $p$ concatenated with the edge $u \to v$. By Definition C.1, we have (highlighting in **orange** the important changes)

$$|b_v|^q + \|\boldsymbol{\theta}^{\to v}\|_q^q (\Pi(\boldsymbol{\theta}, p, q))^q$$

$$= \underbrace{|b_v|^q}_{=|\gamma_v|^q \text{ since } v \notin N_{\text{in}}} + \|\boldsymbol{\theta}^{\to v}\|_q^q \sum_{\ell=0}^{\text{length}(p)} |\gamma_{p_\ell}|^q \prod_{k=\ell+1}^{\text{length}(p)} \|\boldsymbol{\theta}^{\to p_k}\|_q^q$$

$$= |\gamma_{\tilde{p}_{\text{end}}}|^q + \|\boldsymbol{\theta}^{\to \tilde{p}_{\text{end}}}\|_q^q \sum_{\ell=0}^{\text{length}(\tilde{p})-1} |\gamma_{\tilde{p}_\ell}|^q \prod_{k=\ell+1}^{\text{length}(\tilde{p})-1} \|\boldsymbol{\theta}^{\to \tilde{p}_k}\|_q^q \qquad = (\Pi(\boldsymbol{\theta}, \tilde{p}, q))^q$$

We deduce that

$$\|\Phi^{\to v}(\boldsymbol{\theta})\|_q^q \leqslant \max_{u \in \text{ant}(v)} \max_{p \in \mathcal{P}^{\to u}} (\Pi(\boldsymbol{\theta}, p \to u, q))^q = \max_{\tilde{p} \in \mathcal{P}^{\to v}} (\Pi(\boldsymbol{\theta}, \tilde{p}, q))^q.$$

This proves Equation (14) for $v$. To conclude the induction, it remains to treat the equality case for $v$ assuming that $\boldsymbol{\theta}$ is $q$-normalized. Since $v \notin N_{\text{in}}$, $\text{ant}(v) \neq \emptyset$, and since each neuron $u \in \text{ant}(v)$ cannot be an output neuron, the fact that $\boldsymbol{\theta}$ is $q$-normalized implies by Definition B.1 that $\|\Phi^{\to u}(\boldsymbol{\theta})\|_q \in \{0, 1\}$, with $\boldsymbol{\theta}^{u \to} = 0$ as soon as $\|\Phi^{\to u}(\boldsymbol{\theta})\|_q = 0$. This implies

$$\|\Phi^{\to u}(\boldsymbol{\theta})\|_q^q |\boldsymbol{\theta}^{u \to v}|^q = |\boldsymbol{\theta}^{u \to v}|^q$$

As a result

$$\sum_{u \in \text{ant}(v)} \|\Phi^{\to u}(\boldsymbol{\theta})\|_q^q |\boldsymbol{\theta}^{u \to v}|^q = \sum_{u \in \text{ant}(v)} |\boldsymbol{\theta}^{u \to v}|^q = \|\boldsymbol{\theta}^{\to v}\|_q^q = \|\boldsymbol{\theta}^{\to v}\|_q^q \max_{u \in \text{ant}(v)} \|\Phi^{\to u}(\boldsymbol{\theta})\|_q^q.$$

By the induction hypothesis, we also have

$$\max_{u \in \text{ant}(v)} \|\Phi^{\to u}(\boldsymbol{\theta})\|_q^q = \max_{u \in \text{ant}(v)} \max_{p \in \mathcal{P}^{\to u}} (\Pi(\boldsymbol{\theta}, p, q))^q.$$

Putting everything together yields

$$\|\Phi^{\to v}(\boldsymbol{\theta})\|_q^q = |b_v|^q + \sum_{u \in \text{ant}(v)} \|\Phi^{\to u}(\boldsymbol{\theta})\|_q^q |\boldsymbol{\theta}^{u \to v}|^q$$

$$= |b_v|^q + \|\boldsymbol{\theta}^{\to v}\|_q^q \max_{u \in \text{ant}(v)} \max_{p \in \mathcal{P}^{\to u}} (\Pi(\boldsymbol{\theta}, p, q))^q$$

$$= \max_{p \in \mathcal{P}^{\to v}} \Pi(\boldsymbol{\theta}, p, q)^q.$$

This shows the case of equality and concludes the induction. $\square$

# D  RELEVANT (AND APPARENTLY NEW) CONTRACTION LEMMAS

The main result is Lemma D.1.

**Lemma D.1.** *Consider finite sets $I, W, Z$, and for each $z \in Z$, consider a set $T^z \subset (\mathbb{R}^W)^I$. We denote $t = (t_i)_{i \in I} \in T^z$ with $t_i = (t_{i,w})_{w \in W} \in \mathbb{R}^W$. Consider functions $f_{i,z} : \mathbb{R}^W \to \mathbb{R}$ and a finite family $\varepsilon = (\varepsilon_j)_{j \in J}$ of independent identically distributed Rademacher variables, with the index set $J$ that will be clear from the context. Finally, consider a convex and non-decreasing function $g : \mathbb{R} \to \mathbb{R}$. Assume that at least one of the following setting holds.*

***Setting 1: scalar input case.*** *$|W| = 1$ and for every $i \in I$ and $z \in Z$, $f_{i,z}$ is 1-Lipschitz with $f_{i,z}(0) = 0$.*

***Setting 2: ∗-max-pooling case.*** *For every $i \in I$ and $z \in Z$, there is $k_{i,z} \in \mathbb{N}_{>0}$ such that for every $t \in T^z$, $f_{i,z}(t) = t_{(k_{i,z})}$ is the $k_{i,z}$-th largest coordinate of $t$.*

*Then we have:*

$$\mathbb{E} \max_{z \in Z} \sup_{t \in T^z} g\left( \sum_{i \in I} \varepsilon_{i,z} f_{i,z}(t_i) \right) \leqslant \mathbb{E} \max_{z \in Z} \sup_{t \in T^z} g\left( \sum_{i \in I, w \in W} \varepsilon_{i,w,z} t_{i,w} \right). \tag{15}$$

The scalar input case is a simple application of Theorem 4.12 in Ledoux & Talagrand (1991). Indeed, for $z \in Z$ and $t \in T^z$, define $s(z,t) \in \mathbb{R}^{I \times Z}$ to be the matrix with coordinates $(i, v) \in I \times Z$ given by $[s(z,t)]_{i,v} = t_i$ if $v = z$, 0 otherwise. Define also $f_{i,v} = f_i$. Since $f_{i,v}(0) = 0$, it holds:

$$\sum_{i \in I} \varepsilon_{i,z} f_i(t_i) = \sum_{i \in I, v \in Z} \varepsilon_{i,v} f_{i,v}([s(z,t)]_{i,v}).$$

If $S := \{s(z,t), z \in Z, t \in T^z\}$ and $J := I \times Z$ then the result claimed in the scalar case reads

$$\mathbb{E} \sup_{s \in S} g\left( \sum_{j \in J} \varepsilon_j f_j(s_j) \right) \leqslant \mathbb{E} \sup_{s \in S} g\left( \sum_{j \in J} \varepsilon_j s_j \right).$$

The latter is true by Theorem 4.12 of Ledoux & Talagrand (1991). However, we present an additional proof below, which we employ to establish the new scenario involving ∗-max-pooling. This alternative proof closely follows the structure of the proof outlined in Theorem 4.12 of Ledoux & Talagrand (1991): the beginning of the proof is the same for the scalar case and the ∗-max-pooling case, and then the arguments become specific to each case.

Note that Theorem 4.12 of Ledoux & Talagrand (1991) does not apply for the ∗-max-pooling case because the $t_i$'s are now vectors. The most related result we could find is a vector-valued contraction inequality (Maurer, 2016) that is known in the specific case where $|Z| = 1$, $g$ is the identity, and for arbitrary 1-Lipschitz functions $f_{i,z}$ such that $f_{i,z}(0) = 0$ (with a different proof, and with a factor $\sqrt{2}$ on the right-hand side). Here, the vector-valued case we are interested in is $f_{i,z} = k_{i,z}$-`pool` and $g = \exp$, which is covered by Lemma D.1. We could not find it stated elsewhere.

In the proof of Lemma D.1, we reduce to the simpler case where $|Z| = 1$ and $|I| = 1$ that corresponds to the next lemma. Again, the scalar input case is given by Ledoux & Talagrand (1991, Equation (4.20)) while the ∗-max-pooling case is apparently new.

**Lemma D.2.** *Consider a finite set $W$, a set $T$ of elements $t = (t_1, t_2) \in \mathbb{R}^W \times \mathbb{R}$ and a function $f : \mathbb{R}^W \to \mathbb{R}$. Consider also a convex non-decreasing function $F : \mathbb{R} \to \mathbb{R}$ and a family of iid Rademacher variables $(\varepsilon_j)_{j \in J}$ where $J$ will be clear from the context. Assume that we are in one of the two following situations.*

***Scalar input case.*** *$f$ is 1-Lipschitz, satisfies $f(0) = 0$ and has a scalar input ($|W| = 1$).*

***∗-max-pooling case.*** *There is $k \in \mathbb{N}_{>0}$ such that $f$ computes the $k$-th largest coordinate of its input.*

*Denoting $t_1 = (t_{1,w})_{w \in W}$, it holds:*

$$\mathbb{E} \sup_{t \in T} F\left( \varepsilon_1 f(t_1) + t_2 \right) \leqslant \mathbb{E} \sup_{t \in T} F\left( \sum_w \varepsilon_{1,w} t_{1,w} + t_2 \right).$$

The proof of Lemma D.2 is postponed. We now prove Lemma D.1.

*Proof of Lemma D.1.* First, because of the Lipschitz assumptions on the $f_i$'s and the convexity of $g$, everything is measurable and the expectations are well defined.

We prove the result by reducing to the simpler case of Lemma D.2. This is inspired by the reduction done in the proof of Ledoux & Talagrand (1991, Theorem 4.12) in the special case of scalar $t_i$'s ($|W| = 1$).

**Reduce to the case $|Z| = 1$ by conditioning and iteration.** For $z \in Z$, define

$$A_z := \sup_{t \in T^z} g\left(\sum_{i \in I} \varepsilon_{i,z} f_{i,z}(t_i)\right),$$

$$B_z := \sup_{t \in T^z} g\left(\sum_{i \in I, w \in W} \varepsilon_{i,w,z} t_{i,w}\right).$$

Lemma D.3 applies since these random variables are independent. Thus, it is enough to prove that for every $c \in [-\infty, \infty)$:

$$\mathbb{E}\max(A_z, c) \leqslant \mathbb{E}\max(B_z, c).$$

Define $F(x) = \max(g(x), c)$. This can be rewritten as (inverting the supremum and the maximum)

$$\mathbb{E}\sup_{t \in T^z} F\left(\sum_{i \in I} \varepsilon_{i,z} f_{i,z}(t_i)\right) \leqslant \mathbb{E}\sup_{t \in T^z} F\left(\sum_{i \in I, w \in W} \varepsilon_{i,w,z} t_{i,w}\right). \tag{16}$$

We just reduced to the case where there is a single $z$ to consider, up to the price of replacing $g$ by $F$. Since $g$ and $x \mapsto \max(x, c)$ are non-decreasing and convex, so is $F$ by composition. Alternatively, note that we could also have reduced to the case $|Z| = 1$ by defining $S := \{s(z, t), z \in Z, t \in T^z\}$ just as it is done right after the statement of Lemma D.1. In order to apply Lemma D.2, it remains to reduce to the case $|I| = 1$.

**Reduce to the case $|I| = 1$ by conditioning and iteration.** Lemma D.4 shows that in order to prove Equation (16), it is enough to prove that for every $i \in I$ and every subset $R \subset \mathbb{R}^W \times R$, denoting $r = (r_1, r_2) \in \mathbb{R}^W \times \mathbb{R}$, it holds

$$\mathbb{E}\sup_{r \in R} F\left(\varepsilon_{i,z} f_{i,z}(r_1) + r_2\right) \leqslant \mathbb{E}\sup_{r \in R} F\left(\sum_{w \in W} \varepsilon_{i,w,z} r_{1,w} + r_2\right).$$

We just reduced to the case $|I| = 1$ since one can now consider the indices $i$ one by one. The latter inequality is now a direct consequence of Lemma D.2. This proves the result. □

**Lemma D.3.** *Consider a finite set $Z$ and independent families of independent real random variables $(A_z)_{z \in Z}$ and $(B_z)_{z \in Z}$. If for every $z \in Z$ and every constant $c \in [-\infty, \infty)$, it holds $\mathbb{E}\max(A_z, c) \leqslant \mathbb{E}\max(B_z, c)$, then*

$$\mathbb{E}\max_{z \in Z} A_z \leqslant \mathbb{E}\max_{z \in Z} B_z.$$

*Proof of Lemma D.3.* The proof is by conditioning and iteration. To prove the result, it is enough to prove that if

$$\mathbb{E}\max_{z \in Z} A_z \leqslant \mathbb{E}\max\left(\max_{z \in Z_1} A_z, \max_{z \in Z_2} B_z\right)$$

for some partition $Z_1, Z_2$ of $Z$, with $Z_2$ possibly empty for the initialization of the induction, then for every $z_0 \in Z_1$:

$$\mathbb{E}\max_{z \in Z} A_z \leqslant \mathbb{E}\max\left(\max_{z \in Z_1 \setminus \{z_0\}} A_z, \max_{z \in Z_2 \cup \{z_0\}} B_z\right),$$

with the convention that the maximum over an empty set is $-\infty$. Indeed, the claim would then come directly by induction on the size of $Z_2$.

Now, consider an arbitrary partition $Z_1, Z_2$ of $Z$, with $Z_2$ possibly empty, and consider $z_0 \in Z_1$. It is then enough to prove that

$$\mathbb{E}\max\left(\max_{z\in Z_1} A_z, \max_{z\in Z_2} B_z\right) \leqslant \mathbb{E}\max\left(\max_{z\in Z_1\setminus\{z_0\}} A_z, \max_{z\in Z_2\cup\{z_0\}} B_z\right). \qquad (17)$$

Define the random variable $C = \max\left(\max_{z\in Z_1\setminus\{z_0\}} A_z, \max_{z\in Z_2} B_z\right)$ which may be equal to $-\infty$ when the maximum is over empty sets, and which is independent of $A_{z_0}$ and $B_{z_0}$. It holds:

$$\max\left(\max_{z\in Z_1} A_z, \max_{z\in Z_2} B_z\right) = \max\left(A_{z_0}, C\right)$$

and

$$\max\left(\max_{z\in Z_1\setminus\{z_0\}} A_z, \max_{z\in Z_2\cup\{z_0\}} B_z\right) = \max\left(B_{z_0}, C\right).$$

Equation (17) is then equivalent to

$$\mathbb{E}\max(A_{z_0}, C) \leqslant \mathbb{E}\max(B_{z_0}, C)$$

with $C$ independent of $A_{z_0}$ and $B_{z_0}$. For a constant $c \in [-\infty, \infty)$, denote $A(c) = \mathbb{E}\max(A_{z_0}, c)$ and $B(c) = \mathbb{E}\max(B_{z_0}, c)$. We have:

$$\begin{aligned}\mathbb{E}\max(A_{z_0}, C) &= \mathbb{E}\left(\mathbb{E}\left(\max\left(A_{z_0}, C\right)|A_{z_0}\right)\right) && \text{law of total expectation}\\ &= \mathbb{E}A(C) && \text{independence of } C \text{ and } A_{z_0}.\end{aligned}$$

and similarly $\mathbb{E}\max(B_{z_0}, C) = \mathbb{E}B(C)$. It is then enough to prove that $A(C) \leqslant B(C)$ almost surely. Since $C \in [-\infty, \infty)$, this is true by assumption. This proves the claims. $\qquad\square$

**Lemma D.4.** *Consider finite sets $I, W$ and independent families of independent real random variables $(\varepsilon_i)_{i\in I}$ and $(\varepsilon_{i,w})_{i\in I, w\in W}$. Consider functions $f_i : \mathbb{R}^W \to \mathbb{R}$ and $F : \mathbb{R} \to \mathbb{R}$ that are continuous. Assume that for every $i \in I$ and every subset $R \subset \mathbb{R}^W \times \mathbb{R}$, denoting $r = (r_1, r_2) \in R$ with $r_1 = (r_{1,w})_w \in \mathbb{R}^W$ and $r_2 \in \mathbb{R}$ the components of $r$, it holds*

$$\mathbb{E}\sup_{r\in R} F(\varepsilon_i f_i(r_1) + r_2) \leqslant \mathbb{E}\sup_{r\in R} F(\sum_{w\in W} \varepsilon_{i,w} r_{1,w} + r_2).$$

*Consider an arbitrary $T \subset (\mathbb{R}^W)^I$ and for $t = (t_i)_{i\in I} \in T$, denote $t_{i,w}$ the $w$-th coordinate of $t_i \in \mathbb{R}^W$. It holds:*

$$\mathbb{E}\sup_{t\in T} F(\sum_{i\in I} \varepsilon_i f_i(t_i)) \leqslant \mathbb{E}\sup_{t\in T} F(\sum_{i\in I, w\in W} \varepsilon_{i,w} t_{i,w}).$$

*Proof of Lemma D.4.* The continuity assumption on $F$ and the $f_i$'s is only used to make all the considered suprema measurable. The proof goes by conditioning and iteration. For any $J \subset I$, denote $\varepsilon_J$ the family that contains both $(\varepsilon_j)_{j\in J}$ and $(\varepsilon_{j,w})_{j\in J, w\in W}$. Define

$$\begin{aligned} h_J(t, \varepsilon_J) &:= \sum_{j\in J} \varepsilon_j f_j(t_j),\\ H_J(t, \varepsilon_J) &:= \sum_{j\in J, w\in W} \varepsilon_{j,w} t_{j,w}, \end{aligned}$$

with the convention that an empty sum is zero. To make notations lighter, if $J = \{j\}$ then we may write $h_j$ and $H_j$ instead of $h_J$ and $H_J$. We also omit to write the dependence on $\varepsilon_J$ as soon as possible. What we want to prove is thus equivalent to

$$\mathbb{E}\sup_{t\in T} F(h_I(t)) \leqslant \mathbb{E}\sup_{t\in T} F(H_I(t)).$$

It is enough to prove that for every partition $I_1, I_2$ of $I$, with $I_2$ possibly empty, if

$$\mathbb{E}\sup_{t\in T} F(h_I(t)) \leqslant \mathbb{E}\sup_{t\in T} F(h_{I_1}(t) + H_{I_2}(t)),$$

then for every $j \in I_1$,

$$\mathbb{E} \sup_{t \in T} F(h_I(t)) \leqslant \mathbb{E} \sup_{t \in T} F(h_{I_1 \setminus \{j\}}(t) + H_{I_2 \cup \{j\}}(t)).$$

Indeed, the result would then come by induction on the size of $I_2$. Fix an arbitrary partition $I_1$, $I_2$ of $I$ with $I_2$ possibly empty, and $j \in I_1$. It is then enough to prove that

$$\mathbb{E} \sup_{t \in T} F(h_{I_1}(t) + H_{I_2}(t)) \leqslant \mathbb{E} \sup_{t \in T} F(h_{I_1 \setminus \{j\}}(t) + H_{I_2 \cup \{j\}}(t)). \tag{18}$$

Denote $\varepsilon_{-j} := \varepsilon_{I \setminus \{j\}}$ and $\varphi(t, \varepsilon_{-j}) := h_{I_1 \setminus \{j\}}(t, \varepsilon_{I_1 \setminus \{j\}}) + H_{I_2}(t, \varepsilon_{I_2}))$. It holds:

$$h_{I_1}(t) + H_{I_2}(t) = h_j(t, \varepsilon_j) + \varphi(t, \varepsilon_{-j})$$

and, writing $\varepsilon_{j,\cdot} = (\varepsilon_{j,w})_{w \in W}$:

$$h_{I_1 \setminus \{j\}}(t) + H_{I_2 \cup \{j\}}(t) = H_j(t, \varepsilon_{j,\cdot}) + \varphi(t, \varepsilon_{-j}).$$

Consider the measurable functions

$$g(\varepsilon_j, \varepsilon_{-j}) := \sup_{t \in T} F(h_j(t, \varepsilon_j) + \varphi(t, \varepsilon_{-j}))$$

and

$$G(\varepsilon_{j,\cdot}, \varepsilon_{-j}) := \sup_{t \in T} F(H_j(t, \varepsilon_{j,\cdot}) + \varphi(t, \varepsilon_{-j})).$$

Denote $\Delta$ the ambient space of $\varepsilon_{-j}$ and consider a constant $\delta \in \Delta$. Define $\hat{g}(\delta) = \mathbb{E} g(\varepsilon_j, \delta)$ and $\hat{G}(\delta) = \mathbb{E} G(\varepsilon_{j,\cdot}, \delta)$. It holds

$$\mathbb{E} \sup_{t \in T} F(h_{I_1}(t) + H_{I_2}(t)) = \mathbb{E} g(\varepsilon_j, \varepsilon_{-j}) \qquad \text{by definition of } g$$

$$= \mathbb{E} \left( \mathbb{E} \left( g(\varepsilon_j, \varepsilon_{-j}) | \varepsilon_{-j} \right) \right) \qquad \text{law of total expectation}$$

$$= \mathbb{E} \hat{g}(\varepsilon_{-j}) \qquad \text{independence of } \varepsilon_j \text{ and } \varepsilon_{-j}$$

and similarly $\mathbb{E} \sup_{t \in T} F(h_{I_1 \setminus \{j\}}(t) + H_{I_2 \cup \{j\}}(t)) = \mathbb{E} \hat{G}(\varepsilon_{-j})$. Thus, Equation (18) is equivalent to $\mathbb{E} \hat{g}(\varepsilon_{-j}) \leqslant \mathbb{E} \hat{G}(\varepsilon_{-j})$. For every $\delta \in \Delta$, we can define $R(\delta) = \{(t_j, \varphi(t, \delta)) \in \mathbb{R}^W \times \mathbb{R}, t \in T\}$ and it holds

$$\hat{g}(\delta) = \mathbb{E} \sup_{r \in R} F(\varepsilon_j f_j(r_1) + r_2)$$

and

$$\hat{G}(\delta) = \mathbb{E} \sup_{r \in R} F(\sum_{w \in} \varepsilon_{j,w} r_{1,w} + r_2).$$

Thus, $\hat{g}(\delta) \leqslant \hat{G}(\delta)$ for every $\delta \in \Delta$ by assumption. This shows the claim. $\qquad \square$

*Proof of Lemma D.2.* Recall that we want to prove

$$\mathbb{E} \sup_{t \in T} F(\varepsilon_1 f(t_1) + t_2) \leqslant \mathbb{E} \sup_{t \in T} F\left( \sum_{w \in W} \varepsilon_{1,w} t_{1,w} + t_2 \right). \tag{19}$$

**Scalar input case.** In this case, $|W| = 1$ *i.e.* the inputs $t_1$ are scalar and the result is well-known, see Ledoux & Talagrand (1991, Equation (4.20)).

**$k$-max-pooling case.** In this case, $f$ computes the $k$-th largest coordinate of its input. Computing explicitly the expectation where the only random thing is $\varepsilon_1 \in \{-1, 1\}$, the left-hand side of Equation (19) is equal to

$$\frac{1}{2} \sup_{t \in T} F(f(t_1) + t_2) + \frac{1}{2} \sup_{s \in T} F(-f(s_1) + s_2).$$

Consider $s, t \in T$. Recall that $s_1, t_1 \in \mathbb{R}^W$. Denote $s_{1,(k)}$ the $k$-th largest component of vector $s_1$. The set $\{w \in W : s_{1,w} \leqslant s_{1,(k)}\}$ has at least $|W| - k + 1$ elements, and $\{w \in W : t_{1,(k)} \leqslant t_{1,w}\}$

has at least $k$ elements, so their intersection is not empty. Consider *any*[8] $w(s,t)$ in this intersection. We are now going to use that both $f(t_1) = t_{1,(k)} \leqslant t_{1,w(s,t)}$ and $-f(s_1) = -s_{1,(k)} \leqslant -s_{1,w(s,t)}$. Even if we are not going to use it, note that this implies $f(t) - f(s) \leqslant t_{1,w(s,t)} - s_{1,w(s,t)}$: we are exactly using an argument that establishes that $f$ is 1-Lipschitz. Since $f(t_1) = t_{1,(k)} \leqslant t_{1,w(s,t)}$ and $F$ is non-decreasing, it holds:

$$F\left(f(t_1) + t_2\right) \leqslant F\left(t_{1,w(s,t)} + t_2\right)$$

$$\underset{\boldsymbol{\varepsilon} \text{ centered}}{=} F\left(t_{1,w(s,t)} + \mathbb{E}\left(\sum_{w \neq w(s,t)} \varepsilon_{1,w} t_{1,w}\right) + t_2\right)$$

$$\underset{\text{Jensen}}{\leqslant} \mathbb{E} F\left(t_{1,w(s,t)} + \sum_{w \neq w(s,t)} \varepsilon_{1,w} t_{1,w} + t_2\right).$$

Moreover, $-f(s_1) = -s_{1,(k)} \leqslant -s_{1,w(s,t)}$ so that in a similar way:

$$F\left(-f(s_1) + s_2\right) \leqslant F\left(-s_{1,w(s,t)} + s_2\right)$$

$$\leqslant F\left(-s_{1,w(s,t)} + \mathbb{E}\left(\sum_{w \neq w(s,t)} \varepsilon_{1,w} s_{1,w}\right) + s_2\right)$$

$$\leqslant \mathbb{E} F\left(-s_{1,w(s,t)} + \sum_{w \neq w(s,t)} \varepsilon_{1,w} s_{1,w} + s_2\right).$$

At the end, we get

$$\frac{1}{2} F\left(f(t_1) + t_2\right) + \frac{1}{2} F\left(-f(s_1) + s_2\right)$$

$$\leqslant \frac{1}{2} \mathbb{E} F\left(t_{1,w(s,t)} + \sum_{w \neq w(s,t)} \varepsilon_{1,w} t_{1,w} + t_2\right)$$

$$+ \frac{1}{2} \mathbb{E} F\left(-s_{1,w(s,t)} + \sum_{w \neq w(s,t)} \varepsilon_{1,w} s_{1,w} + s_2\right)$$

$$\leqslant \frac{1}{2} \mathbb{E} \sup_{r \in T} F\left(r_{1,w(s,t)} + \sum_{w \neq w(s,t)} \varepsilon_{1,w} r_{1,w} + r_2\right)$$

$$+ \frac{1}{2} \mathbb{E} \sup_{r \in T} F\left(-r_{1,w(s,t)} + \sum_{w \neq w(s,t)} \varepsilon_{1,w} r_{1,w} + r_2\right)$$

$$= \mathbb{E} \sup_{r \in T} F\left(\varepsilon_{1,w(s,t)} r_{1,w(s,t)} + \sum_{w \neq w(s,t)} \varepsilon_{1,w} r_{1,w} + r_2\right)$$

$$= \mathbb{E} \sup_{r \in T} F\left(\sum_{w} \varepsilon_{1,w} r_{1,w} + r_2\right).$$

The latter is independent of $s, t$. Taking the supremum over all $s, t \in T$ yields Equation (19) and thus the claim. $\qquad\square$

## E   PEELING ARGUMENT

First, we state a simple lemma that will be used several times.

---

[8]The choice of a specific $w$ has no importance, unlike when defining the activations of $k$-max-pooling neurons.

**Lemma E.1.** *Consider a vector $\varepsilon \in \mathbb{R}^n$ with iid Rademacher coordinates, meaning that $\mathbb{P}(\varepsilon_i = 1) = \mathbb{P}(\varepsilon_i = -1) = 1/2$. Consider a function $g : \mathbb{R} \to \mathbb{R}_{\geqslant 0}$. Consider a set $X \subset \mathbb{R}^n$. It holds:*

$$\mathbb{E}_{\boldsymbol{\varepsilon}} \sup_{x \in X} g\left(\left|\sum_{i=1}^{n} \varepsilon_i x_i\right|\right) \leqslant 2\mathbb{E}_{\boldsymbol{\varepsilon}} \sup_{x \in X} g\left(\sum_{i=1}^{n} \varepsilon_i x_i\right).$$

*Proof of Lemma E.1.* Since $g \geqslant 0$, it holds $g(|x|) \leqslant g(x) + g(-x)$. Thus

$$\mathbb{E}_{\boldsymbol{\varepsilon}} \sup_{x \in X} g\left(\left|\sum_{i=1}^{n} \varepsilon_i x_i\right|\right) \leqslant \mathbb{E}_{\boldsymbol{\varepsilon}} \sup_{x \in X} g\left(\sum_{i=1}^{n} \varepsilon_i x_i\right) + \mathbb{E}_{\boldsymbol{\varepsilon}} \sup_{x \in X} g\left(\sum_{i=1}^{n} (-\varepsilon_i) x_i\right).$$

Since $\varepsilon$ is symmetric, that is $-\varepsilon$ has the same distribution as $\varepsilon$, we deduce that the latter is just $2\mathbb{E}_{\boldsymbol{\varepsilon}} \sup_{x \in X} g\left(\sum_{i=1}^{n} \varepsilon_i x_i\right)$. This proves the claim. $\qquad\square$

**Notations** We now fix for all the next results of this section $n$ vectors $x_1, \ldots, x_n \in \mathbb{R}^{d_{\mathrm{in}}}$, for some $d_{\mathrm{in}} \in \mathbb{N}_{>0}$. We denote $x_{i,u}$ the coordinate $u$ of $x_i$.

For any neural network architecture, recall that $v(\boldsymbol{\theta}, x)$ is the output of neuron $v$ for parameters $\boldsymbol{\theta}$ and input $x$, and $\mathrm{ant}^d(v)$ is the set of neurons $u$ for which there exists a path from $u$ to $v$ of distance $d$. For a set of neurons $V$, denote $R_V(\boldsymbol{\theta}, x) = (v(\boldsymbol{\theta}, x))_{v \in V}$.

**Introduction to peeling** This section shows that some expected sum over output neurons $v$ can be reduced to an expected maximum over $\mathrm{ant}(v)$, and iteratively over an expected maximum over $\mathrm{ant}^d(v)$ for increasing $d$'s. Eventually, the maximum is only over input neurons as soon as $d$ is large enough. We start with the next lemma which is the initialization of the induction over $d$: it peels off the output neurons $v$ to reduce to their antecedents $\mathrm{ant}(v)$.

**Lemma E.2.** *Consider a neural network architecture as in Definition 2.2 with $N_{in} \cap N_{out} = \emptyset$. Consider an associated set $\boldsymbol{\Theta}$ of parameters $\boldsymbol{\theta}$ such that $\sum_{v \in N_{out}} \|\boldsymbol{\theta}^{\to v}\|_1 + |b_v| \leqslant r$. Consider a family of independent Rademacher variables $(\varepsilon_j)_{j \in J}$ with $J$ that will be clear from the context. Consider a non-decreasing function $g : \mathbb{R} \to \mathbb{R}_{\geqslant 0}$. Consider a new neuron $v_{bias}$ and set by convention $x_{v_{bias}} = 1$ for every input $x$. It holds*

$$\mathbb{E}_{\boldsymbol{\varepsilon}} g\left(\sup_{\boldsymbol{\theta} \in \boldsymbol{\Theta}} \sum_{\substack{i=1,\ldots,n, \\ v \in N_{out}}} \varepsilon_{i,v} v(\boldsymbol{\theta}, x_i)\right)$$

$$\leqslant \mathbb{E}_{\boldsymbol{\varepsilon}} g\left(r \max_{v \in N_{out}} \max_{u \in (\mathrm{ant}(v) \cap N_{in}) \cup \{v_{bias}\}} \left|\sum_{i=1}^{n} \varepsilon_{i,v} x_{i,u}\right|\right)$$

$$+ \mathbb{E}_{\boldsymbol{\varepsilon}} g\left(r \max_{v \in N_{out}} \max_{u \in \mathrm{ant}(v) \setminus N_{in}} \sup_{\boldsymbol{\theta}} \left|\sum_{i=1}^{n} \varepsilon_{i,v} u(\boldsymbol{\theta}, x_i)\right|\right)$$

*where in the last term if $\mathrm{ant}(v) \setminus N_{in} = \emptyset$, by convention $\max_{u \in \mathrm{ant}(v) \setminus N_{in}} = -\infty$, and $g(-\infty) := 0$.*

*Proof of Lemma E.2.* Recall that for a set of neurons $V$, we denote $R_V(\boldsymbol{\theta}, x) = (v(\boldsymbol{\theta}, x))_{v \in V}$. Recall that by Definition 2.2, output neurons $v$ have $\rho_v = \mathrm{id}$ so for every $v \in N_{\mathrm{out}}$, $\boldsymbol{\theta} \in \boldsymbol{\Theta}$ and every input $x$:

$$v(\boldsymbol{\theta}, x) = \left\langle \left( \begin{array}{c} \boldsymbol{\theta}^{\to v} \\ b_v \end{array} \right), \left( \begin{array}{c} R_{\mathrm{ant}(v)}(\boldsymbol{\theta}, x) \\ 1 \end{array} \right) \right\rangle.$$

Denote by $v_{\mathtt{bias}}$ a new neuron that computes the constant function equal to one ($v_{\mathtt{bias}}(\boldsymbol{\theta}, x) = 1$), we get:

$$
\mathbb{E}_{\boldsymbol{\varepsilon}} g \left( \sup_{\boldsymbol{\theta}} \sum_{\substack{i=1,\ldots,n \\ v \in N_{\text{out}}}} \varepsilon_{i,v} v(\boldsymbol{\theta}, x_i) \right)
$$

$$
= \mathbb{E}_{\boldsymbol{\varepsilon}} g \left( \sup_{\boldsymbol{\theta}} \sum_{v \in N_{\text{out}}} \left\langle \begin{pmatrix} \boldsymbol{\theta}^{\to v} \\ b_v \end{pmatrix}, \sum_{i=1}^n \varepsilon_{i,v} \begin{pmatrix} R_{\text{ant}(v)}(\boldsymbol{\theta}, x_i) \\ 1 \end{pmatrix} \right\rangle \right)
$$

$$
\underset{\text{Hölder}}{\leqslant} \mathbb{E}_{\boldsymbol{\varepsilon}} g \left( \sup_{\boldsymbol{\theta}} \underbrace{\left( \sum_{v \in N_{\text{out}}} \| \boldsymbol{\theta}^{\to v} \|_1 + |b_v| \right)}_{\leqslant r \text{ by assumption}} \max_{v \in N_{\text{out}}} \left( \left| \sum_{i=1}^n \varepsilon_{i,v} \right|, \max_{u \in \text{ant}(v)} \left| \sum_{i=1}^n \varepsilon_{i,v} u(\boldsymbol{\theta}, x_i) \right| \right) \right)
$$

$$
\leqslant \mathbb{E}_{\boldsymbol{\varepsilon}} g \left( r \max_{v \in N_{\text{out}}} \max_{u \in \text{ant}(v) \cup \{v_{\mathtt{bias}}\}} \sup_{\boldsymbol{\theta}} \left| \sum_{i=1}^n \varepsilon_{i,v} u(\boldsymbol{\theta}, x_i) \right| \right).
$$

Everything is non-negative so the maximum over $u \in \text{ant}(v) \cup \{v_{\mathtt{bias}}\}$ is smaller than the sum of the maxima over $u \in (\text{ant}(v) \cap N_{\text{in}}) \cup \{v_{\mathtt{bias}}\}$ and $u \in \text{ant}(v) \setminus N_{\text{in}}$. Note that when $u$ is an input neuron, it simply holds $u(\boldsymbol{\theta}, x_i) = x_{i,u}$. This proves the result. $\qquad \square$

We now show how to peel neurons to reduce the maximum over $\text{ant}^d(v)$ to $\text{ant}^{d+1}(v)$. Later, we will repeat that until the maximum is only on input neurons. Compared to the previous lemma, note the presence of an index $m = 1, \ldots, M$ in the maxima. This is because after $d$ steps of peeling (when the maximum over $u$ has been reduced to $u \in \text{ant}^d(v)$), we will have $M = K^{d-1}$ where $K$ is the kernel size. Indeed, the number of copies indexed by $m$ gets multiplied by $K$ after each peeling step.

**Lemma E.3.** *Consider a neural network architecture with an associated set $\boldsymbol{\Theta}$ of parameters $\boldsymbol{\theta}$ such that every neuron $v \notin N_{\text{out}} \cup N_{\text{in}}$ satisfies $\| \boldsymbol{\theta}^{\to v} \|_1 + |b_v| \leqslant 1$. Assume that $b_v = 0$ for every $v \in N_{\ast\text{-pool}}$. Consider a family of independent Rademacher variables $(\varepsilon_j)_{j \in J}$ with $J$ that will be clear from the context. Consider arbitrary $M, d \in \mathbb{N}$ and a convex non-decreasing function $g : \mathbb{R} \to \mathbb{R}_{\geqslant 0}$. Take a symbol $v_{bias}$ which does not correspond to a neuron ($v_{bias} \notin N$) and set by convention $x_{v_{bias}} = 1$ for every input $x$. Define $P := |\{k \in \mathbb{N}_{>0}, \exists u \in N_{k\text{-pool}}\}|$ as the number of different types of $\ast$-max-pooling neurons in $G$, and $K := \max_{u \in N_{\ast\text{-pool}}} |\text{ant}(u)|$ the maximal kernel size of the network ($K := 1$ if $P = 0$). It holds:*

$$
\mathbb{E}_{\boldsymbol{\varepsilon}} g \left( \max_{\substack{v \in N_{out}, \\ m=1,\ldots,M}} \max_{u \in \text{ant}^d(v) \setminus N_{in}} \sup_{\boldsymbol{\theta}} \left| \sum_{i=1}^n \varepsilon_{i,v,m} u(\boldsymbol{\theta}, x_i) \right| \right)
$$

$$
\leqslant (3 + 2P) \mathbb{E}_{\boldsymbol{\varepsilon}} g \left( \max_{\substack{v \in N_{out}, \\ m=1,\ldots,KM}} \max_{u \in (\text{ant}^{d+1}(v) \cap N_{in}) \cup \{v_{bias}\}} \left| \sum_{i=1}^n \varepsilon_{i,v,m} x_{i,u} \right| \right)
$$

$$
+ (3 + 2P) \mathbb{E}_{\boldsymbol{\varepsilon}} g \left( \max_{\substack{v \in N_{out}, \\ m=1,\ldots,KM}} \max_{u \in \text{ant}^{d+1}(v) \setminus N_{in}} \sup_{\boldsymbol{\theta}} \left| \sum_{i=1}^n \varepsilon_{i,v,m} u(\boldsymbol{\theta}, x_i) \right| \right)
$$

*with similar convention as in Lemma E.2 for empty maxima.*

*Proof.* **Step 1: split the neurons depending on their activation function.** In the term that we want to bound from above, the neurons $u \in \text{ant}^d(v) \setminus N_{\text{in}}$ are not input neurons so they compute something of the form $\rho_u(\ldots)$ where $\rho_u$ is the activation associated with $u$, 1-Lipschitz, and satisfies $\rho_u(0) = 0$. The first step of the proof is to get rid of $\rho_u$ using a contraction lemma similar to Theorem 4.12 in Ledoux & Talagrand (1991). However, here, the function $\rho_u$ depends on the neuron $u$, what we are taking a maximum over so that classical contraction lemmas do not apply directly. To resolve

this first obstacle, we split the neurons according to their activation function. Below, we highlight in **orange** what is important and/or the changes from one line to another. Denote $N_\rho$ the neurons that have $\rho$ as their associated activation function, and the term with a maximum over all $u \in N_\rho$ is denoted:

$$e(\rho) := \mathbb{E}_{\boldsymbol{\varepsilon}} g \left( \max_{\substack{v \in N_{\text{out}}, \\ m=1,\dots,M}} \max_{u \in (\text{ant}^d(v) \cap \boldsymbol{N_\rho}) \setminus N_{\text{in}}} \sup_{\boldsymbol{\theta}} \left| \sum_{i=1}^n \varepsilon_{i,v,m} u(\boldsymbol{\theta}, x_i) \right| \right),$$

with the convention $e(\rho) = 0$ if $N_\rho$ is empty. This yields a first bound

$$\mathbb{E}_{\boldsymbol{\varepsilon}} g \left( \max_{\substack{v \in N_{\text{out}}, \\ m=1,\dots,M}} \max_{u \in \text{ant}^d(v) \setminus N_{\text{in}}} \sup_{\boldsymbol{\theta}} \left| \sum_{i=1}^n \varepsilon_{i,v,m} u(\boldsymbol{\theta}, x_i) \right| \right) \leqslant e(\text{ReLU}) + e(\text{id}) + \sum_k e(k\text{-pool})$$

where the sum of the right-hand side is on all the $k \in \mathbb{N}_{>0}$ such that there is at least one neuron in $N_{k\text{-pool}}$. Define $E(\rho)$ to be the same thing as $e(\rho)$ but without the absolute values:

$$E(\rho) := \mathbb{E}_{\boldsymbol{\varepsilon}} g \left( \max_{\substack{v \in N_{\text{out}}, \\ m=1,\dots,M}} \max_{u \in (\text{ant}^d(v) \cap \boldsymbol{N_\rho}) \setminus N_{\text{in}}} \sup_{\boldsymbol{\theta}} \sum_{i=1}^n \varepsilon_{i,v,m} u(\boldsymbol{\theta}, x_i) \right).$$

Lemma E.1 gets rid of the absolute values by paying a factor 2:

$$e(\rho) \leqslant 2E(\rho).$$

We now want to bound each $E(\rho)$.

**Step 2: get rid of the $*$-max-pooling and ReLU activation functions.** Since the maximal kernel size is $K$, any $*$-max-pooling neuron $u$ must have at most $K$ antecedents. When a $u \in N_{*\text{-pool}}$ has less than $K$ antecedents, we artificially add neurons $w$ to $\text{ant}(u)$ to make it of cardinal $K$, and we set by convention $\boldsymbol{\theta}^{w \to u} = 0$. We also fix an arbitrary order on the antecedents of $u$ and write $\text{ant}(u)_w$ for the antecedent number $w$, with $R_{\text{ant}(u)_w}$ the function associated with this neuron. For a ReLU or $*$-max-pooling neuron $u$, define the pre-activation of $u$ to be

$$\text{pre}_u(\boldsymbol{\theta}, x) := \begin{cases} \left\langle \begin{pmatrix} \boldsymbol{\theta}^{\to u} \\ b_u \end{pmatrix}, \begin{pmatrix} R_{\text{ant}(u)}(\boldsymbol{\theta}, x) \\ 1 \end{pmatrix} \right\rangle & \text{if } u \in N_{\text{ReLU}}, \\ \left( b_u + \boldsymbol{\theta}^{\text{ant}(u)_w \to u} R_{\text{ant}(u)_w}(\boldsymbol{\theta}, x) \right)_{w=1,\dots,k} & \text{otherwise when } u \in N_{*\text{-pool}}. \end{cases}$$

where we recall that $b_u = 0$ for $u \in N_{*\text{-pool}}$ by assumption. We nevertheless keep $b_u$ in the computation until the point where this assumption is apparently actually needed to continue. Note that the pre-activation has been defined to satisfy $u(\boldsymbol{\theta}, x) = \rho_u(\text{pre}_u(\boldsymbol{\theta}, x))$. When $\rho$ is the ReLU or $k$-pool, we can thus rewrite $E(\rho)$ in terms of the pre-activations:

$$E(\rho) = \mathbb{E}_{\boldsymbol{\varepsilon}} g \left( \max_{\substack{v \in N_{\text{out}}, \\ m=1,\dots,M}} \max_{u \in (\text{ant}^d(v) \cap N_\rho) \setminus N_{\text{in}}} \sup_{\boldsymbol{\theta}} \sum_{i=1}^n \varepsilon_{i,v,m} \boldsymbol{\rho(\text{pre}_u(\boldsymbol{\theta}, x_i))} \right).$$

Consider the finite set $Z = \{(v,m), v \in N_{\text{out}}, m = 1, \dots, M\}$ and for every $z = (v,m) \in Z$, define $T^z = \{(\text{pre}_u(\boldsymbol{\theta}, x_i))_{i=1,\dots,n} : u \in (\text{ant}^d(v) \cap N_\rho) \setminus N_{\text{in}}, \boldsymbol{\theta} \in \boldsymbol{\Theta}\}$. An element of $T^z$ will be denoted $t = (t_i)_{i=1}^n \in T^z \subset \mathbb{R}^n$ if $\rho = \text{ReLU}$, and $t = (t_i)_{i=1}^n \in T^z \subset (\mathbb{R}^k)^n$ with $t_i = (t_{i,w})_{w=1}^k \in \mathbb{R}^k$ if $u \in N_{*\text{-pool}}$. We can again rewrite $E(\rho)$ as

$$E(\rho) = \mathbb{E}_{\boldsymbol{\varepsilon}} g \left( \max_{\boldsymbol{z \in Z}} \sup_{\boldsymbol{t \in T^z}} \sum_{i=1}^n \varepsilon_{i,\boldsymbol{z}} \rho(t_i) \right).$$

We now want to get rid of the activation function $\rho$ with a contraction lemma. There is a second difficulty that prevents us from directly applying classical contraction lemmas such as Theorem 4.12 of Ledoux & Talagrand (1991). It is the presence of a maximum over multiple copies indexed by $z \in Z$ of a supremum that depends on iid families $(\varepsilon_{i,z})_{i=1\dots n}$. Indeed, Theorem 4.12 of Ledoux & Talagrand (1991) only deals with a single copy ($|Z| = 1$). This motivates the contraction lemma established for the occasion in Lemma D.1. Once the activation functions removed, we can conclude separately for $\rho = \text{ReLU}, \text{id}$ and $\rho = k\text{-pool}$.

**Step 3a: deal with $\rho = k\text{-pool}$ via rescaling.** In the case $\rho = k\text{-pool}$, Lemma D.1 shows that

$$\mathbb{E}_{\boldsymbol{\varepsilon}} g \left( \max_{z \in Z} \sup_{t \in T^z} \sum_{i=1}^{n} \varepsilon_{i,z} \boldsymbol{k}\text{-pool}(t_i) \right)$$

$$\leqslant \mathbb{E}_{\boldsymbol{\varepsilon}} g \left( \max_{z \in Z} \sup_{t \in T^z} \sum_{\substack{i=1,\ldots,n, \\ w=1,\ldots,K}} \varepsilon_{i,z,w} t_{i,w} \right).$$

The right-hand side is equal to

$$\mathbb{E}_{\boldsymbol{\varepsilon}} g \left( \max_{\substack{v \in N_{\text{out}}, \\ m=1,\ldots,M}} \sup_{\substack{u \in (\text{ant}^d(v) \cap N_{*\text{-pool}}) \setminus N_{\text{in}}, \\ \boldsymbol{\theta} \in \Theta}} \sum_{\substack{i=1,\ldots,n, \\ w=1,\ldots,K}} \varepsilon_{i,v,m,w} (b_u + \boldsymbol{\theta}^{\text{ant}(u)_w \to u} R_{\text{ant}(u)_w}(\boldsymbol{\theta}, x_i)) \right).$$
$$(20)$$

We now deal with this using the assumption on the norm of incoming weights. Recalling that $b_u = 0$ for every $u \in N_{*\text{-pool}}$:

$$\sum_{\substack{i=1,\ldots,n, \\ w=1,\ldots,K}} \varepsilon_{i,v,m,w} \big( \underbrace{b_u}_{=0} + \boldsymbol{\theta}^{\text{ant}(u)_w \to u} R_{\text{ant}(u)_w}(\boldsymbol{\theta}, x_i) \big)$$

$$= \sum_{w=1,\ldots,K} \boldsymbol{\theta}^{\text{ant}(u)_w \to u} \left( \sum_{i=1,\ldots,n} \varepsilon_{i,v,m,w} R_{\text{ant}(u)_w}(\boldsymbol{\theta}, x_i) \right)$$

$$\underset{\text{Hölder}}{\leqslant} \underbrace{\|\boldsymbol{\theta}^{\to u}\|_1}_{\leqslant 1 \text{ by assumption}} \max_{w=1,\ldots,K} \left| \sum_{i=1,\ldots,n} \varepsilon_{i,v,m,w} R_{\text{ant}(u)_w}(\boldsymbol{\theta}, x_i) \right|$$

$$\underset{\substack{\text{decoupling } w \text{ and ant}(u)_w}}{\leqslant} \max_{\boldsymbol{w} \in \text{ant}(u)} \max_{\boldsymbol{w'}=1,\ldots,K} \left| \sum_{i=1}^{n} \varepsilon_{i,v,m,\boldsymbol{w'}} \boldsymbol{w}(\boldsymbol{\theta}, x_i) \right|.$$

Note for the curious reader that this inequality is the current obstacle when the biases are nonzero since we would end up with $K|b_u| + \|\boldsymbol{\theta}^{\to u}\|_1$ and we could not anymore use that this is $\leqslant 1$.

We deduce that Equation (20) is bounded from above by

$$\mathbb{E}_{\boldsymbol{\varepsilon}} g \left( \max_{\substack{v \in N_{\text{out}}, \\ m=1,\ldots,M}} \sup_{u \in (\text{ant}^d(v) \cap N_{*\text{-pool}}) \setminus N_{\text{in}}, \boldsymbol{\theta} \in \Theta} \max_{w \in \text{ant}(u)} \max_{w'=1,\ldots,K} \left| \sum_{i=1}^{n} \varepsilon_{i,v,m,w'} w(\boldsymbol{\theta}, x_i) \right| \right).$$

Instead of having $\varepsilon_{i,v,m,w'}$ with $m = 1, \ldots, M$ and $w' = 1, \ldots, K$, we re-index it as $\varepsilon_{i,v,m}$ with $m = 1, \ldots, KM$. Note also that $u \in (\text{ant}^d(v) \cap N_{*\text{-pool}}) \setminus N_{\text{in}}$ and $w \in \text{ant}(u)$ implies $w \in \text{ant}^{d+1}(v)$, so considering a maximum over $w \in \text{ant}^{d+1}(v)$ can only yield something larger. Moreover, we can add a new neuron $v_{\text{bias}}$ that computes the constant function equal to one ($v_{\text{bias}}(\boldsymbol{\theta}, x) = 1$) and add $v_{\text{bias}}$ to the maximum over $w$. Implementing all these changes, Equation (20) is bounded by

$$H := \mathbb{E}_{\boldsymbol{\varepsilon}} g \left( \max_{\substack{v \in N_{\text{out}}, \\ m=1,\ldots,KM}} \sup_{\boldsymbol{w} \in \text{ant}^{d+1}(\boldsymbol{v}) \cup \{\boldsymbol{v}_{\text{bias}}\}} \sup_{\boldsymbol{\theta} \in \Theta} \left| \sum_{i=1}^{n} \varepsilon_{i,v,\boldsymbol{m}} w(\boldsymbol{\theta}, x_i) \right| \right).$$

We now derive similar inequalities when $\rho = \text{id}$ and $\rho = \text{ReLU}$.

**Step 3b: deal with $\rho = \mathrm{id}, \mathrm{ReLU}$ via rescaling.** In the case $\rho = \mathrm{ReLU}$, Lemma D.1 shows that

$$\mathbb{E}_{\boldsymbol{\varepsilon}} g \left( \max_{z \in Z} \sup_{t \in T^z} \sum_{i=1}^{n} \varepsilon_{i,z} \mathbf{ReLU}(t_i) \right)$$

$$\leqslant \mathbb{E}_{\boldsymbol{\varepsilon}} g \left( \max_{z \in Z} \sup_{t \in T^z} \sum_{i=1,\ldots,n} \varepsilon_{i,z} t_i \right).$$

The difference with the $*$-max-pooling case is that each $t_i$ is scalar so this does not introduce an additional index $w$ to the Rademacher variables. The right-hand side can be rewritten as

$$\mathbb{E}_{\boldsymbol{\varepsilon}} g \left( \max_{\substack{v \in N_{\mathrm{out}}, \\ m=1,\ldots,M}} \sup_{u \in (\mathrm{ant}^d(v) \cap \mathbf{N_{ReLU}}) \backslash N_{\mathrm{in}}, \boldsymbol{\theta} \in \Theta} \sum_{i=1,\ldots,n} \varepsilon_{i,v,m} \left\langle \begin{pmatrix} \boldsymbol{\theta}^{\to u} \\ b_u \end{pmatrix}, \begin{pmatrix} R_{\mathrm{ant}(u)}(\boldsymbol{\theta}, x_i) \\ 1 \end{pmatrix} \right\rangle \right)$$

We can only increase the latter by considering a maximum over all $u \in \mathrm{ant}^d(v)$, not only the ones in $N_{\mathrm{ReLU}}$. We also add absolutes values. This is then bounded by

$$F := \mathbb{E}_{\boldsymbol{\varepsilon}} g \left( \max_{\substack{v \in N_{\mathrm{out}}, \\ m=1,\ldots,M}} \sup_{u \in \mathrm{ant}^d(v) \backslash N_{\mathrm{in}}, \boldsymbol{\theta} \in \Theta} \left| \sum_{i=1,\ldots,n} \varepsilon_{i,v,m} \left\langle \begin{pmatrix} \boldsymbol{\theta}^{\to u} \\ b_u \end{pmatrix}, \begin{pmatrix} R_{\mathrm{ant}(u)}(\boldsymbol{\theta}, x_i) \\ 1 \end{pmatrix} \right\rangle \right| \right). \tag{21}$$

This means that $E(\mathrm{ReLU}) \leqslant F$. Let us also observe that $e(\mathrm{id}) \leqslant F$. Indeed, recall that by definition

$$e(\mathrm{id}) = \mathbb{E}_{\boldsymbol{\varepsilon}} g \left( \max_{\substack{v \in N_{\mathrm{out}}, \\ m=1,\ldots,M}} \max_{u \in (\mathrm{ant}^d(v) \cap \mathbf{N_{id}}) \backslash N_{\mathrm{in}}} \sup_{\boldsymbol{\theta}} \left| \sum_{i=1}^{n} \varepsilon_{i,v,m} u(\boldsymbol{\theta}, x_i) \right| \right).$$

We can only increase the latter by considering a maximum over all $u \in \mathrm{ant}^d(v)$. Moreover, for an identity neuron $u$, it holds $u(\boldsymbol{\theta}, x) = \left\langle \begin{pmatrix} \boldsymbol{\theta}^{\to u} \\ b_u \end{pmatrix}, \begin{pmatrix} R_{\mathrm{ant}(u)}(\boldsymbol{\theta}, x) \\ 1 \end{pmatrix} \right\rangle$. This shows that $e(\mathrm{id}) \leqslant F$. It remains to bound $F$ using that the assumption on the norm of the parameters. Introduce a new neuron $v_{\mathtt{bias}}$ that computes the constant function equal to one: $v_{\mathtt{bias}}(\boldsymbol{\theta}, x) = 1$. Note that

$$\sum_{i=1,\ldots,n} \varepsilon_{i,v,m} \left\langle \begin{pmatrix} \boldsymbol{\theta}^{\to u} \\ b_u \end{pmatrix}, \begin{pmatrix} R_{\mathrm{ant}(u)}(\boldsymbol{\theta}, x_i) \\ 1 \end{pmatrix} \right\rangle$$

$$= \left\langle \begin{pmatrix} \boldsymbol{\theta}^{\to u} \\ b_u \end{pmatrix}, \sum_{i=1,\ldots,n} \varepsilon_{i,v,m} \begin{pmatrix} R_{\mathrm{ant}(u)}(\boldsymbol{\theta}, x_i) \\ 1 \end{pmatrix} \right\rangle$$

$$\underset{\mathrm{H\ddot{o}lder}}{\leqslant} \underbrace{(\|\boldsymbol{\theta}^{\to u}\|_1 + |b_u|)}_{\leqslant 1 \text{ by assumption}} \max_{w \in \mathrm{ant}(u) \cup \{v_{\mathtt{bias}}\}} \left| \sum_{i=1}^{n} \varepsilon_{i,v,m} w(\boldsymbol{\theta}, x_i) \right|.$$

This shows that

$$F \leqslant \mathbb{E}_{\boldsymbol{\varepsilon}} g \left( \max_{\substack{v \in N_{\mathrm{out}}, \\ m=1,\ldots,M}} \sup_{u \in \mathrm{ant}^d(v) \backslash N_{\mathrm{in}}, \boldsymbol{\theta} \in \Theta} \max_{w \in \mathrm{ant}(u) \cup \{v_{\mathtt{bias}}\}} \left| \sum_{i=1}^{n} \varepsilon_{i,v,m} w(\boldsymbol{\theta}, x_i) \right| \right).$$

Obviously, introducing additional copies of $\boldsymbol{\varepsilon}$ to make the third index going from $m = 1$ to $KM$ can only make it larger. Moreover, $u \in \mathrm{ant}^d(v) \backslash N_{\mathrm{in}}$ and $w \in \mathrm{ant}(u)$ implies $w \in \mathrm{ant}^{d+1}(u)$, so we can instead consider a maximum over $w \in \mathrm{ant}^{d+1}(v)$. This gives the upper-bound

$$F \leqslant \mathbb{E}_{\boldsymbol{\varepsilon}} g \left( \max_{\substack{v \in N_{\mathrm{out}}, \\ m=1,\ldots,KM}} \max_{\mathbf{w \in ant^{d+1}(v) \cup \{v_{\mathtt{bias}}\}}} \sup_{\boldsymbol{\theta} \in \Theta} \left| \sum_{i=1}^{n} \varepsilon_{i,v,m} w(\boldsymbol{\theta}, x_i) \right| \right)$$

$$= H.$$

**Step 4: putting everything together.** At the end, recalling that there are at most $P$ different $k \in \mathbb{N}_{>0}$ associated with an existing $k$-max-pooling neuron, we get the final bound

$$
\mathbb{E}_{\boldsymbol{\varepsilon}} g \left( \max_{\substack{v \in N_{\text{out}}, \\ m=1,\dots,M}} \max_{u \in \text{ant}^d(v) \backslash N_{\text{in}}} \sup_{\boldsymbol{\theta}} \left| \sum_{i=1}^n \boldsymbol{\varepsilon}_{i,v,m} u(\boldsymbol{\theta}, x_i) \right| \right)
$$

$$
\leqslant e(\text{id}) + e(\text{ReLU}) + \sum_k e(k\text{-pool})
$$

$$
\leqslant e(\text{id}) + 2E(\text{ReLU}) + 2\sum_k E(k\text{-pool})
$$

$$
\leqslant F + 2F + 2\sum_k E(k\text{-pool})
$$

$$
\leqslant H + 2H + 2\sum_k H
$$

$$
\leqslant H + 2H + 2PH = (3 + 2P)H.
$$

The term $(3+2P)H$ can again be bounded by splitting the maximum over $w \in \text{ant}^{d+1}(v) \cup \{v_{\text{bias}}\}$ between the $w$'s that are input neurons, and those that are not, since everything is non-negative. This yields the claim. $\qquad\square$

**Remark E.1** (Improved dependencies on the kernel size). *Note that in the proof of Lemma E.3, the multiplication of $M$ by $K$ can be avoided if there are no $*$-max-pooling neurons in $\text{ant}^d(v)$. Because of skip connections, even if there is a single $*$-max-pooling neuron in the architecture, it can be in $\text{ant}^d(v)$ for many $d$'s. A more advanced version of the argument is to peel only the ReLU and identity neurons, by leaving the $*$-max-pooling neurons as they are, until we reach a set of $*$-max-pooling neurons large enough that we decide to peel simultaneously. This would prevent the multiplication by $K$ every time $d$ is increased.*

We can now state the main peeling theorem, which directly result from Lemma E.2 and Lemma E.3 by induction on $d$. Note that these lemmas contain assumptions on the size of the incoming weights of the different neurons. These assumptions are met using Algorithm 1, which rescales the parameters without changing the associated function nor the path-norm.

**Theorem E.1.** *Consider a neural network architecture as in Definition 2.2 with $N_{\text{out}} \cap N_{\text{in}} = \emptyset$. Assume that $b_v = 0$ for every $v \in N_{*\text{-pool}}$. Define $P := |\{k \in \mathbb{N}_{>0}, \exists u \in N_{k\text{-pool}}\}|$ the number of different types of $*$-max-pooling neurons in $G$, and $K := \max_{u \in N_{*\text{-pool}}}$ the maximum kernel size ($K := 1$ by convention if $P = 0$). Denote by $v_{bias}$ a new input neuron and define $x_{v_{bias}} = 1$ for any input $x$. For any set of parameters $\boldsymbol{\Theta}$ associated with the network, such that $\|\Phi(\boldsymbol{\theta})\|_1 \leqslant r$ for every $\boldsymbol{\theta} \in \boldsymbol{\Theta}$, it holds for every convex non-decreasing function $g : \mathbb{R} \to \mathbb{R}_{\geqslant 0}$*

$$
\mathbb{E}_{\boldsymbol{\varepsilon}} g \left( \sup_{\boldsymbol{\theta} \in \boldsymbol{\Theta}} \sum_{\substack{i=1,\dots,n, \\ v \in N_{out}}} \boldsymbol{\varepsilon}_{i,v} v(\boldsymbol{\theta}, x_i) \right)
$$

$$
\leqslant \frac{(3 + 2P)^D}{2 + 2P} \mathbb{E}_{\boldsymbol{\varepsilon}} g \left( r \max_{\substack{v \in N_{out}, \\ m=1,\dots,K^{D-1}}} \max_{u \in N_{in} \cup \{v_{bias}\}} \left| \sum_{i=1}^n \boldsymbol{\varepsilon}_{i,v,m} x_{i,u} \right| \right).
$$

*Proof of Theorem E.1.* Without loss of generality, we can replace $\boldsymbol{\Theta}$ by its image under Algorithm 1 with $q = 1$, as Algorithm 1 does not change the associated function $R_{\boldsymbol{\theta}}$ nor the path-norm $\|\Phi(\boldsymbol{\theta})\|_1$ (Lemma B.1) so that we still have $\|\Phi(\boldsymbol{\theta})\|_1 \leqslant r$ and the supremum over $\boldsymbol{\theta} \in \boldsymbol{\Theta}$ on the left-hand side can be taken over rescaled parameters. By Lemma B.1, the parameters are 1-normalized (Definition B.1), so we will be able to use Lemma E.2 and Lemma E.3. Indeed, 1-normalized parameters $\boldsymbol{\theta}$ satisfy $\sum_{v \in N_{\text{out}}} \|\boldsymbol{\theta}^{\to v}\|_1 + |b_v| = \|\Phi(\boldsymbol{\theta})\|_1 \leqslant r$ so Lemma E.2 applies. We also have $\|\boldsymbol{\theta}^{\to v}\|_1 + |b_v| \leqslant 1$ for every $v \notin N_{\text{in}} \cup N_{\text{out}}$, by definition of 1-normalization, so Lemma E.3 also applies.

By induction on $d \geqslant 1$, we prove that (highlighting in orange what is important)

$$
\mathbb{E}_{\boldsymbol{\varepsilon}} g \left( \sup_{\boldsymbol{\theta} \in \Theta} \sum_{\substack{i=1,\ldots,n, \\ v \in N_{\mathrm{out}}}} \varepsilon_{i,v} v(\boldsymbol{\theta}, x_i) \right)
$$

$$
\leqslant \sum_{\ell=1}^{d} (3+2P)^{\ell-1} \mathbb{E}_{\boldsymbol{\varepsilon}} g \left( r \max_{\substack{v \in N_{\mathrm{out}}, \\ m=1,\ldots,K^{\ell-1}}} \max_{u \in (\mathrm{ant}^{\ell}(v) \cap \textcolor{orange}{N_{\mathrm{in}}}) \cup \{v_{\mathrm{bias}}\}} \left| \sum_{i=1}^{n} \varepsilon_{i,v,m} \textcolor{orange}{\boldsymbol{x_{i,u}}} \right| \right)
$$

$$
+ (3+2P)^{d-1} \mathbb{E}_{\boldsymbol{\varepsilon}} g \left( r \max_{\substack{v \in N_{\mathrm{out}}, \\ m=1,\ldots,K^{d-1}}} \max_{u \in \mathrm{ant}^{d}(v) \setminus \textcolor{orange}{N_{\mathrm{in}}}} \sup_{\boldsymbol{\theta} \in \Theta} \left| \sum_{i=1}^{n} \varepsilon_{i,v,m} \textcolor{orange}{\boldsymbol{u}(\boldsymbol{\theta}, \boldsymbol{x_i})} \right| \right) ,
$$

with the same convention as in Lemma E.2 for maxima over empty sets. This is true for $d = 1$ by Lemma E.2. The induction step is verified using Lemma E.3. This concludes the induction. Applying the result for $d = D$, and since $\mathrm{ant}^D(v) \setminus N_{\mathrm{in}} = \emptyset$, we get:

$$
\mathbb{E}_{\boldsymbol{\varepsilon}} g \left( \sup_{\boldsymbol{\theta} \in \Theta} \sum_{\substack{i=1,\ldots,n, \\ v \in N_{\mathrm{out}}}} \varepsilon_{i,v} v(\boldsymbol{\theta}, x_i) \right)
$$

$$
\leqslant \sum_{d=1}^{D} (3+2P)^{d-1} \mathbb{E}_{\boldsymbol{\varepsilon}} g \left( r \max_{\substack{v \in N_{\mathrm{out}}, \\ m=1,\ldots,K^{d-1}}} \max_{u \in (\mathrm{ant}^{d}(v) \cap N_{\mathrm{in}}) \cup \{v_{\mathrm{bias}}\}} \left| \sum_{i=1}^{n} \varepsilon_{i,v,m} x_{i,u} \right| \right) .
$$

We can only increase the right-hand side by considering maximum over all $u \in N_{\mathrm{in}} \cup \{v_{\mathrm{bias}}\}$ and by adding independent copies indexed from $m = 1$ to $m = K^{D-1}$. Moreover, $\sum_{d=1}^{D} (3+2P)^{d-1} = ((3+2P)^D - 1)/(2 + 2P)$. This shows the final bound:

$$
\mathbb{E}_{\boldsymbol{\varepsilon}} g \left( \sup_{\boldsymbol{\theta} \in \Theta} \sum_{\substack{i=1,\ldots,n, \\ v \in N_{\mathrm{out}}}} \varepsilon_{i,v} v(\boldsymbol{\theta}, x_i) \right)
$$

$$
\leqslant \frac{(3+2P)^D}{2+2P} \mathbb{E}_{\boldsymbol{\varepsilon}} g \left( r \max_{\substack{v \in N_{\mathrm{out}}, \\ m=1,\ldots,K^{D-1}}} \max_{u \in N_{\mathrm{in}} \cup \{v_{\mathrm{bias}}\}} \left| \sum_{i=1}^{n} \varepsilon_{i,v,m} x_{i,u} \right| \right) .
$$

$\square$

## F  DETAILS TO DERIVE THE GENERALIZATION BOUND (THEOREM 3.1)

*Proof of Theorem 3.1.* The proof is given directly in the general case with nonzero biases on every $v \notin N_{\text{*-pool}}$ and uses an additional input neuron $v_{\mathrm{bias}}$. We highlight along the proof where improved results can be obtained assuming zero biases, yielding Theorem 3.1 as a consequence. Define the random matrices $E = (\varepsilon_{i,v})_{i,v} \in \mathbb{R}^{n \times d_{\mathrm{out}}}$ and $R(\boldsymbol{\theta}, \mathbf{X}) = (v(\boldsymbol{\theta}, \mathbf{X}_i))_{i,v} \in \mathbb{R}^{n \times d_{\mathrm{out}}}$ so that $\langle E, R(\boldsymbol{\theta}, \mathbf{X}) \rangle = \sum_{i,v} \varepsilon_{i,v} (R_{\boldsymbol{\theta}}(\mathbf{X}_i))_v$. It holds:

$$
\mathbb{E}_{\mathbf{Z}} \, \ell\text{-generalization error of } \hat{\boldsymbol{\theta}}(\mathbf{Z}) \leqslant \frac{2}{n} \mathbb{E}_{\mathbf{Z},\boldsymbol{\varepsilon}} \left( \sup_{\boldsymbol{\theta}} \sum_{i=1}^{n} \varepsilon_i \ell \left( R_{\boldsymbol{\theta}}(\mathbf{X}_i), \mathbf{Y}_i \right) \right)
$$

$$
\leqslant \frac{2\sqrt{2}L}{n} \mathbb{E}_{\mathbf{Z},\boldsymbol{\varepsilon}} \left( \sup_{\boldsymbol{\theta}} \langle E, R(\boldsymbol{\theta}, \mathbf{X}) \rangle \right) .
$$

The first inequality is the symmetrization property given by Shalev-Shwartz & Ben-David (2014, Theorem 26.3), and the second inequality is the vector-valued contraction property given by Maurer (2016). These are the relevant versions of very classical arguments that are widely used to

reduce the problem to the Rademacher complexity of the model (Bach, 2024, Propositions 4.2 and 4.3)(Wainwright, 2019, Equations (4.17) and (4.18))(Bartlett & Mendelson, 2002, Proof of Theorem 8)(Shalev-Shwartz & Ben-David, 2014, Theorem 26.3)(Ledoux & Talagrand, 1991, Equation (4.20)). In particular, this step has nothing specific with neural networks. Note that the assumption on the loss is used for the second inequality.

We now condition on $\mathbf{Z} = (\mathbf{X}, \mathbf{Y})$ and denote $\mathbb{E}_{\boldsymbol{\varepsilon}}$ the conditional expectation. For any random variable $\lambda(\mathbf{Z}) > 0$ measurable in $\mathbf{Z}$, it holds

$$
\begin{aligned}
\mathbb{E}_{\boldsymbol{\varepsilon}} \left( \sup_{\boldsymbol{\theta}} \langle E, R(\boldsymbol{\theta}, \mathbf{X}) \rangle \right) =& \frac{1}{\lambda(\mathbf{Z})} \log \exp \left( \lambda(\mathbf{Z}) \mathbb{E}_{\boldsymbol{\varepsilon}} \left( \sup_{\boldsymbol{\theta}} \langle E, R(\boldsymbol{\theta}, \mathbf{X}) \rangle \right) \right) \\
\underset{\lambda \text{ measurable in } \mathbf{Z}}{=}& \frac{1}{\lambda(\mathbf{Z})} \log \exp \left( \mathbb{E}_{\boldsymbol{\varepsilon}} \left( \lambda(\mathbf{Z}) \sup_{\boldsymbol{\theta}} \langle E, R(\boldsymbol{\theta}, \mathbf{X}) \rangle \right) \right) \\
\underset{\text{Jensen}}{\leqslant}& \frac{1}{\lambda(\mathbf{Z})} \log \mathbb{E}_{\boldsymbol{\varepsilon}} \exp \left( \lambda(\mathbf{Z}) \sup_{\boldsymbol{\theta}} \langle E, R(\boldsymbol{\theta}, \mathbf{X}) \rangle \right).
\end{aligned}
$$

For $z = ((x_i, y_i))_{i=1}^n \in (\mathbb{R}^{d_{\text{in}}} \times \mathbb{R}^{d_{\text{out}}})^n$, denote

$$
e(z) = \mathbb{E}_{\boldsymbol{\varepsilon}} \exp \left( \lambda(z) \sup_{\boldsymbol{\theta}} \langle E, R(\boldsymbol{\theta}, x) \rangle \right).
$$

Since $\mathbf{Z}$ is independent of $\varepsilon$, it holds

$$
\mathbb{E}_{\boldsymbol{\varepsilon}} \exp \left( \lambda(\mathbf{Z}) \sup_{\boldsymbol{\theta}} \langle E, R(\boldsymbol{\theta}, \mathbf{X}) \rangle \right) = e(\mathbf{Z}).
$$

Denote $r = \sup_{\boldsymbol{\theta} \in \boldsymbol{\Theta}} \|\Phi(\boldsymbol{\theta})\|_1$. For $z$ as above, simply denote $\lambda := \lambda(z)$. Since $N_{\text{in}} \cap N_{\text{out}} = \emptyset$ and the biases of $*$-max-pooling neurons are null, the peeling argument given by Theorem E.1 for $g : t \in \mathbb{R} \mapsto \exp(\lambda t)$ guarantees:

$$
e(z) \leqslant \frac{(3 + 2P)^D}{2 + 2P} \mathbb{E}_{\boldsymbol{\varepsilon}} \exp \left( \lambda r \max_{\substack{v \in N_{\text{out}}, \\ m=1,\ldots,K^{D-1}}} \max_{u \in N_{\text{in}} \cup \{v_{\text{bias}}\}} \left| \sum_{i=1}^n \boldsymbol{\varepsilon}_{i,v,m} x_{i,u} \right| \right),
$$

where $x_{i,u}$ is coordinate $u$ of vector $x_i \in \mathbb{R}^{d_{\text{in}}}$, and where $v_{\text{bias}}$ is an added neuron for which we set by convention $x_{v_{\text{bias}}} = 1$ for any input $x$. It is easy to check that the same bound holds true with a maximum only over $u \in N_{\text{in}}$ (not considering $v_{\text{bias}}$ in the maximum) when all biases are constrained to be null. In such a setting, all the $\max_{u \in N_{\text{in}} \cup \{v_{\text{bias}}\}}$ below can be replaced by $\max_{u \in N_{\text{in}}}$. Denote

$$
\sigma(x) := \max_{u \in N_{\text{in}} \cup \{v_{\text{bias}}\}} \left( \sum_{i=1}^n x_{i,u}^2 \right)^{1/2} \geqslant \sqrt{n}.
$$

Using Lemma F.1, it holds

$$
\mathbb{E}_{\boldsymbol{\varepsilon}} \exp \left( \lambda r \max_{\substack{v \in N_{\text{out}}, \\ u \in N_{\text{in}} \cup \{v_{\text{bias}}\}, \\ m=1,\ldots,K^{D-1}}} \left| \sum_{i=1}^n \boldsymbol{\varepsilon}_{i,v,m} (\mathbf{X}_i)_u \right| \right) \leqslant 2 K^{D-1} (d_{\text{in}} + 1) d_{\text{out}} \exp \left( \frac{(r \lambda(z) \sigma(x))^2}{2} \right).
$$

When biases are constrained to be null, $d_{\text{in}} + 1$ is replaced by $d_{\text{in}}$. Putting everything together, we get:

$$
\mathbb{E}_{\boldsymbol{\varepsilon}} \left( \sup_{\boldsymbol{\theta}} \langle E, R(\boldsymbol{\theta}, \mathbf{X}) \rangle \right) = e(\mathbf{Z}) \leqslant \left( \frac{1}{\lambda} \log(C_1) + \lambda(\mathbf{Z}) C_2(\mathbf{X}) \right)
$$

with

$$
C_1 = 2 K^{D-1} (d_{\text{in}} + 1) d_{\text{out}} \times \frac{(3 + 2P)^D}{2 + 2P} = \frac{3 + 2P}{1 + P} ((3 + 2P) K)^{D-1} (d_{\text{in}} + 1) d_{\text{out}}
$$

(again with $d_{\text{in}} + 1$ replaced by $d_{\text{in}}$ when all biases are null) and

$$C_2(\mathbf{X}) = \frac{1}{2}(r\sigma(\mathbf{X}))^2.$$

Choosing $\lambda(\mathbf{Z}) = \sqrt{\frac{\log(C_1)}{C_2(\mathbf{X})}}$ yields:

$$\mathbb{E}_{\boldsymbol{\varepsilon}}\left(\sup_{\boldsymbol{\theta}}\langle E, R(\boldsymbol{\theta}, \mathbf{X})\rangle\right) \leqslant 2\sqrt{\log(C_1)C_2(\mathbf{X})}$$

$$\leqslant \underbrace{\sqrt{2}\sigma(\mathbf{X})r}_{=2\sqrt{C_2(\mathbf{X})}}\underbrace{\left(\log\left(\frac{3+2P}{1+P}(d_{\text{in}}+1)d_{\text{out}}\right) + D\log\left((3+2P)K\right)\right)^{1/2}}_{\geqslant\sqrt{\log(C_1)}}$$

with $d_{\text{in}} + 1$ replaced by $d_{\text{in}}$ when all biases are null. Taking the expectation on both sides over $\mathbf{Z}$, and multiplying this by $\frac{2\sqrt{2}L}{n}$ yields Theorem 3.1. $\qquad\square$

The next lemma is classical (Golowich et al., 2018, Section 7.1) and is here only for completeness.

**Lemma F.1.** *For any $d, k \in \mathbb{N}_{>0}$ and $\lambda > 0$, it holds*

$$\mathbb{E}_{\boldsymbol{\varepsilon}}\exp\left(\lambda\max_{\substack{m=1,\dots,k,\\u=1,\dots,d}}\left|\sum_{i=1}^{n}\varepsilon_{i,m}(\mathbf{X}_i)_u\right|\right) \leqslant 2kd\max_{u=1,\dots,d}\exp\left(\frac{\lambda^2}{2}\sum_{i=1}^{n}(\mathbf{X}_i)_u^2\right).$$

*Proof.* It holds

$$\mathbb{E}_{\boldsymbol{\varepsilon}}\exp\left(\lambda\max_{\substack{m=1,\dots,k,\\u=1,\dots,d}}\left|\sum_{i=1}^{n}\varepsilon_{i,m}(\mathbf{X}_i)_u\right|\right) \leqslant \sum_{\substack{m=1,\dots,k,\\u=1,\dots,d}}\mathbb{E}_{\boldsymbol{\varepsilon}}\exp\left(\lambda\left|\sum_{i=1}^{n}\varepsilon_{i,m}(\mathbf{X}_i)_u\right|\right).$$

For given $u$ and $m$:

$$\mathbb{E}_{\boldsymbol{\varepsilon}}\exp\left(\lambda\left|\sum_{i=1}^{n}\varepsilon_{i,m}(\mathbf{X}_i)_u\right|\right) \underset{\text{Lemma E.1}}{\leqslant} 2\mathbb{E}_{\boldsymbol{\varepsilon}}\exp\left(\lambda\sum_{i=1}^{n}\varepsilon_{i,m}(\mathbf{X}_i)_u\right)$$

$$= 2\prod_{i=1}^{n}\frac{\exp\left(\lambda(\mathbf{X}_i)_u\right) + \exp\left(-\lambda(\mathbf{X}_i)_u\right)}{2} \leqslant 2\exp\left(\frac{\lambda^2}{2}\sum_{i=1}^{n}(\mathbf{X}_i)_u^2\right)$$

using $\exp(x) + \exp(-x) \leqslant 2\exp(x^2/2)$ in the last inequality. $\qquad\square$

## G  THE CROSS-ENTROPY LOSS IS LIPSCHITZ

Theorem 3.1 *applies to the cross-entropy loss with $L = \sqrt{2}$.* To see this, first recall that with $C$ classes, the cross-entropy loss is defined as

$$\ell : (x, y) \in \mathbb{R}^C \times \{0,1\}^C \mapsto -\sum_{c=1}^{d_{\text{out}}} y_c \log\left(\frac{\exp(x_c)}{\sum_d \exp(x_d)}\right).$$

Consider $y \in \{0,1\}^C$ with exactly one nonzero coordinate and an exponent $p \in [1, \infty]$ with conjugate exponent $p'$ ($1/p + 1/p' = 1$). For every $x, x' \in \mathbb{R}^C$:

$$\ell(x, y) - \ell(x', y) \leqslant 2^{1/p'}\|x - x'\|_p.$$

Consider a class $c \in \{1, \dots, C\}$ and take $y \in \{0,1\}^C$ to be a one-hot encoding of $c$ (meaning that $y_{c'} = \mathbb{1}_{c'=c}$). Consider an exponent $p \in [1, \infty]$ with conjugate exponent $p'$ ($1/p + 1/p' =$

1). The function $f : x \mapsto \ell(x, y) = -\sum_c y_c \log\left(\frac{\exp(x_c)}{\sum_{c'=1}^{C} \exp(x_{c'})}\right) = -\log\left(\frac{\exp(x_c)}{\sum_{c'=1}^{C} \exp(x_{c'})}\right)$ is continuously differentiable so that for every $x, x' \in \mathbb{R}^C$:

$$f(x) - f(x') = \int_0^1 \langle \nabla f(tx + (1-t)x'), x - x' \rangle \, dt \leqslant \sup_{t \in [0,1]} \|\nabla f(tx + (1-t)x')\|_p \|x - x'\|_{p'}.$$

In order to differentiate $f$, let's start to differentiate $g(x) = \frac{\exp(x_c)}{\sum_{c'=1}^{C} \exp(x_{c'})}$. Denote $\partial_i$ the partial derivative with respect to coordinate $i$. For $i \neq c$:

$$\partial_c g(x) = \frac{\exp(x_c)\left(\sum_{c'} \exp(x_{c'})\right) - \exp(x_c)\left(\exp(x_c)\right)}{\left(\sum_{c'} \exp(x_{c'})\right)^2}$$

$$= g(x) \frac{\sum_{c' \neq c} \exp(x_{c'})}{\sum_{c'} \exp(x_{c'})}.$$

$$\partial_i g(x) = \frac{0\left(\sum_{c'} \exp(x_{c'})\right) - \exp(x_c)\left(\exp(x_i)\right)}{\left(\sum_{c'} \exp(x_{c'})\right)^2}$$

$$= g(x) \frac{-\exp(x_i)}{\sum_{c'} \exp(x_{c'})}.$$

Since $f(x) = (-\log \circ h)(x)$:

$$\partial_i f(x) = -\frac{\partial_i g(x)}{g(x)}$$

$$= \frac{1}{\sum_{c'=1}^{C} \exp(x_{c'})} \times \begin{cases} -\sum_{c' \neq c} e^{x_{c'}} & \text{if } i = c, \\ e^{x_i} & \text{otherwise.} \end{cases}$$

Thus

$$\|\nabla f(x)\|_p^p = \sum_{i=1}^{C} |\partial_i f(x)|^p$$

$$= \frac{\left(\sum_{c' \neq c} \exp(x_{c'})\right)^p + \sum_{c' \neq c} \exp(x_{c'})^p}{\left(\sum_{c'=1}^{C} \exp(x_{c'})\right)^p}$$

$$\leqslant 2 \frac{\left(\sum_{c' \neq c} \exp(x_{c'})\right)^p}{\left(\sum_{c'=1}^{C} \exp(x_{c'})\right)^p}$$

$$\leqslant 2 \frac{\left(\sum_{c' \neq c} \exp(x_{c'})\right)^p}{\left(\sum_{c' \neq c} \exp(x_{c'})\right)^p}$$

$$= 2.$$

where we used in the first inequality that $\|v\|_p^p \leqslant \|v\|_1^p$ for any vector $v$. This shows that for every $x, x' \in \mathbb{R}^C$:

$$\ell(x, y) - \ell(x', y) \leqslant 2^{1/p} \|x - x'\|_{p'}.$$

## H  THE TOP-1 ACCURACY LOSS IS NOT LIPSCHITZ

Theorem 3.1 *does not apply to the top-1 accuracy loss* $\ell(\hat{y}, y) = \mathbb{1}_{\arg\max \hat{y} = \arg\max y}$ as Equation (3) cannot be satisfied by $\ell$. Indeed, it is easy to construct situations where $\hat{y}_1 = R_{\boldsymbol{\theta}}(x_1)$ is arbitrarily close to $\hat{y}_2 = R_{\boldsymbol{\theta}}(x_2)$ with $x_2$ correctly classified, while $x_1$ is not (just take $x_2$ on the boundary decision of the network and $x_1$ on the wrong side of the boundary), so that the left-hand side is equal to 1 and the right-hand side is arbitrarily small. Thus, there is no finite $L > 0$ that could satisfy Equation (3).

## I    THE MARGIN-LOSS IS LIPSCHITZ

For $\hat{y} \in \mathbb{R}^{d_{\text{out}}}$ and a one-hot encoding $y \in \mathbb{R}^{d_{\text{out}}}$ of the class $c$ of $x$ (meaning that $y_{c'} = \mathbb{1}_{c'=c}$ for every $c'$), the margin $M(\hat{y}, y)$ is defined by

$$M(\hat{y}, y) := [\hat{y}]_c - \max_{c' \neq c} [\hat{y}]_{c'}.$$

For $\gamma > 0$, recall that the $\gamma$-margin-loss is defined by

$$\ell(\hat{y}, y) = \begin{cases} 0 & \text{if } \gamma < M(\hat{y}, y), \\ 1 - \frac{M(\hat{y}, y)}{\gamma} & \text{if } 0 \leqslant M(\hat{y}, y) \leqslant \gamma, \\ 1 & \text{if } M(\hat{y}, y) < 0. \end{cases} \tag{22}$$

For any class $c$ and one-hot encoding $y$ of $c$, it is known that $\hat{y} \in \mathbb{R}^{d_{\text{out}}} \mapsto M(\hat{y}, y)$ is 2-Lipschitz with respect to the $L^2$-norm on $\hat{y}$ (Bartlett et al., 2017, Lemma A.3). Moreover, the function

$$r \in \mathbb{R} \mapsto \begin{cases} 0 & \text{if } r < -\gamma, \\ 1 + \frac{r}{\gamma} & \text{if } -\gamma \leqslant r \leqslant 0, \\ 1 & \text{if } r > 0. \end{cases}$$

is $\frac{1}{\gamma}$-Lipschitz. By composition, this shows that $\hat{y} \in \mathbb{R}^{d_{\text{out}}} \mapsto \ell_\gamma(\hat{y}, y)$ is $\frac{2}{\gamma}$-Lipschitz with respect to the $L^2$-norm.

*Proof of Theorem 3.2.* Since the labels $\mathbf{Y}$ are one-hot encodings, we equivalently consider $\mathbf{Y}$ either in $\mathbb{R}^{d_{\text{out}}}$ or in $\{1, \ldots, d_{\text{out}}\}$. It holds (Bartlett et al., 2017, Lemma A.4)

$$\mathbb{P}\left( \arg\max_c [R_{\boldsymbol{\theta}}(\mathbf{X})]_c \neq \mathbf{Y} \right) \leqslant \mathbb{E}\left( \ell_\gamma(R_{\boldsymbol{\theta}}(\mathbf{X}), \mathbf{Y}) \right)$$

for any $\gamma > 0$ and associated $\gamma$-margin-loss $\ell_\gamma$. Thus, considering the generalization error for $\ell_\gamma$:

$$\mathbb{P}\left( \arg\max_c [R_{\boldsymbol{\theta}}(\mathbf{X})]_c \neq \mathbf{Y} \right) \leqslant \underbrace{\frac{1}{n} \sum_{i=1}^{n} \ell_\gamma \left( R_{\hat{\boldsymbol{\theta}}(\mathbf{Z})}(\mathbf{X}_i), \mathbf{Y}_i \right)}_{= \text{ training error of } \hat{\boldsymbol{\theta}}(\mathbf{Z})} + \mathbb{E}_{\mathbf{Z}} \, \ell_\gamma\text{-generalization error of } \hat{\boldsymbol{\theta}}(\mathbf{Z}).$$

By definition of $\ell_\gamma$, the training error of $\hat{\boldsymbol{\theta}}(\mathbf{Z})$ is at most $\frac{1}{n} \sum_{i=1}^{n} \mathbb{1}_{[R_{\hat{\boldsymbol{\theta}}(\mathbf{Z})}(\mathbf{X}_i)]_{\mathbf{Y}_i} \leqslant \gamma + \max_{c \neq \mathbf{Y}_i} [R_{\hat{\boldsymbol{\theta}}(\mathbf{Z})}(\mathbf{X}_i)]_c}$. Moreover, Theorem 3.1 can be used to bound the generalization error associated with $\ell_\gamma$ with $L = 2/\gamma$. This proves the claim. $\qquad\square$

## J    DETAILS ON THE EXPERIMENTS OF SECTION 4

**Details for Table 2.** All the experiments are done on ImageNet-1k using 99% of the 1,281,167 images of the training set for training, the other 1% is used for validation. Thus, $n = 1268355 = \lfloor 0.99 \times 1281167 \rfloor$ in our experiments, $d_{\text{in}} = 224 \times 224 \times 3 = 150528$, $d_{\text{out}} = 1000$. We also estimated $B = 2.640000104904175$ by taking the maximum of the $L^\infty$ norms of the training images normalized for inference[9]. The PyTorch code for normalization at inference is standard:

```
1  inference_normalization = transforms.Compose([
2        transforms.Resize(256),
3        transforms.CenterCrop(224),
4        transforms.ToTensor(),
5        transforms.Normalize(mean=[0.485, 0.456, 0.406], std=[0.229,
      0.224,  0.225]),
6      ])
```

---

[9]The constant $\sigma$ in Theorem 3.1 corresponds to data $\mathbf{Z}_i$ drawn from the distribution for which we want to evaluate the test error. This is then the data normalized for inference. Thus, the training loss appearing in Theorem 3.1 is also evaluated on the training data $\mathbf{Z}_i$ normalized for inference. In the experiments, we ignore this fact and still evaluate the training loss on the data augmented for training. Moreover, note that it is not possible to recover the training images augmented for training from the images normalized for inference, because cropping is done at random for training. Thus, the real life estimator is not a function of the images $\mathbf{Z}_i$ normalized for inference, and Theorem 3.1 does not apply stricto sensu. This fact is ignored here.

We consider ResNets. They have a single max-pooling layer of kernel size $3 \times 3$ so that $K = 9$. The depth is $D = 3 + \#$ basic blocks $\times \#$ conv per basic block, where 3 accounts for the conv1 layer, the average-pooling layer, the fc layer, and the rest accounts for all the convolutional layers in the basic blocks. Note that the average-pooling layer can be incorporated into the next fc layer as it only contains identity neurons, see the discussion after Theorem 3.1, so we can actually consider $D = 2 + \#$ basic blocks $\times \#$ conv per basic block. Table 4 details the relevant values related to basic blocks.

Table 4: Number of basic blocks, of convolutional layer per basic blocks and associated $D$ for ResNets (He et al., 2016, Table 1).

| ResNet | 18 | 34 | 50 | 101 | 152 |
|---|---|---|---|---|---|
| # basic blocks | 8 | 16 | | 33 | 50 |
| # conv per basic block | 2 | | | 3 | |
| $D$ | 18 | 34 | 50 | 101 | 152 |

**Pretrained ResNets.** The PyTorch pretrained weights that have been selected are the ones with the best performance: `ResNetX_Weights.IMAGENET1K_V1` for ResNets 18 and 34, and `ResNetX_Weights.IMAGENET1K_V2` otherwise.

**Choice of $\gamma > 0$ for Theorem 3.2.** In Equation (4), note that there is a trade-off when choosing $\gamma > 0$. Indeed, the first term of the right-hand side is non-decreasing with $\gamma$ while the second one is non-increasing. The first term is simply the proportion of datapoints that are not correctly classified with a margin at least equal to $\gamma$. Defining the margin of input $i$ on parameters $\boldsymbol{\theta}$ to be $R_{\boldsymbol{\theta}}(\mathbf{X}_i)_{\mathbf{Y}_i} - \arg\max_{c \neq \mathbf{Y}_i} R_{\boldsymbol{\theta}}(\mathbf{X}_i)_c$, this means that the first term is (approximately) equal to $q$ if $\gamma = \gamma(q)$ is the $q$-quantile of the distribution of the margins over the training set.

Note that since the second term in Equation (4) is of order $1/\sqrt{n}$, it would be desirable to choose the $1/\sqrt{n}$-quantile (up to a constant) for $\gamma$. However, this is not possible in practice as soon as the training top-1 accuracy is too large compared to $1/\sqrt{n}$ (eg. on ImageNet). Indeed, if the training top-1 error is equal to $e \in [0, 1]$, then at least a proportion $e$ of the data margins should be negative[10] so that any $q$-quantile with $q < e$ is negative and cannot be considered for Theorem 3.2

The distribution of the margins on the training set of ImageNet can be found in Figure 3. The maximum training margin is roughly of size 30, which is insufficient to compensate the size of the $L^1$ path-norm of pretrained ResNets reported in Table 3. For $\gamma > 30$, the first term of the right-hand side of Theorem 3.2 is greater than one, so that the bound is not informative. This shows that there is no possible choice for $\gamma > 0$ that makes the bound informative on these pretrained ResNets. Table 5 reports a quantile for these pretrained ResNets.

Table 5: The $q$-quantile $\gamma(q)$ for $q = \frac{1}{3}e + \frac{2}{3}$, with $e$ being the top-1 error, on ImageNet, of pretrained ResNets available on PyTorch.

| ResNet | 18 | 34 | 50 | 101 | 152 |
|---|---|---|---|---|---|
| $\gamma(q)$ | 5.0 | 5.6 | 4.2 | 5.6 | 5.8 |

**Details for sparse networks.** ResNet18 is trained on $99\%$ of ImageNet with a single GPU using SGD for 90 epochs, learning rate 0.1, weight-decay 0.0001, batch size 1024, and a multi-step scheduler where the learning rate is divided by 10 at epochs 30, 60 and 80. The epoch out of the 90 ones with maximum validation top-1 accuracy is considered as the final epoch. Pruning is done iteratively accordingly to Frankle et al. (2021). We prune $20\%$ of the remaining weights of each convolutional layer, and $10\%$ of the final fully connected layer, at each pruning iteration, save the mask and rewind the weights to their values after the first 5 epochs of the dense network, and train for 85 remaining epochs, before pruning again etc. Results for a single run are shown in Figure 4.

**Details for increasing the train size.** Instead of training on $99\%$ of ImageNet ($n = 1268355$), we trained a ResNet18 on $n/2^k$ samples drawn at random, for $1 \leqslant k \leqslant 5$. For each given $k$, the results

---

[10]A data margin is negative if and only if it is misclassified.

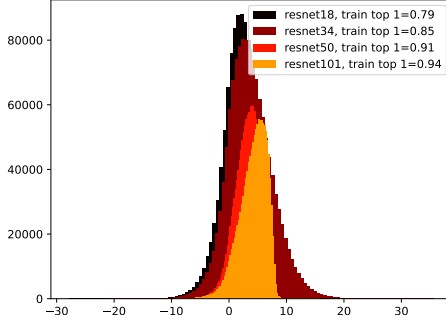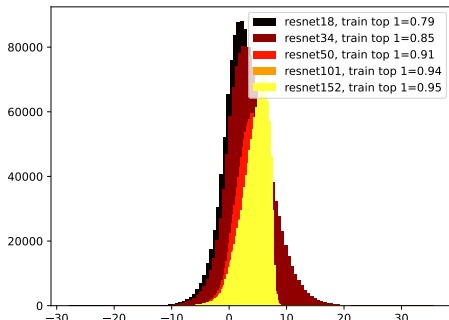

Figure 3: Distribution of the margins on the training set of ImageNet, with the pretrained ResNets available on PyTorch.

are averaged over 3 seeds. The hyperparameters are the same as for sparse networks (except that we do not perform any pruning here): 90 epochs etc. Results are in Figure 5.

## K    MAIN RELATED WORKS

Given the extent of the literature on generalization bounds, we apologize in advance for papers the reader may find missing below.

**Previous definitions of the path-lifting and the path-activations** In the work of Kawaguchi et al. (2017, Section 5.1) the path-lifting and the path-activations are evoked in the case of a ReLU DAG with max-pooling neurons *and no biases*, but with no explicit definitions. Definition A.3 gives a formal definition for these objects, and extend it to the case where there are biases, which requires extending the path-lifting to paths starting from neurons that are not input neurons. Moreover, Definition A.3 extends it to arbitrary $k$-max-pooling neurons (classical max-pooling neurons correspond to $k = 1$).

Note also that the formula Equation (5) is stated in the specific case of Kawaguchi et al. (2017, Section 5.1) (as an explicit sum rather than an inner product), without proof since the objects are not explicitly defined in Kawaguchi et al. (2017).

A formal definition of the path-lifting is given in the specific case of layered fully-connected ReLU neural networks with biases in the work of Stock & Gribonval (2023, Definition 6). Moreover, it is proved that Equation (5) holds *in this specific case* in Stock & Gribonval (2023, Corollary 3). Definition A.3 and Equation (5) generalize the latter to an arbitrary DAG with $*$-max-pooling or identity neurons (allowing in particular for skip connections, max-pooling and average-pooling).

The rest of the works we are aware of only define and consider the norm of the path-lifting, but not the lifting itself. The most general setting being the one of Neyshabur et al. (2015) with a general DAG, *but without max or identity neurons, nor biases*. Not defining the path-lifting and the path-activations makes notations arguably heavier since Equation (5) is then always written with an explicit sum over all paths, with explicit product of weights along each path, and so on.

**Previous generalization bounds based on path-norm** See Table 1 for a comparison. Appendix L tackles some other bounds that do not appear in Table 1.

**Empirical evaluation of path-norm** The formula given in Theorem A.1 is the first one to fully encompass modern networks with biases, average/$*$-max-pooling, and skip connections, such as ResNets. An equivalent formula is stated for *layered fully-connected* ReLU networks *without biases* (and *no pooling/skip connections*) in Dziugaite et al. (2020, Appendix C.6.5) and Jiang et al. (2020, Equations (43) and (44)) but without proof (and only for $L^2$-path-norm instead of general mixed $L^{q,r}$ path-norm). Actually, *this equivalent formula turns out to be false when there are $*$-max-pooling neurons* as one must replace $*$-max-pooling neurons with identity ones, see Theorem A.1. Care must

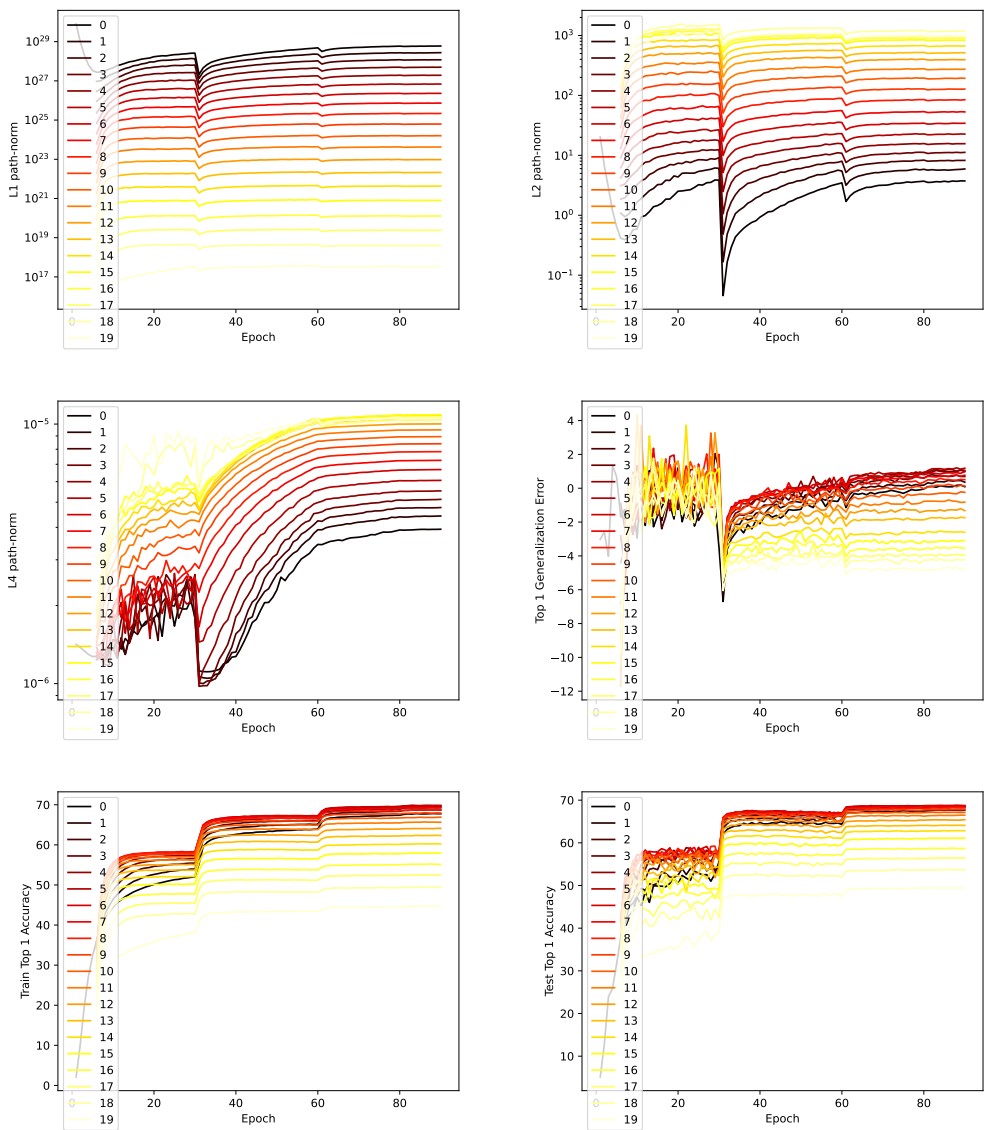

Figure 4: $L^q$ path-norm ($q = 1, 2, 4$), test top-1 accuracy, training top-1 accuracy, and the top-1 generalization error (difference between test top-1 and train top-1) during the training of a ResNet18 on ImageNet. The pruning iteration is indicated in legend, with $0$ corresponding to the dense network. The color also indicates the degree of sparsity: from dense (black) to extremely sparse (yellow).

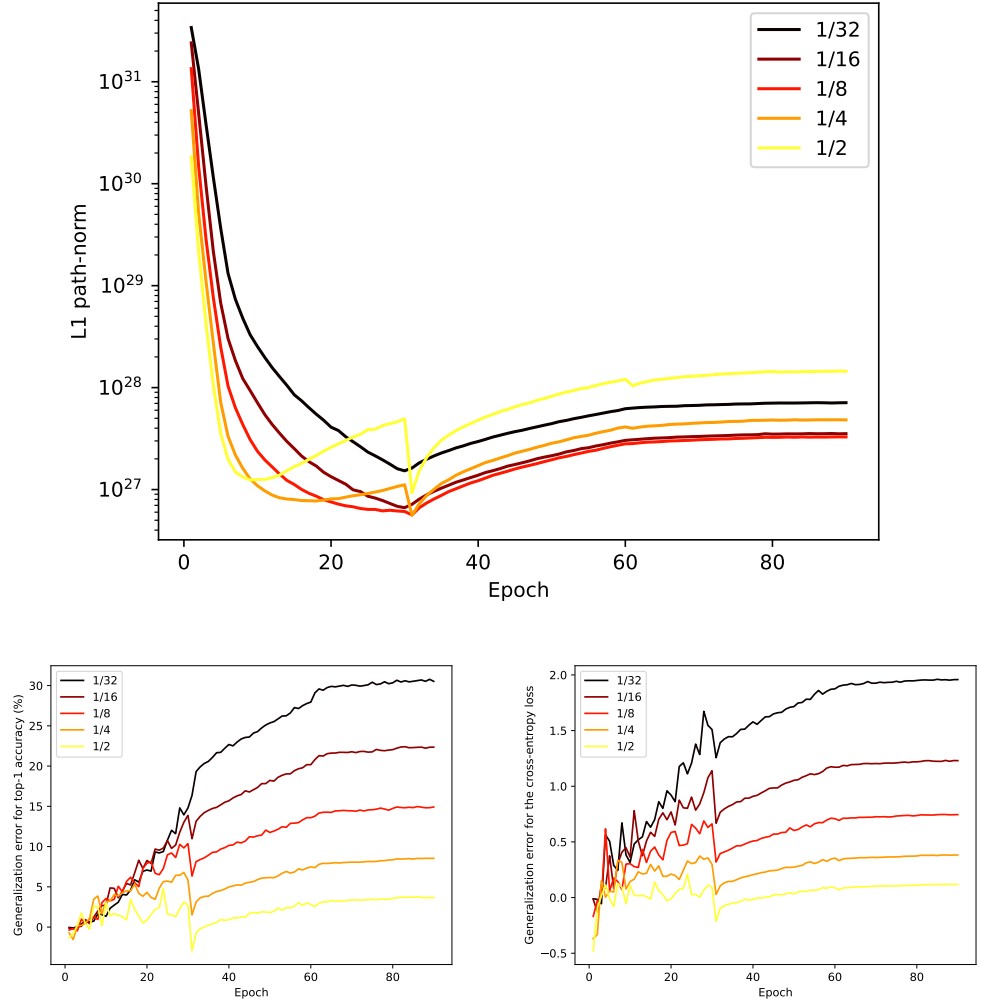

Figure 5: $L^1$ path-norm, and empirical generalization errors for both the top-1 accuracy and the cross-entropy during the training of a ResNet18 on a subset of the training images of ImageNet. The legend indicates the size of the subset considered, *e.g.* $1/m$ corresponds to $1/m$ of $99\%$ of ImageNet, leaving the other $1\%$ out for validation. The color also indicates the size of the subset: from small (black) to large (yellow).

also be taken with average-pooling neurons that must be rescaled by considering them as identity neurons.

We could not find reported numerical values of the path-norm except for toy examples (Dziugaite, 2018; Furusho, 2020; Zheng et al., 2019). Details are in Appendix L.

Appendix L also discusses 1) inherent limitations of Theorem 3.1 which are common to many generalization bounds, and 2) the applicability of Theorem 3.1 compared to PAC-Bayes bounds.

## L    MORE RELATED WORKS

**More generalization bounds using path-norm** In E et al. (2022) is established an additional bound to the one appearing in Table 1, for a class of functions and a complexity measure that are related to the *infinite-depth* limits of residual networks and the path-norm. However, it is unclear how this result implies anything for actual neural networks with the actual path-norm [11].

The bound in Zheng et al. (2019) only holds for layered fully-connected ReLU neural networks (*no max or identity neurons*) with *no biases* and *it grows exponentially* with the depth of the network. It is not included in Table 1 because it requires an additional assumption: the coordinates of the path-lifting must not only be bounded from above, but also *from below*. The reason for this assumption is not discussed in Zheng et al. (2019), and it is unclear whether this is at all desirable, since a small path-norm can only be better for generalization in light of Theorem 3.1.

Theorem 4.3 in Golowich et al. (2018) is a bound that holds for layered fully-connected ReLU neural networks *with no biases (no max and no identity neurons)* and it *depends on the product of operator norms of the layers*. It has the merit of having no dependence on the size of the architecture (depth, width, number of neurons etc.). However, it requires an additional assumption: each layer must have an operator norm bounded from below, so that it only applies to a restricted set of layered fully-connected networks. Moreover, it is unclear whether such an assumption is desirable: there are networks with arbitrary small operator norms that realize the zero function, and the latter has a generalization error equal to zero.

Theorem 8 in Kawaguchi et al. (2017) gives a generalization bound for *scalar-valued* ($d_{\text{out}} = 1$) models with an output of the form $\langle \Phi(\boldsymbol{\theta}), \boldsymbol{A}(\boldsymbol{\theta}', x)x \rangle$ for some specific parameters $\boldsymbol{\theta}, \boldsymbol{\theta}'$ that have no reason to be equal. This is orthogonal to the case of neural networks where one must have $\boldsymbol{\theta} = \boldsymbol{\theta}'$, and it is therefore not included in Table 1. Theorem 5 in Kawaguchi et al. (2017) can be seen as a possible first step to derive a bound based on path-norm in the specific case of the mean squared error loss. However, as discussed in more details below, Theorem 5 in Kawaguchi et al. (2017) is a rewriting of the generalization error with several terms that are as complex to bound as the original generalization error, resulting in a bound being as hard as the generalization error to evaluate/estimate.

**More details about Theorem 5 in Kawaguchi et al. (2017)** We start by re-deriving Theorem 5 in Kawaguchi et al. (2017). In the specific case of mean squared error, using that

$$\|R_{\boldsymbol{\theta}}(x) - y\|_2^2 = \|R_{\boldsymbol{\theta}}(x)\|_2^2 + \|y\|_2^2 - 2\langle R_{\boldsymbol{\theta}}(x), y \rangle,$$

---

[11] E et al. (2022) starts from the fact that the infinite-depth limits of residual networks can be characterized with partial differential equations. Thus E et al. (2022) establishes a bound for functions characterized by similar, but different, partial differential equations, using what seems to be an analogue of path-norm for these new functions. However, even if the characterizations of these functions are closed, as it is said in E et al. (2022), "it is unclear how the two spaces are related".

it is possible to rewrite the generalization error as follows:

$$\text{generalization error of } \hat{\boldsymbol{\theta}}(\mathbf{Z}) = \mathbb{E}\left(\|R_{\hat{\boldsymbol{\theta}}(\mathbf{Z})}(\tilde{\mathbf{X}}) - \tilde{\mathbf{Y}}\|_2^2 | \mathbf{Z}\right) - \frac{1}{n}\sum_{i=1}^{n}\|R_{\hat{\boldsymbol{\theta}}(\mathbf{Z})}(\mathbf{X}_i) - \mathbf{Y}_i\|_2^2$$

$$= \mathbb{E}\left(\|R_{\hat{\boldsymbol{\theta}}(\mathbf{Z})}(\tilde{\mathbf{X}})\|_2^2 | \mathbf{Z}\right) - \frac{1}{n}\sum_{i=1}^{n}\|R_{\hat{\boldsymbol{\theta}}(\mathbf{Z})}(\mathbf{X}_i)\|_2^2$$

$$+ \mathbb{E}\left(\|\tilde{\mathbf{Y}}\|_2^2 | \mathbf{Z}\right) - \sum_{i=1}^{n}\|\mathbf{Y}_i\|_2^2$$

$$- 2\mathbb{E}\left(\left\langle R_{\hat{\boldsymbol{\theta}}(\mathbf{Z})}(\tilde{\mathbf{X}}), \tilde{\mathbf{Y}}\right\rangle | \mathbf{Z}\right) - \frac{1}{n}\sum_{i=1}^{n}\left\langle R_{\hat{\boldsymbol{\theta}}(\mathbf{Z})}(\mathbf{X}_i), \mathbf{Y}_i\right\rangle.$$

It is then possible to make the $L^2$ path-norm appear. For instance, for one-dimensional output networks, it can be proven (see Lemma A.1) that $R_{\boldsymbol{\theta}}(x) = \langle \Phi(\boldsymbol{\theta}), z(x, \boldsymbol{\theta}) \rangle$ with $\Phi(\boldsymbol{\theta})$ the path-lifting of parameters $\boldsymbol{\theta}$, and $z$ that typically depends on the path-activations and the input, so that the first term above can be rewritten

$$\Phi(\hat{\boldsymbol{\theta}}(\mathbf{Z}))^T \left( \mathbb{E}\left( z(\tilde{\mathbf{X}}, \hat{\boldsymbol{\theta}}(\mathbf{Z})) z(\tilde{\mathbf{X}}, \hat{\boldsymbol{\theta}}(\mathbf{Z}))^T | \mathbf{Z} \right) - \frac{1}{n}\sum_{i=1}^{n} z(\mathbf{X}_i, \hat{\boldsymbol{\theta}}(\mathbf{Z})) z(\mathbf{X}_i, \hat{\boldsymbol{\theta}}(\mathbf{Z}))^T \right) \Phi(\hat{\boldsymbol{\theta}}(\mathbf{Z})).$$

Let us call a "generalization error like quantity" any term of the form

$$\mathbb{E}\left( f_{\hat{\boldsymbol{\theta}}(\mathbf{Z})}(\tilde{\mathbf{Z}}) | \mathbf{Z} \right) - \frac{1}{n}\sum_{i=1}^{n} f_{\hat{\boldsymbol{\theta}}(\mathbf{Z})}(\mathbf{Z}_i),$$

that is, any term that can be represented as a difference between the estimator learned from training data $\mathbf{Z}$ evaluated on test data $\tilde{\mathbf{Z}}$, and the evaluation on the training data. We see that the derivation above replaces the classical generalization error with two others quantities similar in definition to the generalization error. This derivation, which is specific to mean squared error, leads to Theorem 5 in Kawaguchi et al. (2017). Very importantly, note that this derivation trades a single quantity similar to generalization error for two new such quantities. There is no discussion in Kawaguchi et al. (2017) on how to bound these two new terms. Without any further new idea, there is no other way than the ones developed in the literature so far: reduce the problem to bounding a Rademacher complexity (as it is done in Theorem 3.1), or use the PAC-Bayes framework, and so on.

**More on numerical evaluation of path-norm** In Dziugaite (2018, Section 2.9.1) is reported numerical evaluations after 5 epochs of SGD on a one hidden layer network trained on a binary variant of MNIST. Furusho (2020, Figure 9 and Section 3.3.1) deals with 1d regression with 5 layers and 100 width. Experiments in Zheng et al. (2019) are on MNIST. Note that it is not clear whether the path-norm used in Zheng et al. (2019) corresponds to the one defined in Definition A.3. Indeed, the references given in Zheng et al. (2019) for the definition of the path-norm are both Neyshabur et al. (2015) and Neyshabur et al. (2017), but these two papers have two different definitions of the path-norm. The one in Neyshabur et al. (2015) corresponds to the norm of the path-lifting as defined in Definition A.3 (but in simpler settings: no pooling etc.), while the one in Neyshabur et al. (2017) corresponds to the latter divided by the margin of the estimator.

For completeness, let us also mention that it is reported in Dziugaite et al. (2020); Jiang et al. (2020) whether the path-norm correlates with the empirical generalization error or not, but there is no report of numerical values of the path-norm. In Neyshabur et al. (2017) are reported the quotient of path-norms with margins, but not the path-norms alone.

**Inherent limitations of uniform convergence bounds** Theorem 3.1 has some inherent limitations due to its nature. It is *data-dependent* as it depends on the input distribution. However, it does not depend on the label distribution, making it uninformative as soon as $\Theta$ is so much expressive that it can fit random labels. Networks that can fit random labels have already been found empirically (Zhang et al., 2021), and it is open whether this stays true with a constraint on the path-norm.

Theorem 3.1 is *based on a uniform convergence bound*[12] as any other bound also based on a control of a Rademacher complexity. Nagarajan & Kolter (2019) empirically argue that even the tightest

---

[12] A uniform convergence bound on a model class $\boldsymbol{F}$ is a bound on $\mathbb{E}_{\mathbf{Z}} \sup_{f(Z) \in \boldsymbol{F}}$ generalization error $f(Z)$. This worst-case type of bound can lead to potential limitations when $\boldsymbol{F}$ is too expressive.

uniform convergence bound holding with high probability must be loose on some synthetic datasets. If this was confirmed theoretically, this would still allow uniform bounds to be tight when considering other datasets than the one in Nagarajan & Kolter (2019), such as real-world datasets, or when the estimator considered in Nagarajan & Kolter (2019) is not in $\Theta$ (for instance because of constraints on the slopes via the path-norm).

Finally, Theorem 3.1 can provide theoretical guarantees on the generalization error of the output of a learning algorithm, but only *a posteriori*, after training. In order to have a priori guarantees, one should have to derive a priori knowledge on the path-norm at the end of the learning algorithm.

**Comparison to PAC-Bayes bounds** Another interesting direction to understand generalization of neural networks is the PAC-Bayes framework (Guedj, 2019; Alquier, 2021). Unfortunately, PAC-Bayes bounds cannot be exactly computed on networks that are trained in a usual way. Indeed, these bounds typically involve a KL-divergence, or something related, for which there is no closed form except for very specific distributions (iid Gaussian/Cauchy weights...) that do not correspond to the distributions of the weights after a usual training[13][14]. We are aware of two research directions that try to get over this issue. The first way is to change the learning algorithm by enforcing the weights to be iid normally distributed, and then optimize the parameters of these normal distributions, see for instance the pioneer work of Dziugaite & Roy (2017). The merit of this new learning algorithm is that it has explicitly been designed with the goal of having a small generalization error. Practical performance are worse than with usual training, but this leads to networks with an associated non-vacuous generalization bound. To the best of our knowledge, this is the only way to get a non-vacuous bound[15], and unfortunately, this does not apply to usual training. The second way to get over the intractable evaluation of the KL-divergence is to 1) try to approximate the bound within reasonable time, and 2) try to quantify the error made with the approximation (Pérez & Louis, 2020). Unfortunately, to the best of our knowledge, approximation is often based on a distribution assumption of the weights that is not met in practice (*e.g.* iid Gaussian weights), approximation is costly, and the error is unclear when applied to networks trained usually. For instance, the bound in Pérez & Louis (2020, Section 5) 1) requires at least $O(n^2)$ operations to be evaluated, with $n$ being the number of training examples, thus being prohibitive for large $n$ (Pérez & Louis, 2020, Section 7), and 2) it is unclear what error is being made when using a Gaussian process as an approximation of the neural network learned by SGD.

---

[13]The randomness of the weights after training comes from the random initialization and the randomness in the algorithm (*e.g.* random batch in SGD).

[14]For instance, independence is not empirically observed, see Frankle et al. (2020, Section 5.2)

[15]Except, of course, for methods that are based on the evaluation of the performance on held-out data.

