# OpenReview forum: "A path-norm toolkit for modern networks: consequences, promises and challenges"
_ICLR.cc/2024/Conference — ICLR 2024 spotlight_

### Official Review · Reviewer_m3hb · 2023-11-01

**Soundness:** 3 good
**Presentation:** 3 good
**Contribution:** 3 good
**Rating:** 6
**Confidence:** 3

**Summary:**

Motivated by the fact that path-norms provide a way to bound generalization errors and satisfy several nice properties (e.g. easy to compute, invariant under parameter rescalings and neuron permutations, etc), the paper introduces a notion of path-norms that can handle general ReLU networks with biases, skip connections, and max pooling. Furthermore, they provide path-norm based generalization bounds that match or beat existing path-norm based generalization bounds. Along the way, they prove new peeling and contraction lemmas for computing Rademacher complexities.

**Strengths:**

Peeling and contraction lemmas for computing Rademacher complexities for more complex networks are novel and interesting.

The notion of path-norms the paper introduces can handle very general ReLU networks, and the generalization bounds depending on the more general path-norm do better or recover existing path-norm based generalization bounds.

**Weaknesses:**

The paper establishes generalization bounds with respect to $L^1$ path-norms (as opposed to general $L^p$ path-norms).
In the experiments, $L^1$ path-norms tend to be extremely large (at least $10^{30}$) which makes the generalization bounds vacuous.

**Questions:**

Can similar generalization bounds be proven for more general $L^p$ path-norms without incurring a large dimension dependence?

---

> ### Author Response · Authors · 2023-11-17
>
> Thank you for your review.
>
> **Weaknesses.** Thank you for your comment. Establishing generalization bounds based on general $L^p$-norms would be very interesting given their magnitude order reported in the experiments but it is still a major open question in the literature. Apart from the bounds that derive from the $L^1$-bound using basic comparisons between $L^p$-norms, we believe that non-trivial bounds can only be obtained if the activations are taken more carefully into account, the latter being a non-trivial matter. We plan to investigate that in the future.
>
> Regarding the size of the $L^1$ path-norm, the main point of the document is precisely to assess and report these *existing* weaknesses *of current theory* for the first time on modern neural networks.
>
> **Questions.** As already mentioned, establishing bounds involving general $L^p$-path-norms is currently an open question. We think that in order to avoid large dependencies in the dimension, the activations have to be taken more carefully into account (see also the conclusion). We plan to investigate this in the future.

---

### Official Review · Reviewer_fyuf · 2023-11-06

**Soundness:** 4 excellent
**Presentation:** 4 excellent
**Contribution:** 3 good
**Rating:** 8
**Confidence:** 4

**Summary:**

This paper presents the state of art result on generalizaiton bound of deep neural networks using path norm. In detail, it allows to consider any neural netowrks whose backbone is a DAG and the activation function is either identity, ReLU, or max pooling. Because this result handles any DAG, one can append a dimension of value equal to constant 1 for all data to extend the analysis networks with bias. Because the theoretical result in thie paper is so general, for the first time, the authors are able to evaluate the actual generalization bound on ResNet trained on Imagenet. Though the path norm based bound presented in this paper is still vacuous, this paper contains several instereting experimental findings and shed light to future researches.

**Strengths:**

1. This paper is very well-written. It is very easy for me to understand the main result and the main theoretical challenges to overcome. The notations are consistent and the mathematical statements are precise. I checked the proofs of several core lemmas and they all look correct to me.

2. The result presented in this paper is very elegant, which unifies several previous results on path-norm. The core proof strategy is a new peeling argument, which relates the Rademacher complexity over a set of nodes to that of the incoming nodes of the set of nodes. The proof uses several new lemmas, which generalizes previous results.

3. The authors not only derive a better and unified path-norm generalization bound but also apply their bound on real datasets and architectures, i.e. ResNets of various sizes on Imagenets. Though their experimental results indicate that the new path norm-based bounds are still vacuous, they note that sparse networks greatly reduce the bound, which may shed light for future research.

**Weaknesses:**

1. I think Setting 1 of Lemma C.1 can be obtained by a simple application of the well-known scalar contraction inequality (Ledoux & Talagrand, 1991). Given $v\in V$ and $t\in T^v$, we can embed it into a higher dimensional space $\mathbb{R}^{I\times T}$ by defining $\tilde t(v,t) = [0, 0 ,\ldots, v,\ldots, 0]$ as a $I\times T$ matrix. We further define $\tilde f_{i,v}(x) = f_i(x)$ for all $x\in\mathbb{R}$. Therefore we have the following which is exactly the conclusion of Setting 1 of Lemma C.1

$$\mathbb{E} \max_{v \in V} \sup_{t \in T^v} G \left( \sum_{i \in I} \varepsilon_{i,v} f_i(t_i) \right) $$

$$=\mathbb{E} \max_{v \in V} \sup_{t \in T^v} G \left( \sum_{i \in I} \varepsilon_{i,v} f_i([\tilde t(v,t)] _{i,v}) \right)$$

$$=\mathbb{E} \max_{v \in V} \sup_{t \in T^v} G \left( \sum_{i \in I, v' \in V} \varepsilon_{i,v'} \tilde f_{i,v'}([\tilde t(v,t)] _{i,v'}) \right)$$

$$= \mathbb{E} \max_{v \in V} \sup_{t \in T^v} G \left( \sum_{i \in I, v' \in V} \varepsilon_{i,v'} [\tilde t(v,t)] _{i,v'} \right) $$

$$=\mathbb{E} \max_{v \in V} \sup_{t \in T^v} G \left( \sum_{i \in I} \varepsilon_{i,v} [\tilde t(v,t)] _{i,v} \right)$$

$$=\mathbb{E} \max_{v \in V} \sup_{t \in T^v} G \left( \sum_{i \in I} \varepsilon_{i,v} t_i \right) $$

**Questions:**

1. measurable condition in Lemma D.1 seems unnecessary?

2. Why 2-norm of $\Phi(\theta)$ is relevant? Isn't it small just because the number of paths is huge? It doesn't seem to be a valid generalization bound even for a very simple 2-layer linear network. Is there any real evidence that 2-norm of $\Phi(\theta)$ could be useful for indicating generalization?

3. Some minor typos: (1). $\overline N_{in}$ seems not defined in appendix A; (2). The third row of the first equation on page 29 seems missing a factor of $5^d$

4. I wonder if the authors can just ignore all the linear activations by linking the incoming nodes of the current nodes to the outgoing nodes of the current node. This should not change the path norm. If this is true, then we should not count the number of Conv layers in ResNet, but the max number of ReLU and max pooling along the path, which is smaller than the former for ResNets.

5. Though the writing quality is already great, I still feel like the paper can benefit from first providing a result without any bias, which would greatly simplify the notations in the theoretical statement and the proof. Then the authors can point out that the results can be generalized to networks with bias by appending a constant 1 dimension in the input (as it is done now), adding an edge between the constant 1 input dimension to the neuron, and then calling the main result. Writing the theorem in this way would clearly explain to the readers where the plus one in $d_{in}+1$ and the $\max(n,\ldots)$ in $\sigma$ comes from. (Theorem 3.1)


6. The statement of Lemma D.2 is a little bit confusing because $\Theta$ is not arbitrary, but the one preprocessed by Algorithm 1. Maybe it is more clear to just define $r$ as the sup of 1-norm of the last layer of $\Theta$? A similar issue occurs for Lemma D.3, where the authors can just say the 1-norm of incoming weights of non-output node in $\Theta$ are bounded. In this way, Lemma D.2 and Lemma D.2 will together motivate Algorithm 1 and the path norm as a generalization metric.

---

> ### Author Response · Authors · 2023-11-17
>
> Thank you very much for your review and for the time you devoted to check as many details as you could in the appendices.
>
> **Weaknesses.**
> Thank you very much for your comment, one can indeed reduce to (Ledoux and Talagrand, 1991) in the more elegant and easier way you describe. However, this proves only the case where the $t_i$ are scalar because (Ledoux and Talagrand, 1991) does not apply to the general vector-valued case. Thus, it still remains to prove a similar contraction lemma in the case of max-pooling neurons. In this situation, the same argument would basically correspond to the part of the proof that reduces to the case $|V|=1$ and then, it still remains a significant part of work. All of this will be mentioned in the final version.
>
> **Questions.**
> 1. Thank you for your remark. What is actually needed is that the expectations are well-defined. For that, we will rather assume in the final version of the document that the suprema are measurable (not only $G$). Everything being non-negative, the expectations will make sense in $[0,\infty]$.
> 2. This is an interesting and open question that we plan to investigate in the future. The experiments report the values of $L^p$-path-norms for $p\neq 1$ for the simple reason that it is the first time they are computed on modern neural networks. However, the relevance of general $L^p$-path-norm for theory and practice is not clear for now. What is known is that: 1) in practice, $L^2$-path-norm is found to empirically correlate better with the generalization error than the $L^1$-path-norm (Jiang et al, 2020), and 2) because of basic comparisons between $L^p$-norms, any $L^1$-path-norm bound yield $L^p$-path-norm bounds ($p>1$) but with large dimensional constants. We think that these dimensional dependencies could be avoided by carefully taking into account the activations (for now, the bound corresponds to the worst case where all paths are activated), see also the discussion in the conclusion.
> 3. Thank you, this will be corrected in the final version.
> 4. You are right, thank you. This will be added in the final version. For experiments, this allows us to reduce by one the depth $D$ by incorporating the average-pooling layer into the next fully connected layer of ResNets (we still count one for each Conv Layer as each such layer ends up with a ReLU). The numerical results will be updated accordingly (it will not change the associated message). We will still keep the definition of the model as it is (with identity neurons) as we believe it to be clearer for implementing operations such as average-pooling.
> 5. Thank you for your suggestion. The final version will contain the statement without biases, and we will mention how to adapt the result in the case with biases. The proof will still be given in the general case directly with biases.
> 6. We agree with your suggestion and this will be taken into account in the final version.

---

> > ### Comment · Reviewer_fyuf · 2023-12-05
> >
> > Thanks for your response. I will keep my rating.
> >
> > Regarding the measurable condition in Lemma D.1, I mean this is a distribution of finite support, so every function should be measurable?

---

### Official Review · Reviewer_wKAQ · 2023-11-15

**Soundness:** 4 excellent
**Presentation:** 4 excellent
**Contribution:** 3 good
**Rating:** 8
**Confidence:** 4

**Summary:**

The author extends the embedding in "Stock & Gribonval, 2022" to handle max pooling, averaging and bias in general neural network DAGs. The author uses the fact that neural networks are locally affine to introduce generalization bounds on modern architectures such as ResNets, VGGs, etc. The paper includes the embedding's definition and basic properties, the generalization bound and numerical experiments on multiple pre-trained ResNet models. It is evident from the experiments that the bound is tight and meaningful when the network is sparse, and irrelevant for deep dense networks. Despite this drawback, the work done in this paper takes the theory a step closer to practice.

**Strengths:**

The paper is well written and provide an important contribution for the understanding of generalization bounds. The included mathematical proofs and definitions are accessible and easy to read.

**Weaknesses:**

1) The numerical experiments were partial, and I would expect to see how the theoretical bound differs from an estimation of the generalization using some validation dataset.

2) "Proof of the Lipschitz Property" in page 14 ends with an incorrect statement (IMHO). The boundaries of the constant regions must be examined, and the proof holds because there is a finite number of such regions.

**Questions:**

No questions

---

> ### Author Response · Authors · 2023-11-17
>
> Thank you for your review and for the time you devoted to check the proofs.
>
> Note that even if the bound is smaller for sparse networks, it is an open question to know whether it can be improved in this regime or if it is already tight (the latter being unlikely in our opinion for all the different reasons given in the conclusion).
>
> **Weaknesses.**
> 1. In such a situation, the validation error is very close to the empirical test error so that the generalization error is well predicted using a validation set (with an error typically of the order of one or less for the top-1 accuracy), as it is usually done in practice.
> 2. Thank you for your remark. The final version will contain more details that were indeed missing here: this remains true on the borders by continuity, and it extends to the whole domain because between any pair of points, one changes of region only a finite number of times.

---

### Author Response · Authors · 2023-11-17

We thank the reviewers for their useful feedback and questions.

Beyond the modifications implied by the reviews, the final version of the document will have several typos fixed, and it will contain minor modifications in the proofs to make them easier to follow.

Moreover, we realized since the first submission that the very same proof also works for $k$-max-pooling neurons (that extracts the $k$-th largest component of its input, recovering max-pooling for $k=1$) with arbitrary $k$ rather than just $k=1$ as before. Dealing with general $k$-max-pooling neurons makes it possible to cover more recent activations such as GroupSort (which includes as a special case FullSort and MaxMin), see (Anil et al., 2019). Since the generalization bound can be immediately adapted for free to tackle this case, we thought that it would be a good idea to include it in the final version of the paper in order to save the readers the burden of rederiving a proper result while we could have done it easily. Of course, if you don't like this change, we can easily cancel it.

---

### Meta-Review · Area_Chair_5hwe · 2023-12-05

**Metareview:**

A work introducing a set of tools for investigating how sharp path-norms generalization bounds are for real modern neural networks. The resulting bounds improve over existing ones and further indicate inherent limitations to this statistical approach. The reviewers found both the theoretical and practical aspects of the work appealing, praised the clear writing, and unanimously voted for acceptance.

**Justification For Why Not Higher Score:**

Not too surprising, perhaps

**Justification For Why Not Lower Score:**

Limitations of a dominating statistical learning approach. Can also go for accept (poster).

---

### Decision · Program_Chairs · 2024-01-16

Accept (spotlight)